# Improved Scaling Laws in Linear Regression via Data Reuse

**Licong Lin**
UC Berkeley
liconglin@berkeley.edu

**Jingfeng Wu**
UC Berkeley
uuujf@berkeley.edu

**Peter L. Bartlett**
UC Berkeley and Google DeepMind
peter@berkeley.edu

## Abstract

Neural scaling laws suggest that the test error of large language models trained online decreases polynomially as the model size and data size increase. However, such scaling can be unsustainable when running out of new data. In this work, we show that data reuse can improve existing scaling laws in linear regression. Specifically, we derive sharp test error bounds on $M$-dimensional linear models trained by multi-pass *stochastic gradient descent* (multi-pass SGD) on $N$ data with sketched features. Assuming that the data covariance has a power-law spectrum of degree $a$, and that the true parameter follows a prior with an aligned power-law spectrum of degree $b - a$ (with $a > b > 1$), we show that multi-pass SGD achieves a test error of $\Theta(M^{1-b} + L^{(1-b)/a})$, where $L \lesssim N^{a/b}$ is the number of iterations. In the same setting, one-pass SGD only attains a test error of $\Theta(M^{1-b} + N^{(1-b)/a})$ (see, e.g., Lin et al., 2024). This suggests an improved scaling law via data reuse (i.e., choosing $L > N$) in data-constrained regimes. Numerical simulations are also provided to verify our theoretical findings.

## 1 Introduction

Empirical studies reveal that the performance of large-scale models often improves in a predictable manner as both model size (denoted by $M$) and sample size (denoted by $N$) increase (see, e.g., Hoffmann et al., 2022; Besiroglu et al., 2024). These observations, known as *neural scaling laws*, suggest that the population risk (denoted by $\mathcal{R}$) of large models decreases following a power-law formula, namely,

$$\mathcal{R}(M, N) \approx \mathcal{R}^* + c_1 M^{-a_1} + c_2 N^{-a_2},$$

where $\mathcal{R}^* > 0$ denotes the irreducible error—such as the intrinsic entropy of natural language in the case of language modeling (Kaplan et al., 2020)—and $a_1, a_2, c_1, c_2$ are positive constants. Neural scaling laws predict a path for improving the state-of-the-art models via *scaling model and data size*.

A line of recent work establishes provable scaling laws in simplified settings such as linear regression (see, e.g., Lin et al., 2024; Paquette et al., 2024, other related works will be discussed later in Section 6). Specifically, they consider an infinite-dimensional linear regression problem, where an $M$-dimensional linear model is trained by one-pass *stochastic gradient descent* (SGD) on $N$ Gaussian-sketched samples. Under power-law assumptions on the spectra of the data covariance and the prior covariance, they show power-law type scaling laws in linear regression. However, their results are limited to one-pass SGD, where each sample is used once. In particular, Lin et al. (2024) attributed the nice, power-law type scaling laws to the *implicit regularization* effect of one-pass SGD

39th Conference on Neural Information Processing Systems (NeurIPS 2025).

(see Section 1 therein). It is unclear if scaling laws apply to other training algorithms, particularly those involving multiple passes of the data. Indeed, the scaling laws that only apply to one-pass methods are not sustainable in a data-constrained regime.

There is evidence that data reuse can improve existing scaling laws developed for one-pass training. Empirically, Muennighoff et al. (2023) showed that with up to four passes, scaling laws approximately hold as if the reused data is new. From a theoretical perspective, the work by Pillaud-Vivien et al. (2018) shows that in a class of linear regression problems, the sample complexity of one-pass SGD is strictly suboptimal; and it can be made minimax optimal by considering multiple passes. However, Pillaud-Vivien et al. (2018) did not discuss the effect of model size or sketching. These results motivate the study of scaling laws for multi-pass methods.

**Contributions.** In this work, we study scaling laws induced by multi-pass SGD in the same infinite-dimensional linear regression setting considered by Lin et al. (2024); Paquette et al. (2024). Our results suggest that in certain regimes, the test error of models trained by multi-pass SGD scales strictly better with respect to the number of training samples compared to one-pass SGD.

We assume that the data covariance and the prior covariance exhibit aligned power-law spectra with exponents $a$ and $b - a$, respectively (see Assumption 1C and 1D) (Lin et al., 2024; Paquette et al., 2024). We prove that multi-pass SGD achieves an excess test error of order $\Theta(M^{1-b} + L^{(1-b)/a})$ when $a > b > 1$ and the number of SGD iterations $L \lesssim N^{a/b}$. This improves over the $\Theta(M^{1-b} + N^{(1-b)/a})$ bound for one-pass SGD (Lin et al., 2024) when $L > N$. In particular, when choosing the optimal number of iterations $L \asymp N^{a/b}$, multi-pass SGD achieves an excess test error of order $\Theta(N^{(1-b)/b})$, in contrast to $\Theta(N^{(1-b)/a})$ for one-pass SGD in the data-constrained regime where $N \ll M^b$. Our results thus suggest that, to a certain extent, reusing data can improve the test performance of linear models in data-constrained regimes.

**Notation.** Let $f(x)$ and $g(x)$ be two positive-valued functions. We write $f(x) \lesssim g(x)$ (and $f(x) = \mathcal{O}(g(x))$) if there exists some absolute constant (if not otherwise specified) $c > 0$ such that $f(x) \leqslant cg(x)$ for all $x$. Similarly, $f(x) \gtrsim g(x)$ (and $f(x) = \Omega(g(x))$) means $f(x) \geqslant cg(x)$ for some constant $c > 0$. We write $f(x) \asymp g(x)$ (and $f(x) = \Theta(g(x))$) when $f(x) \lesssim g(x) \lesssim f(x)$. We also occasionally use $\widetilde{\mathcal{O}}(\cdot), \widetilde{\Theta}(\cdot)$ to hide logarithmic factors. In this work, $\log(\cdot)$ denotes the base-2 logarithm. For two vectors $\mathbf{u}, \mathbf{v}$ in a Hilbert space, we denote their inner product by $\langle \mathbf{u}, \mathbf{v} \rangle$ or $\mathbf{u}^\top \mathbf{v}$. We denote the operator norm for matrices by $\| \cdot \|$ (or $\| \cdot \|_2$) and the $\ell_2$-norm for vectors by $\| \cdot \|_2$. For a positive semi-definite (PSD) matrix $\boldsymbol{A}$ and a vector $\boldsymbol{v}$ of compatible dimensions, we write $\|\boldsymbol{v}\|_{\boldsymbol{A}}^2 := \boldsymbol{v}^\top \boldsymbol{A} \boldsymbol{v}$. For symmetric matrices, we denote the $j$-th eigenvalue of $\boldsymbol{A}$ by $\mu_j(\boldsymbol{A})$, and the rank of $\boldsymbol{A}$ by $r(\boldsymbol{A})$.

## 2 Setup

Let $\mathbf{x} \in \mathbb{H}$ denote a feature vector in a Hilbert space $\mathbb{H}$ (finite or countably infinite-dimensional) with dimension $d := \dim(\mathbb{H})$, and $y \in \mathbb{R}$ denote its corresponding response. In linear regression, the test error (i.e., population risk) of the parameter $\mathbf{w} \in \mathbb{H}$ is measured by the mean squared error:

$$\mathcal{R}(\mathbf{w}) := \mathbb{E}_{(\mathbf{x},y) \sim P}\Big[\big(\langle \mathbf{x}, \mathbf{w} \rangle - y\big)^2\Big]$$

for some distribution $P$ on $\mathbb{H} \times \mathbb{R}$. Given samples of the form $(\mathbf{x}, y)$, instead of fitting a $d$-dimensional linear model, we train an $M$-dimensional sketched linear model with $M \ll d$. Namely, we consider linear predictors with $M$ parameters, defined as

$$f_{\mathbf{v}} : \mathbb{H} \to \mathbb{R}, \quad \mathbf{x} \mapsto \langle \mathbf{v}, \mathbf{S}\mathbf{x} \rangle, \tag{1}$$

where $\mathbf{v} \in \mathbb{R}^M$ are the trainable parameters, and $\mathbf{S} \in \mathbb{R}^{M \times d}$ is some fixed sketching matrix. In this work, we consider Gaussian sketching, where the entries of $\mathbf{S}$ are drawn independently from $\mathcal{N}(0, 1/M)$. Given a set of $N$ i.i.d. samples $(\mathbf{x}_i, y_i)_{i=1}^N$ from $P$, we train $f_{\mathbf{v}}$ via *multi-pass stochastic gradient descent* (multi-pass SGD), that is,

$$\begin{aligned}
\mathbf{v}_t &:= \mathbf{v}_{t-1} - \gamma_t \big(f_{\mathbf{v}_{t-1}}(\mathbf{x}_{i_t}) - y_{i_t}\big)\nabla_{\mathbf{v}} f_{\mathbf{v}_{t-1}}(\mathbf{x}_{i_t}) \\
&= \mathbf{v}_{t-1} - \gamma_t \mathbf{S}\mathbf{x}_{i_t}(\mathbf{x}_{i_t}^\top \mathbf{S}^\top \mathbf{v}_{t-1} - y_{i_t}), \qquad t = 1, \ldots, L,
\end{aligned} \tag{multi-pass SGD}$$

where $L$ is the number of total steps, $i_t \overset{iid}{\sim} \mathsf{unif}([N])$ for $t \in [L]$, and $(\gamma_t)_{t=1}^L$ are the stepsizes. Without loss of generality, we assume zero initialization $\mathbf{v}_0 = \mathbf{0}$. We consider a geometric decaying stepsize scheduler (Ge et al., 2019; Wu et al., 2022b; Lin et al., 2024),

$$\gamma_t := \gamma_0/2^\ell \quad \text{for } t = 1, \ldots, L, \quad \text{where } \ell = \lfloor t/(L/\log(L)) \rfloor, \tag{2}$$

and $\gamma_0 > 0$ is the initial stepsize. The output of multi-pass SGD is taken as its last iterate $\mathbf{v}_L$. We emphasize that the algorithm we consider differs slightly from the standard SGD used in practice, where the samples are shuffled at the beginning of each epoch (pass) and then processed sequentially without replacement. In contrast, we assume that at each step, a sample is drawn independently from the training dataset, allowing for repeated sampling within an epoch. Moreover, our analysis applies to other stepsize schedules (such as polynomial decay), but we focus on geometric decay since it is known to yield near minimax optimal excess test error for the last iterate of SGD in the finite-dimensional regime (Ge et al., 2019).

Conditioned on a sketching matrix $\mathbf{S}$, the risk of $\mathbf{v}_L$ is computed as

$$\mathcal{R}_M(\mathbf{v}_L) = \mathcal{R}(\mathbf{S}^\top \mathbf{v}_L) = \mathbb{E}\Big[\big(\langle \mathbf{x}, \mathbf{S}^\top \mathbf{v}_L \rangle - y\big)^2\Big],$$

where the expectation is over $(\mathbf{x}, y)$ from $P$. As an important component of our analysis, we also consider the *gradient descent* (GD) iterates

$$\begin{aligned}
\boldsymbol{\theta}_t &:= \boldsymbol{\theta}_{t-1} - \frac{\gamma_t}{N} \sum_{i=1}^N \big(f_{\boldsymbol{\theta}_{t-1}}(\mathbf{x}_i) - y_i\big) \nabla_{\mathbf{v}} f_{\boldsymbol{\theta}_{t-1}}(\mathbf{x}_i) \\
&= \boldsymbol{\theta}_{t-1} - \frac{\gamma_t}{N} \mathbf{S}\mathbf{X}^\top(\mathbf{X}\mathbf{S}^\top \boldsymbol{\theta}_{t-1} - \mathbf{y}), \qquad t = 1, \ldots, L,
\end{aligned} \tag{GD}$$

where $\mathbf{X} = (\mathbf{x}_1, \ldots, \mathbf{x}_N)^\top$, $\mathbf{y} = (y_1, \ldots, y_N)^\top$, $\boldsymbol{\theta}_0 = \mathbf{0}$, and $(\gamma_t)_{t=1}^L$ are the same stepsizes as in (2). Conditioned on the sketching matrix $\mathbf{S}$ and the dataset $(\mathbf{x}_i, y_i)_{i=1}^N$, it can be verified by induction that $\mathbf{v}_L$ is an unbiased estimate of $\boldsymbol{\theta}_L$, i.e., $\mathbb{E}[\mathbf{v}_L] = \boldsymbol{\theta}_L$, where the expectation is over the randomness of the indices $(i_t)_{t=1}^L$.

**Risk decomposition.** We can decompose the risk (i.e., the test error) achieved by $\mathbf{v}_L$, the last iterate of (multi-pass SGD), into the sum of *irreducible risk*, *approximation error*, the *excess risk* of the last iterate of (GD), and a *fluctuation error*:

$$\mathcal{R}_M(\mathbf{v}_L) = \underbrace{\min \mathcal{R}(\cdot)}_{\text{Irreducible}} + \underbrace{\min \mathcal{R}_M(\cdot) - \min \mathcal{R}(\cdot)}_{\text{Approx}} + \underbrace{\mathcal{R}_M(\boldsymbol{\theta}_L) - \min \mathcal{R}_M(\cdot)}_{\text{Excess}} + \underbrace{\mathcal{R}_M(\mathbf{v}_L) - \mathcal{R}_M(\boldsymbol{\theta}_L)}_{\text{Fluc}}. \tag{3}$$

Compared with Lin et al. (2024) (cf. Eq. 4), the decomposition in (3) includes an additional *fluctuation error* term arising from the randomness of the indices $(i_t)_{t=1}^L$ in multi-pass SGD (Zou et al., 2022). Note that the fluctuation error is non-negative by Jensen's inequality, as $\mathbf{v}_L$ is an unbiased estimate of $\boldsymbol{\theta}_L$.

## 3 Main results

In this section, we present our main result, showing that under certain power-law assumptions on the data covariance and the prior covariance, the expected risk of $\mathbf{v}_L$ from (multi-pass SGD) decays polynomially in the number of training steps $L$ and model size $M$. We begin by introducing the data assumption used throughout this work.

**Assumption 1.** *Assume the following conditions on the data distribution $P$.*

A. **Gaussian design.** *The feature vector satisfies $\mathbf{x} \sim \mathcal{N}(0, \mathbf{H})$.*

B. **Well-specified model.** *The response satisfies $\mathbb{E}[y \mid \mathbf{x}, \mathbf{w}^*] = \mathbf{x}^\top \mathbf{w}^*$. Define $\sigma^2 := \mathbb{E}[(y - \mathbf{x}^\top \mathbf{w}^*)^2]$.*

C. **Power-law spectrum.** *The eigenvalues of $\mathbf{H}$ satisfy $\lambda_i \asymp i^{-a}$ for all $i > 0$ for some $a > 1$.*

D. **Source condition.** *Let $(\lambda_i, \mathbf{v}_i)_{i>0}$ be the eigenvalues and eigenvectors of $\mathbf{H}$. Assume $\mathbf{w}^*$ follows a prior such that*

   *for $i \neq j$, $\mathbb{E}[\langle \mathbf{v}_i, \mathbf{w}^* \rangle \langle \mathbf{v}_j, \mathbf{w}^* \rangle] = 0$; and for $i > 0$, $\mathbb{E}[\lambda_i \langle \mathbf{v}_i, \mathbf{w}^* \rangle^2] \asymp i^{-b}$, for some $b > 1$.*

Assumptions 1A and 1B posit that the feature vector $\mathbf{x}$ follows a Gaussian distribution and that the linear model $y = \langle \mathbf{x}, \mathbf{w}^* \rangle + \epsilon$ is well-specified, which are standard conditions in the analysis of linear regression. Assumptions 1C and 1D assume that both the covariance of $\mathbf{x}$ and the prior on the true parameter $\mathbf{w}^*$ have power-law spectra and share the same eigenspace. In particular, the true parameter $\mathbf{w}^*$ follows an isotropic prior when $a = b$. These conditions are common in theoretical analysis of scaling laws (Bordelon et al., 2024a; Lin et al., 2024; Paquette et al., 2024), and the power-law spectrum in Assumption 1C is also empirically observed in real-world data, such as the frequency distribution of words in natural languages (Piantadosi, 2014). We further note that Assumption 1 matches the conditions of Theorem 4.2 in Lin et al. (2024), which established scaling laws for one-pass SGD under the same setup. This alignment allows a direct comparison between our results and those in Lin et al. (2024), highlighting the benefits of data reuse in certain data-constrained regimes. Finally, we define the number of effective steps $L_{\texttt{eff}} := \lfloor L / \log L \rfloor$.

**Theorem 3.1** (Error bounds for multi-pass SGD). *Suppose Assumption 1 holds. Consider an $M$-dimensional linear predictor trained by* (multi-pass SGD) *on $N$ samples. Recall the decomposition in* (3). *Assume the initial stepsize $\gamma_0 = \min\{\gamma, 1/[4 \max_i \|\mathbf{S}\mathbf{x}_i\|_2^2]\}$ for some $\gamma \lesssim 1/\log N$ and the number of effective steps $L_{\texttt{eff}} \lesssim N^a/\gamma$. Then with probability at least $1 - e^{-\Omega(M)}$ over $\mathbf{S}$*

*1. Irreducible $= \mathcal{R}(\mathbf{w}^*) = \sigma^2$.*

*2. $\mathbb{E}_{\mathbf{w}*}[\textsf{Approx}] \eqsim M^{1-b}$.*

*3. Suppose $\sigma^2 \gtrsim 1$. The expected excess risk (*Excess*) admits a decomposition into a bias term (*Bias*) and a variance term (*Var*), namely,*

$$\mathbb{E}[\textsf{Excess}] \eqsim \textsf{Bias} + \sigma^2 \textsf{Var},$$

*where the expectation is over the randomness of $\mathbf{w}^*$, $(\mathbf{x}_i, y_i)_{i=1}^N$ and $(i_t)_{t=1}^L$. Moreover, when $a > b - 1$, *Bias* and *Var* satisfy*

$$\textsf{Bias} \lesssim \max\{M^{1-b}, (L_{\texttt{eff}}\gamma)^{(1-b)/a}\},$$
$$\textsf{Bias} \gtrsim (L_{\texttt{eff}}\gamma)^{(1-b)/a} \text{ when } (L_{\texttt{eff}}\gamma)^{1/a} \leqslant M/c \text{ for some constant } c > 0,$$
$$\textsf{Var} \eqsim \min\left\{M, (L_{\texttt{eff}}\gamma)^{1/a}\right\}/N.$$

*4. Suppose $\sigma^2 \eqsim 1$ and $L_{\texttt{eff}} \lesssim N^{(1-\varepsilon)a}/\gamma$ for some $\varepsilon \in (0, 1]$. The expected fluctuation error $\mathbb{E}[\textsf{Fluc}]$ satisfies*

$$\mathbb{E}[\textsf{Fluc}] \lesssim \gamma \log N \cdot \left[(L_{\texttt{eff}}\gamma)^{1/a-1} + \frac{(L_{\texttt{eff}}\gamma)^{1/a}}{N}\right], \quad \text{and}$$
$$\mathbb{E}[\textsf{Fluc}] \gtrsim \gamma(L_{\texttt{eff}}\gamma)^{1/a-1} \quad \text{when } L_{\texttt{eff}} \lesssim N/\gamma \text{ and } (L_{\texttt{eff}}\gamma)^{1/a} \leqslant M/c \text{ for some constant } c > 0,$$

*where the expectation is over $\mathbf{w}^*$, $(\mathbf{x}_i, y_i)_{i=1}^N$ and $(i_t)_{t=1}^L$.*

*In the results, the hidden constants depend only on $(a, b)$ for part 1—3, and on $(a, b, \varepsilon)$ for part 4.*

See the proof of Theorem 3.1 in Appendix A.2.1. A few comments on Theorem 3.1 are in order.

**Comparison with Lin et al. (2024).** The results in Theorem 3.1 are more general than those in Theorem 4.1 and 4.2 of Lin et al. (2024). Specifically, we derive matching upper and lower bounds for each term in the decomposition (3) for multi-pass SGD with an arbitrary number of steps $L \lesssim N^a$, except for the lower bound on the fluctuation error, which requires $L \lesssim N$. In contrast, Lin et al. (2024) only considered one-pass SGD where $L = N$. When $a \geqslant b$ and $L = N$, our bounds match those for one-pass SGD given in Theorems 4.1 and 4.2 of Lin et al. (2024) up to logarithmic factors (see Section 3.2 for more details).

**The fluctuation error.** From part 4 of Theorem 3.1, we see that the fluctuation error term $\mathbb{E}[\textsf{Fluc}]$ vanishes as the stepsize $\gamma$ goes to zero. This is intuitive: when $\gamma$ is small, the noise from random sampling becomes negligible and multi-pass SGD closely approximates gradient descent. Moreover, when $a \geqslant b$ and $L_{\texttt{eff}} \lesssim N^{a/b}$, it can be verified that for any $\gamma \lesssim 1/\log N$, the fluctuation error is dominated by the sum of the approximation error and excess risk of (GD), i.e., $\mathbb{E}[\textsf{Fluc}] \lesssim \mathbb{E}_{\mathbf{w}*}[\textsf{Approx}] + \mathbb{E}[\textsf{Excess}]$.

**Choice of the stepsize.** The assumption $\gamma \lesssim 1/\log N$ ensures that the initial stepsize $\gamma_0 = \gamma$ with high probability. However, to guarantee the convergence of GD iterates, it suffices to have $\gamma_0 \approx \gamma \leqslant 1/\|\mathbf{SX}^\top\mathbf{XS}^\top/N\|_2 \overset{(i)}{\approx} 1.$[1] The additional $\log N$ factor and the assumption $\gamma_0 = \min\{\gamma, 1/[4\max_i \|\mathbf{Sx}_i\|_2^2]\}$ are technical conditions needed for analyzing the fluctuation error term. We leave the problem of relaxing these assumptions to future work.

**Constant-stepsize SGD with iterate averaging.** Similar to Lin et al. (2024), the results in Theorem 3.1 also hold for the average of the iterates of multi-pass SGD with a constant stepsize, with the only modification being that $L_{\texttt{eff}}$ is replaced by $L$ in the bounds. We provide simulations supporting this claim in Section 4.

**Relaxation of Assumption 1.** The Gaussian design in Assumption 1A can be relaxed to a sub-Gaussian design when establishing the upper bounds for Bias, Var, Approx in Theorem 3.1 and the upper bounds in subsequent corollaries. Moreover, the exact alignment of the eigenvectors of the prior and data covariance in Assumption 1D can be relaxed. We refer to Appendix A.3 for more details.

Next, we discuss some implications of the error bounds in Theorem 3.1.

## 3.1 Scaling laws for GD

To begin with, we present matching upper and lower bounds for the expected test error of the last iterate of (GD) (denoted by $\mathbb{E}[\mathcal{R}_M(\boldsymbol{\theta}_L)]$). We note that the GD iterates have strictly smaller test error than the corresponding multi-pass SGD iterates when $\gamma > 0$, since the GD iterates $(\boldsymbol{\theta}_t)_{t=1}^L$ are the expectation of the multi-pass SGD iterates $(\mathbf{v}_t)_{t=1}^L$, conditioned on the sketching matrix $\mathbf{S}$ and the dataset $(\mathbf{x}_i, y_i)_{i=1}^N$. By combining part 1—3 of Theorem 3.1, we have

**Corollary 3.2** (Scaling laws for GD). *Let Assumption 1 hold and $a > b - 1$. Consider an $M$-dimensional linear predictor trained by* (GD) *on $N$ samples with stepsizes $\gamma_0 = \min\{\gamma, 1/[4\operatorname{tr}(\mathbf{SX}^\top\mathbf{XS}^\top/N)]\}$ for some $\gamma \lesssim 1$. Suppose $\sigma^2 \approx 1$ and $L_{\texttt{eff}} \lesssim N^a/\gamma$. With probability at least $1 - e^{-\Omega(M)}$ over $\mathbf{S}$, the expected risk of $\boldsymbol{\theta}_L$ satisfies*

$$\mathbb{E}[\mathcal{R}_M(\boldsymbol{\theta}_L)] = \sigma^2 + \underbrace{\Theta\left(\frac{1}{M^{b-1}}\right) + \Theta\left(\frac{1}{(L_{\texttt{eff}}\gamma)^{(b-1)/a}}\right)}_{\text{Approx+Bias}} + \underbrace{\Theta\left(\frac{\min\{M, (L_{\texttt{eff}}\gamma)^{1/a}\}}{N}\right)}_{\text{Var}}.$$

*Here, $\Theta(\cdot)$ hides constants that only depend on $(a, b)$.*

See the proof of Corollary 3.2 in Appendix A.2.2. From Corollary 3.2, we see that the variance error of GD is dominated by the sum of the approximation error and the bias error (i.e. Var $\lesssim$ Approx + Bias) when $L_{\texttt{eff}}\gamma \lesssim N^{a/b}$. To achieve the optimal expected test error, we may choose $\gamma \approx 1$ and the number of effective steps $L_{\texttt{eff}} \approx \min\{N^{a/b}, M^a\}/\gamma \lesssim N^a$. Under this choice, we have

$$\mathbb{E}[\mathcal{R}_M(\boldsymbol{\theta}_L)] - \sigma^2 = \begin{cases} \Theta\left(\frac{1}{N^{(b-1)/b}}\right), & \text{if } N \lesssim M^b, \\ \Theta\left(\frac{1}{M^{b-1}}\right), & \text{if } N \gtrsim M^b. \end{cases}$$

It is worth mentioning that a decreasing stepsize schedule as in (2) is not necessary for our analysis. In fact, Corollary 3.2 remains valid for the last iterate of constant-stepsize GD (i.e., $\gamma_t \equiv \gamma$) when replacing $L_{\texttt{eff}}$ with $L$ in the bounds. In addition, the GD iterate $\boldsymbol{\theta}_L$ achieves the same expected risk (up to logarithmic factors) as one-pass SGD when $L \approx N$, where the performance of one-pass SGD is characterized in Theorem 4.2 of Lin et al. (2024).

However, the computational cost of GD is substantially larger than that of one-pass SGD, since each update requires computing gradients from all samples, resulting in a complexity of $\widetilde{\mathcal{O}}(MN^2)$ compared to $\widetilde{\mathcal{O}}(MN)$ for one-pass SGD. Nevertheless, the excess test error of GD serves as an always-valid lower bound for that of multi-pass SGD, and is also an upper bound (up to logarithmic factors) in certain regimes where the fluctuation error is dominated by the sum of the approximation error and the excess risk of GD.

---

[1]Step (i) follows from e.g., Theorem 4 and 5 in Koltchinskii and Lounici (2017).

## 3.2 Scaling laws for multi-pass SGD

We now analyze the expected test error of the last iterate of (multi-pass SGD). By Theorem 3.1 and Corollary 3.2, we have

**Corollary 3.3** (Scaling laws for multi-pass SGD when $a \geqslant b$). *Suppose the assumptions in Theorem 3.1 are in force, $\sigma^2 \asymp 1$, and $a \geqslant b > 1$. For any $L_{\mathrm{eff}} \lesssim N^{a/b}/\gamma$, we have*

$$\mathbb{E}[\mathcal{R}_M(\mathbf{v}_L)] = \sigma^2 + \Theta\left(\frac{1}{M^{b-1}}\right) + \Theta\left(\frac{1}{(L_{\mathrm{eff}}\gamma)^{(b-1)/a}}\right)$$

*with probability at least $1 - e^{-\Omega(M)}$. Here, all hidden constants depend only on $(a, b)$.*

In contrast, Theorem 4.2 in Lin et al. (2024) proved that one-pass SGD with $\mathbf{v}_0^o = \mathbf{0}$, $\mathbf{v}_t^o = \mathbf{v}_{t-1}^o - \gamma_t \mathbf{S}\mathbf{x}_t(\mathbf{x}_t^\top \mathbf{S}^\top \mathbf{v}_{t-1}^o - y_t)$ for $t \in [N]$ satisfies

$$\mathbb{E}[\mathcal{R}_M(\mathbf{v}_N^o)] = \sigma^2 + \Theta\left(\frac{1}{M^{b-1}}\right) + \Theta\left(\frac{1}{(N_{\mathrm{eff}}\gamma)^{(b-1)/a}}\right)$$

with probability at least $1 - e^{-\Omega(M)}$, where $N_{\mathrm{eff}} := N/\log N$.

Several remarks on Corollary 3.3 are listed below.

**Benefits of data reuse.** When $a \geqslant b > 1$, Corollary 3.3 shows that multi-pass SGD achieves an excess test error of order $\Theta(M^{1-b} + (L_{\mathrm{eff}}\gamma)^{(1-b)/a})$ when the number of effective SGD steps $L_{\mathrm{eff}} \lesssim N^{a/b}$, while one-pass SGD achieves an excess test error of order $\Theta(M^{1-b} + (N_{\mathrm{eff}}\gamma)^{(1-b)/a})$. Therefore, the reused data across multiple passes (epochs) can be viewed as fresh data when the number of passes is smaller than $N^{a/b-1}$. For example, when $L = kN$ for some constant $k > 1$, the test error achieved by $k$-pass SGD matches that of one-pass SGD trained on $kN$ *i.i.d.* samples despite the training data being reused—aligning with the empirical observations in Muennighoff et al. (2023).

Moreover, when the number of effective steps is chosen[2] as $L_{\mathrm{eff}} \asymp \min\{N^{a/b}, M^a\}/\gamma$ and the learning rate $\gamma \asymp 1/\log N$, the excess test error of multi-pass SGD satisfies

$$\mathbb{E}[\mathcal{R}_M(\mathbf{v}_L)] - \sigma^2 \asymp M^{1-b} + N^{(1-b)/b},$$

while choosing $\gamma \asymp 1$ for one-pass SGD yields

$$\mathbb{E}[\mathcal{R}_M(\mathbf{v}_N^o)] - \sigma^2 \asymp M^{1-b} + N_{\mathrm{eff}}^{(1-b)/a}.$$

Therefore, in the data-constrained regime where $N \ll M^b$, reusing data and running multi-pass SGD for $N^{a/b-1}$ epochs yields an improved rate of $\widetilde{\mathcal{O}}(N^{(1-b)/b})$ compared to the one-pass SGD rate of $\widetilde{\mathcal{O}}(N^{(1-b)/a})$ when $a > b$.

**Optimal compute allocation.** Given a total compute budget $C = L \cdot M$, by Corollary 3.3, we can set $L = \mathcal{O}(C^{a/(a+1)})$ and $M = \mathcal{O}(C^{1/(a+1)})$ with stepsize $\gamma \asymp 1/\log L$ to achieve the optimal rate $\widetilde{\mathcal{O}}(C^{(1-b)/(a+1)})$ for the excess test error $\mathbb{E}[\mathcal{R}_M(\mathbf{v}_L)] - \sigma^2$. This matches the optimal rate for one-pass SGD (Lin et al., 2024) given the same compute budget, but requires only $N = \mathcal{O}(C^{b/(a+1)})$ number of i.i.d. samples in contrast to $N = \mathcal{O}(C^{a/(a+1)})$ for one-pass SGD.

**Minimax optimal rate.** When $a > b > 2$ and $M \gg N^{1/b}$, the improved rate $\widetilde{\mathcal{O}}(N^{(1-b)/b})$ achieved by multi-pass SGD matches the minimax optimal rate for a class of linear regression problems with similar spectral conditions (Pillaud-Vivien et al., 2018), up to sub-polynomial factors.

When $a < b < a + 1$, similarly, we have the following corollary from Theorem 3.1.

**Corollary 3.4** (Scaling laws for multi-pass SGD when $a < b < a + 1$). *Suppose the assumptions in Theorem 3.1 are in force and $\sigma^2 \asymp 1$. When $a < b < a + 1$, for any $L_{\mathrm{eff}} \lesssim N/\gamma$, we have*

$$\mathbb{E}[\mathcal{R}_M(\mathbf{v}_L)] = \sigma^2 + \Theta\left(\frac{1}{\min\{M, (L_{\mathrm{eff}}\gamma)^{1/a}\}^{b-1}}\right) + \Theta\left(\frac{\min\{M, (L_{\mathrm{eff}}\gamma)^{1/a}\}}{N}\right) + \mathbb{E}[\mathsf{Fluc}]$$

---

[2]Note that this choice of $L_{\mathrm{eff}}$ (and therefore $L$) is optimal as it minimizes $\mathbb{E}[\mathcal{R}_M(\mathbf{v}_L)] - \sigma^2$ up to logarithmic factors for $L_{\mathrm{eff}} \lesssim N^a$.

with probability at least $1 - e^{-\Omega(M)}$, where the fluctuation error satisfies $\mathbb{E}[\mathsf{Fluc}] \lesssim \gamma \log N \cdot \left(L_{\mathsf{eff}}\gamma\right)^{1/a-1}$, and $\mathbb{E}[\mathsf{Fluc}] \gtrsim \gamma\left(L_{\mathsf{eff}}\gamma\right)^{1/a-1}$ when $(L_{\mathsf{eff}}\gamma)^{1/a} \lesssim M$.

Therefore, in the data-constrained regime where $N \ll M^b$, we have $\mathbb{E}[\mathcal{R}_M(\mathbf{v}_L)] - \sigma^2 = \widetilde{\Theta}((L_{\mathsf{eff}}\gamma)^{(1-b)/a} + \gamma(L_{\mathsf{eff}}\gamma)^{1/a-1})$. Choosing $L_{\mathsf{eff}} \asymp N$ and the optimal learning rate $\gamma \asymp L_{\mathsf{eff}}{}^{a/b-1}$ that balances the excess test error of GD and the fluctuation error, we obtain a rate of $\widetilde{\Theta}(N^{(1-b)/b})$. This matches the bound for one-pass SGD in Lin et al. (2024) (up to logarithmic factors) when $a < b < a + 1$.

# 4 Experiments

We also perform simulations to validate our theoretical findings. Namely, we train $M$-dimensional sketched linear predictors (1) via one-pass SGD and multi-pass SGD following the setup in Section 2 and 3, and analyze how their excess test errors scale with the number of samples $N$ and the model size $M$. In each simulation, we generate $N$ i.i.d. samples $(\mathbf{x}_i, y_i)_{i=1}^N$ from a linear model $y_i = \langle \mathbf{x}_i, \mathbf{w}^* \rangle + \epsilon_i$, where $\mathbf{w}^* \in \mathbb{R}^d$ is an unknown parameter vector and $\epsilon_i \sim \mathcal{N}(0, \sigma^2)$ are i.i.d. Gaussian noise. The covariates $\mathbf{x}_i$ are drawn from $\mathcal{N}(0, \mathbf{H})$, and the true parameter vector $\mathbf{w}^*$ is sampled from a Gaussian prior $\mathcal{N}(0, \mathbf{H}^\mathbf{w})$, where $\mathbf{H} := \mathrm{diag}\{1, 2^{-a}, \ldots, d^{-a}\}/\sum_{i=1}^d i^{-a}$ and $\mathbf{H}^\mathbf{w} := \mathrm{diag}\{1, 2^{a-b}, \ldots, d^{a-b}\}$ for some $a, b > 1$. We set the dimension $d$ to be sufficiently large relative to $M$ so that Assumption 1C and 1D are approximately satisfied. For simplicity, we implement multi-pass SGD by reusing samples sequentially without replacement in each epoch, rather than sampling i.i.d. from the empirical distribution. In all experiments, we set $d = 10000, \sigma^2 = 1$ and $(a, b) = (2, 1.5)$.

Figure 1(a) compares the excess test error of one-pass SGD and multi-pass SGD with the number of steps $L \asymp N^{a/b-1}$. We observe that multi-pass SGD achieves better scaling in the sample size $N$ compared to one-pass SGD when $N$ is relatively small (i.e., $N \ll M^b$). Moreover, the fitted exponents are close to the theoretical predictions in Corollary 3.3 (i.e., $\frac{1-b}{a} = -0.25$ and $\frac{1-b}{b} = -0.33$). Similar results hold for the average of the iterates of constant-stepsize SGD, as shown in Figure 1(b). On the other hand, when $N \gg M^b$, Figure 1(c) shows that one-pass SGD and multi-pass SGD achieve the same scaling in the model size $M$ with the exponent $k \approx 1 - b$, consistent with Corollary 3.3. In addition, Figure 1(d) illustrates that multi-pass SGD achieves the same excess test error as one-pass SGD on fresh data when the number of passes is below a certain threshold. Overall, the empirical observations align closely with our theoretical predictions.

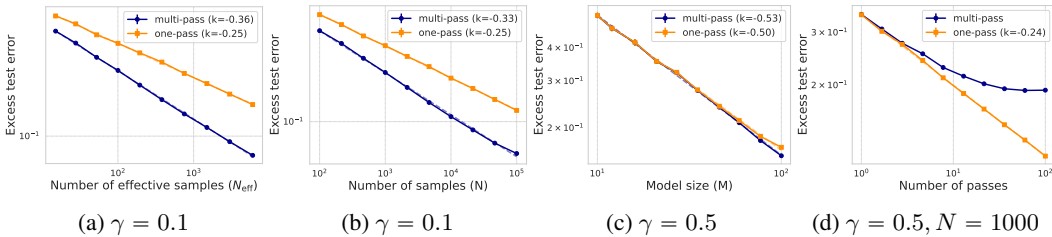

(a) $\gamma = 0.1$     (b) $\gamma = 0.1$     (c) $\gamma = 0.5$     (d) $\gamma = 0.5, N = 1000$

Figure 1: Multi-pass SGD versus one-pass SGD. In (a)—(c), multi-pass SGD is ran for $L \asymp N^{a/b}$ steps. (a), (b), (d): SGD with geometrically decaying stepsizes; (c): SGD with constant stepsizes. We use linear functions to fit the excess test error in log-log scale. The fitted exponents ($k$) are close to the theoretical predictions in Corollary 3.3. The errorbars denote the $\pm 1$ standard deviation of the expected excess test error over 100 i.i.d. samples of $(\mathbf{S}, \mathbf{w}^*)$. Parameters: $\sigma^2 = 1, d = 10000$, $(a, b) = (2, 1.5)$. (a), (b), (d): $M = 1000$; (c): $N = 10^5$.

# 5 Proof Overview

We provide an overview of the proof of Theorem 3.1 in this section. A full proof can be found in Appendix A.2.1. Let $\mathbf{v}^* = (\mathbf{SHS}^\top)^{-1}\mathbf{SHw}^*$ and adopt the shorthand notations

$$\boldsymbol{\Sigma} := \mathbf{SHS}^\top, \quad \widehat{\boldsymbol{\Sigma}} := \frac{\mathbf{SX}^\top\mathbf{XS}^\top}{N}.$$

It can be verified by some basic algebra that

$$\mathbb{E}\mathcal{R}_M(\mathbf{v}_L) = \mathbb{E}\Big[\big(\langle\mathbf{x}, \mathbf{S}^\top\mathbf{v}_L\rangle - y\big)^2\Big] = \underbrace{\sigma^2}_{\text{Irreducible}} + \underbrace{\mathbb{E}\|\mathbf{S}^\top\mathbf{v}^* - \mathbf{w}^*\|_{\mathbf{H}}^2}_{\text{Approx}} + \underbrace{\mathbb{E}\|\boldsymbol{\theta}_L - \mathbf{v}^*\|_{\boldsymbol{\Sigma}}^2}_{\text{Excess}} + \underbrace{\mathbb{E}\|\mathbf{v}_L - \boldsymbol{\theta}_L\|_{\boldsymbol{\Sigma}}^2}_{\text{Fluc}},$$

where the expectations on the R.H.S. are over $\mathbf{w}^*, (\mathbf{x}_i, y_i)_{i=1}^N$ and $(i_t)_{t=1}^L$, and we recall $\mathbf{v}_L$ in (multi-pass SGD) and $\boldsymbol{\theta}_L$ in (GD). From the above decomposition, we immediately have

1. Irreducible $= \mathcal{R}(\mathbf{w}^*) = \sigma^2$.
2. $\mathbb{E}_{\mathbf{w}^*}[\text{Approx}] = \mathbb{E}_{\mathbf{w}^*}\|\mathbf{S}^\top\mathbf{v}^* - \mathbf{w}^*\|_{\mathbf{H}}^2 \asymp M^{1-b}$ with probability at least $1 - e^{-\Omega(M)}$ over $\mathbf{S}$ by Lemma C.5 in Lin et al. (2024) (see also Lemma E.7).

The excess risk of (GD) can be further decomposed into the sum of bias and variance, namely,

$$\mathbb{E}[\text{Excess}] = \text{Bias} + \sigma^2\text{Var},$$

where

$$\text{Bias} := \mathbb{E}\Big\|\prod_{t=1}^L\big(\mathbf{I} - \gamma_t\widehat{\boldsymbol{\Sigma}}\big)\mathbf{v}^*\Big\|_{\boldsymbol{\Sigma}}^2, \quad \text{Var} := \mathbb{E}[\text{tr}(\mathbf{XS}^\top\mathbf{V}(\widehat{\boldsymbol{\Sigma}})\boldsymbol{\Sigma}\mathbf{V}(\widehat{\boldsymbol{\Sigma}})\mathbf{SX}^\top)]$$

with $\mathbf{V}(\widehat{\boldsymbol{\Sigma}}) := \frac{1}{N}[\mathbf{I} - \prod_{t=1}^L(\mathbf{I} - \gamma_t\widehat{\boldsymbol{\Sigma}})]/\widehat{\boldsymbol{\Sigma}}$. The bounds on the bias (Bias) and variance (Var) then follow immediately from Lemma B.3 and C.2, respectively. Lastly, the bounds on the fluctuation error follow from Lemma D.5.

The main technical difficulty of proving Theorem 3.1 lies in bounding the bias, variance, and fluctuation error terms. For bias and variance upper bounds, due to the non-commutativity of the population covariance $\boldsymbol{\Sigma}$ and the empirical covariance $\widehat{\boldsymbol{\Sigma}}$, we apply a covariance replacement trick (Lemma E.1; see also Lemma 7 in Pillaud-Vivien et al. (2018)) to replace the population covariance with the empirical covariance in the expressions of bias and variance, as well as concentration properties of sub-Gaussian covariance to simplify their expressions. For the lower bounds, we show that a specific function of the empirical covariance commutes with the population covariance in expectation, and apply Von Neumann's trace inequality.

For the fluctuation error, we follow the standard practice as in Pillaud-Vivien et al. (2018) and Aguech et al. (2000) to express the difference between the multi-pass SGD and GD trajectories, $\mathbf{v}_t - \boldsymbol{\theta}_t$ as a stochastic process (Eq. (18)). We then bound the fluctuation error $\mathbb{E}[\|\boldsymbol{\Sigma}^{1/2}(\mathbf{v}_L - \boldsymbol{\theta}_L)\|^2]$ through controlling the accumulated error of the stochastic process using Lemma D.2 and D.3, which involves a novel leave-one-out argument to control the model parameters. Although several upper bounds on the fluctuation error have been established for infinite-dimensional linear models (Pillaud-Vivien et al., 2018; Zou et al., 2022), the interaction between the sketching matrix $\mathbf{S}$ and the samples $(\mathbf{x}_i, y_i)_{i=1}^N$ in our setup introduces additional technical challenges (Lin et al., 2024). Moreover, we derive a novel lower bound on the fluctuation error that matches the upper bound up to a logarithmic factor in certain regimes by carefully controlling the accumulated variance from random sampling (i.e. the accumulated variance induced by the random indices $(i_t)_{t=1}^L$).

# 6 Related Works

**Empirical scaling laws.** Scaling laws have been extensively studied in recent years as a way to understand and predict how model performance improves with increasing model size and data size (Hestness et al., 2017; Rosenfeld et al., 2019; Kaplan et al., 2020; Henighan et al., 2020; Hoffmann et al., 2022; Zhai et al., 2022; Muennighoff et al., 2023). The seminal work by Kaplan et al. (2020) introduced the concept of *neural scaling laws*, demonstrating empirically that the test

error of large transformer models decreases predictably following a power law with respect to the model size and data size. Subsequent works refined and extended these observations by proposing more accurate scaling formulas (Henighan et al., 2020; Hoffmann et al., 2022; Alabdulmohsin et al., 2022; Caballero et al., 2022; Muennighoff et al., 2023) and extending them to other settings (Kumar et al., 2024; Busbridge et al., 2025). In particular, Hoffmann et al. (2022) proposed the *Chinchilla scaling law*, which advocates scaling the model and data size proportionally as compute budget increases. Muennighoff et al. (2023) investigated the effect of data reuse and multiple training epochs, introducing an empirically refined scaling formula that accounts for the number of training epochs. They demonstrated that reused data can be approximately viewed as fresh data when the number of epochs is small.

**Theoretical studies of scaling laws.** Although scaling laws have been observed across diverse settings, their theoretical understanding remains relatively limited. A number of recent works have attempted to formalize and explain the observed scaling behaviors in simplified settings (Sharma and Kaplan, 2020; Bahri et al., 2021; Maloney et al., 2022; Hutter, 2021; Michaud et al., 2024; Bordelon et al., 2024a; Atanasov et al., 2024; Dohmatob et al., 2024; Paquette et al., 2024; Lin et al., 2024; Bordelon et al., 2024b; Ren et al., 2025). For example, Bahri et al. (2021) considered a linear teacher-student model with a power-law spectrum and showed that the test error of the ordinary least squares estimator scales following a power law in $N$ (or $M$) when the other parameter goes to infinity. Bordelon et al. (2024a) analyzed the test error of the solution found by gradient flow in a linear random feature model and established power-law scaling in one of $N, M$ and $T$ (training time) while the other two parameters go to infinity. The results in these works are derived based on statistical physics heuristics and characterize scaling in only one variable in the asymptotic regime. More recently, Lin et al. (2024) analyzed the test error of the last iterate of one-pass SGD in a sketched linear model and showed that the test error scales as $\Theta(\sigma^2 + M^{1-b} + N^{(1-b)/a})$ under the source condition (Assumption 1). This is the first work to establish a finite-sample joint scaling law (in $M$ and $N$) for linear models that aligns with empirical observations (Kaplan et al., 2020; Hoffmann et al., 2022). Similarly, Ren et al. (2025) analyzed the complexity of one-pass SGD for learning two-layer neural networks in a teacher-student setup, and derived joint scaling laws for the test error under power-law assumptions on the teacher network. While previous works study the scaling behavior of the one-pass (online) SGD solutions, our work complements them by analyzing the effect of data reuse (i.e., multi-pass SGD) in data-constrained regimes.

**Risk bounds for SGD.** The generalization behavior of stochastic gradient descent (SGD), particularly in linear regression, has been extensively studied across both classical and high-dimensional regimes (Polyak and Juditsky, 1992; Défossez and Bach, 2015; Dieuleveut et al., 2017; Jain et al., 2018, 2017; Pillaud-Vivien et al., 2018; Ge et al., 2019; Dieuleveut and Bach, 2015; Berthier et al., 2020; Zou et al., 2023, 2021, 2022; Wu et al., 2022b,c; Varre et al., 2021). For one-pass SGD, several works have developed tight test error bounds in overparameterized linear models (Zou et al., 2023; Wu et al., 2022a,c). For multi-pass SGD, early works (Lin and Rosasco, 2017; Pillaud-Vivien et al., 2018; Mücke et al., 2019; Zou et al., 2022) have established test error bounds for the average of its iterates in linear regression. Compared with prior works, our main technical contribution is to precisely control the effect of random sketching and to refine the characterization of fluctuation error (see Fluc in Eq. 3) in the multi-pass setting. Under comparable regimes where the approximation error is zero, our test error bounds match those derived in Pillaud-Vivien et al. (2018), which are minimax optimal for a specific class of linear regression problems in certain cases.

## 7 Conclusion

In this work, we provide a theoretical analysis of multi-pass stochastic gradient descent (multi-pass SGD) in a sketched linear regression problem and establish refined scaling laws that characterize how the test error scales with the model size $M$, sample size $N$, and number of optimization steps $L$. Our results show that, under suitable power-law conditions on the true parameter and data distribution, data reuse via multi-pass SGD can improve model performance when the number of samples is limited. This offers a theoretical explanation for the empirical benefits of multiple passes in modern large-scale training.

Our analysis has several limitations. One limitation is the assumption that the eigenvectors of the prior and data covariance are aligned (implied by Assumption 1D). While this assumption cannot be

fully removed without affecting the error rate, it would be interesting to investigate what alternative rates are achieved when the eigenvectors are not aligned. Another limitation is that our lower bound results require Gaussian design of the covariates (i.e., Assumption 1A); a next step is to extend them to non-Gaussian design.

Beyond the limitations, many other directions remain open for future research. First, our analysis focuses on multi-pass SGD with batch size one; it would be worthwhile to understand how the test error scales with the batch size and to develop corresponding batch size scaling laws (see Jain et al., 2017). Another important direction is to study how data reuse interacts with other optimization algorithms, such as SGD with momentum or $\ell_2$-regularization and Adam. In addition, it is valuable to extend our analysis to non-linear settings and classification problems, such as logistic regression, kernel methods, and neural networks. Notably, modern large language model pretraining is based on minimizing the cross-entropy loss for next-word prediction. Understanding the scaling behavior in logistic regression—the simplest classification model—thus represents an important step toward unraveling the mysteries of LLM scaling.

## Acknowledgements

We gratefully acknowledge the NSF's support of FODSI through grant DMS-2023505 and of the NSF and the Simons Foundation for the Collaboration on the Theoretical Foundations of Deep Learning through awards DMS-2031883 and #814639 and of the ONR through MURI award N000142112431.

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

# Appendix

## Table of Contents

## A   Preliminary

### A.1   Comments and additional notations

**Comments on Assumption 1D.**   Throughout the appendix (except for Appendix A.3), we assume without loss of generality that the covariance matrix $\mathbf{H}$ is diagonal, with diagonal entries given by the eigenvalues $(\lambda_i)_{i \geqslant 1}$ in non-increasing order. This reduction is justified by the rotational invariance of the Gaussian sketching matrix $\mathbf{S}$. Under this diagonalization, Assumption 1D can be restated more explicitly as follows:

**Assumption 2** (Source condition). *Suppose $\mathbf{H} = (h_{ij})_{i,j \geqslant 1}$ is a diagonal matrix with non-increasing diagonal entries. Assume that the true parameter $\mathbf{w}^*$ satisfies:*

$$\text{for all } i \neq j, \quad \mathbb{E}[\mathbf{w}_i^* \mathbf{w}_j^*] = 0; \qquad \text{and for all } i > 0, \quad \mathbb{E}[\lambda_i \mathbf{w}_i^{*2}] \asymp i^{-b}, \quad \text{for some } b > 1.$$

Given that $\mathbf{H}$ is diagonal, we adopt the following notation. For integers $0 \leqslant k^* \leqslant k^\dagger$ (allowing $k^\dagger = \infty$), define

$$\mathbf{H}_{k*:k^\dagger} := \mathrm{diag}\{\lambda_{k*+1}, \ldots, \lambda_{k^\dagger}\} \in \mathbb{R}^{(k^\dagger - k^*) \times (k^\dagger - k^*)}.$$

For example,

$$\mathbf{H}_{0:k} = \mathrm{diag}\{\lambda_1, \ldots, \lambda_k\}, \quad \mathbf{H}_{k:\infty} = \mathrm{diag}\{\lambda_{k+1}, \lambda_{k+2}, \ldots\}.$$

Similarly, for any vector $\mathbf{w} \in \mathbb{H}$, define

$$\mathbf{w}_{k*:k^\dagger} := \left(\mathbf{w}_{k*+1}, \ldots, \mathbf{w}_{k^\dagger}\right)^\top \in \mathbb{R}^{k^\dagger - k^*}.$$

In addition, we define $\mathbf{S}_{k*:k^\dagger}$ to be the submatrix of the sketching matrix $\mathbf{S}$ consisting of the $k^* + 1$-th through $k^\dagger$-th columns.

### A.1.1 Assumptions on the stepsize

In the proofs of the general upper and lower bounds on the bias, variance, and fluctuation error, we will require that the stepsize $\gamma_0, \gamma$ satisfy certain conditions, which are summarized in the following assumption.

**Assumption 3** (Stepsize conditions). *Under the notations in Theorem 3.1 and its proof, with probability at least $1 - \exp(-\Omega(M))$ over the randomness of $\mathbf{S}$, we have*

1. $\gamma \leqslant \min\{c/\log N, c/[\mathrm{tr}(\boldsymbol{\Sigma})]\}$;

2. $\mathrm{tr}(\boldsymbol{\Sigma}^2) \lesssim 1$;

3. $\sum_{i=1}^{M} \frac{\mu_i(\boldsymbol{\Sigma})}{\mu_i(\boldsymbol{\Sigma}) + 1/(L_{\mathrm{eff}}\gamma)} \leqslant N/4$;

4. *the initial stepsize $\gamma_0 = \min\{1/[4\max_i \|\mathbf{S}\mathbf{x}_i\|_2^2], \gamma\}$ satisfies $\mathbb{P}(\gamma_0 < \gamma/t) \leqslant N^{-ct}$ for all $t \geqslant 1$.*

We will show that Assumption 3 holds when the conditions in Theorem 3.1 are satisfied.

## A.2 Proof of Theorem 3.1 and the corollaries

### A.2.1 Proof of Theorem 3.1

*Proof of Theorem 3.1.* Let $\mathbf{v}^* = (\mathbf{SHS}^\top)^{-1}\mathbf{SHw}^*$ and adopt the shorthand

$$\boldsymbol{\Sigma} := \mathbf{SHS}^\top, \quad \widehat{\boldsymbol{\Sigma}} := \frac{\mathbf{SX}^\top\mathbf{XS}^\top}{N}.$$

Also, let $\mathcal{D} := (\mathbf{x}_i, y_i)_{i=1}^N$ denote the set of training samples. Then we have the decomposition

$$\mathbb{E}\mathcal{R}_M(\mathbf{v}_L) = \mathbb{E}\Big[\big(\langle \mathbf{x}, \mathbf{S}^\top\mathbf{v}_L\rangle - y\big)^2\Big] = \mathbb{E}\Big[\big(\langle \mathbf{x}, \mathbf{S}^\top\mathbf{v}_L - \mathbf{w}^*\rangle - \epsilon\big)^2\Big] = \sigma^2 + \mathbb{E}\big[\langle \mathbf{x}, \mathbf{S}^\top\mathbf{v}_L - \mathbf{w}^*\rangle^2\big]$$

$$= \sigma^2 + \mathbb{E}\big[\langle \mathbf{x}, \mathbf{S}^\top(\mathbf{v}_L - \mathbf{v}^*) + \mathbf{S}^\top\mathbf{v}^* - \mathbf{w}^*\rangle^2\big]$$

$$\overset{(i)}{=} \sigma^2 + \mathbb{E}\big[\langle \mathbf{x}, \mathbf{S}^\top\mathbf{v}^* - \mathbf{w}^*\rangle^2\big] + \mathbb{E}\big[\langle \mathbf{x}, \mathbf{S}^\top(\mathbf{v}_L - \mathbf{v}^*)\rangle^2\big]$$

$$\overset{(ii)}{=} \sigma^2 + \mathbb{E}\big[\langle \mathbf{x}, \mathbf{S}^\top\mathbf{v}^* - \mathbf{w}^*\rangle^2\big] + \mathbb{E}\big[\langle \mathbf{x}, \mathbf{S}^\top(\boldsymbol{\theta}_L - \mathbf{v}^*)\rangle^2\big] + \mathbb{E}\big[\langle \mathbf{x}, \mathbf{S}^\top(\mathbf{v}_L - \boldsymbol{\theta}_L)\rangle^2\big]$$

$$= \underbrace{\sigma^2}_{\text{Irreducible}} + \mathbb{E}\underbrace{\|\mathbf{S}^\top\mathbf{v}^* - \mathbf{w}^*\|_{\mathbf{H}}^2}_{\text{Approx}} + \mathbb{E}\underbrace{\|\boldsymbol{\theta}_L - \mathbf{v}^*\|_{\boldsymbol{\Sigma}}^2}_{\text{Excess}} + \mathbb{E}\underbrace{\|\mathbf{v}_L - \boldsymbol{\theta}_L\|_{\boldsymbol{\Sigma}}^2}_{\text{Fluc}},$$

where step (i) uses the fact that $\mathbb{E}[\mathbf{Sxx}^\top(\mathbf{S}^\top\mathbf{v}^* - \mathbf{w}^*)] = \mathbb{E}[\mathbf{SHS}^\top\mathbf{v}^* - \mathbf{SHw}^*] = 0$, and step (ii) uses the fact that $\mathbb{E}[\mathbf{v}_L|\mathbf{S}, \mathbf{w}^*, \mathcal{D}] = \boldsymbol{\theta}_L$.

**Irreducible error.** From the above decomposition, we have $\mathsf{Irreducible} = \mathcal{R}(\mathbf{w}^*) = \sigma^2$.

**Approximation error.** We have from Lemma C.5 in Lin et al. (2024) that $\mathbb{E}_{\mathbf{w}^*}\mathsf{Approx} = \mathbb{E}_{\mathbf{w}^*}\|\mathbf{S}^\top\mathbf{v}^* - \mathbf{w}^*\|_{\mathbf{H}}^2 \asymp M^{1-b}$ with probability at least $1 - e^{-\Omega(M)}$ over $\mathbf{S}$.

**Excess risk of** (GD). Let $\tilde{\epsilon}_i = y_i - \mathbf{x}_i^\top\mathbf{S}^\top\mathbf{v}^*$ for $i \in [N]$ and write $\tilde{\boldsymbol{\epsilon}} = (\tilde{\epsilon}_1, \ldots, \tilde{\epsilon}_N)^\top$. It can be verified that, conditioned on $(\mathbf{S}, \mathbf{w}^*)$, $\mathbb{E}[\tilde{\epsilon}_i] = 0$ and $\tilde{\epsilon}_i$ is independent of $\mathbf{S}\mathbf{x}_i$. Moreover,

$$\sigma^2 \leqslant \tilde{\sigma}^2 := \mathbb{E}[\tilde{\epsilon}_i^2] = \sigma^2 + \mathbb{E}_{\mathbf{w}^*}\|\mathbf{w}^* - \mathbf{S}^\top\mathbf{v}^*\|_{\mathbf{H}}^2$$

$$= \sigma^2 + \mathbb{E}_{\mathbf{w}^*}[\mathbf{w}^{*\top}\mathbf{H}^{1/2}(\mathbf{I} - \mathbf{H}^{1/2}\mathbf{S}^\top(\mathbf{SHS}^\top)^{-1}\mathbf{SH}^{1/2})\mathbf{H}^{1/2}\mathbf{w}^*]$$

$$\leqslant \sigma^2 + \mathbb{E}_{\mathbf{w}^*}\|\mathbf{w}^*\|_{\mathbf{H}}^2 \lesssim \sigma^2.$$

Note that by definition of (GD), we have

$$\boldsymbol{\theta}_t - \mathbf{v}^* = \boldsymbol{\theta}_t - \mathbf{v}^* - \frac{\gamma_t}{N}\mathbf{SX}^\top(\mathbf{XS}^\top\boldsymbol{\theta}_{t-1} - \mathbf{y}) = \boldsymbol{\theta}_{t-1} - \mathbf{v}^* - \frac{\gamma_t}{N}\mathbf{SX}^\top(\mathbf{XS}^\top(\boldsymbol{\theta}_{t-1} - \mathbf{v}^*) - \tilde{\boldsymbol{\epsilon}})$$

$$= \big(\mathbf{I} - \gamma_t\widehat{\boldsymbol{\Sigma}}\big)(\boldsymbol{\theta}_{t-1} - \mathbf{v}^*) + \frac{\gamma_t}{N} \cdot \mathbf{SX}^\top\tilde{\boldsymbol{\epsilon}},$$

and therefore

$$\boldsymbol{\theta}_L - \mathbf{v}^* = \prod_{t=1}^{L}\left(\mathbf{I} - \gamma_t\widehat{\boldsymbol{\Sigma}}\right)(\boldsymbol{\theta}_0 - \mathbf{v}^*) + \mathbf{V}(\widehat{\boldsymbol{\Sigma}})\mathbf{S}\mathbf{X}^\top\tilde{\boldsymbol{\epsilon}}, \tag{4}$$

where

$$\mathbf{V}(\widehat{\boldsymbol{\Sigma}}) := \frac{1}{N}\sum_{t=1}^{L}\gamma_t \cdot \prod_{i=t+1}^{L}(\mathbf{I} - \gamma_i\widehat{\boldsymbol{\Sigma}}) = \frac{\mathbf{I} - \prod_{t=1}^{L}(\mathbf{I} - \gamma_t\widehat{\boldsymbol{\Sigma}})}{N\widehat{\boldsymbol{\Sigma}}}.$$

As a result, the excess risk of (GD) satifies

$$\mathbb{E}[\mathsf{Excess}] = \mathbb{E}\|\boldsymbol{\theta}_L - \mathbf{v}^*\|_{\boldsymbol{\Sigma}}^2 \overset{(iii)}{=} \mathbb{E}\left\|\prod_{t=1}^{L}\left(\mathbf{I} - \gamma_t\widehat{\boldsymbol{\Sigma}}\right)\mathbf{v}^*\right\|_{\boldsymbol{\Sigma}}^2 + \mathbb{E}\|\mathbf{V}(\widehat{\boldsymbol{\Sigma}})\mathbf{S}\mathbf{X}^\top\tilde{\boldsymbol{\epsilon}}\|_{\boldsymbol{\Sigma}}^2$$

$$\eqsim \mathsf{Bias} + \sigma^2\mathsf{Var},$$

where $\mathsf{Bias} := \mathbb{E}_{\mathbf{w}*}[\mathsf{Bias}(\mathbf{w}^*)]$ and

$$\mathsf{Bias}(\mathbf{w}^*) := \mathbb{E}_{\mathbf{X}}\left\|\prod_{t=1}^{L}\left(\mathbf{I} - \gamma_t\widehat{\boldsymbol{\Sigma}}\right)\mathbf{v}^*\right\|_{\boldsymbol{\Sigma}}^2, \qquad \mathsf{Var} := \mathbb{E}[\mathrm{tr}(\mathbf{X}\mathbf{S}^\top\mathbf{V}(\widehat{\boldsymbol{\Sigma}})\boldsymbol{\Sigma}\mathbf{V}(\widehat{\boldsymbol{\Sigma}})\mathbf{S}\mathbf{X}^\top)],$$

and step (iii) follows from the fact that $\mathbf{S}\mathbf{X}^\top$ is independent of $\tilde{\boldsymbol{\epsilon}}$ conditioned on $\mathbf{S}$. The bounds on the bias and variance follow immediately from Lemma B.3 and C.2.

**Fluctuation error.** It follows from Lemma D.5 and the assumption $\gamma \leqslant c/\log N$ that

$$\mathbb{E}[\mathsf{Fluc}] \lesssim \gamma\log N \cdot [(L_{\mathsf{eff}}\gamma)^{1/a-1} + \frac{(L_{\mathsf{eff}}\gamma)^{1/a}}{N}]$$

with probability at least $1 - e^{-\Omega(M)}$ over the randomness of $\mathbf{S}$. The lower bound on $\mathbb{E}[\mathsf{Fluc}]$ also follows from Lemma D.5.

$\square$

### A.2.2 Proof of Corollary 3.2

The proof follows immediately by combining parts 1–3 of Theorem 3.1, although we make a different assumption on the initial stepsize $\gamma_0$. In Theorem 3.1, we assume $\gamma_0 = \min\{\gamma, 1/[4\max_i\|\mathbf{S}\mathbf{x}_i\|_2^2]\}$ for some $\gamma \lesssim 1/\log N$, while in Corollary 3.2, we assume $\gamma_0 = \min\{\gamma, 1/[4\,\mathrm{tr}(\mathbf{S}\mathbf{X}^\top\mathbf{X}\mathbf{S}^\top/N)]\}$ for some $\gamma \lesssim 1$. This modification is valid because Lemmas B.1, B.2, and C.1, used in the proof of parts 1–3 of Theorem 3.1, continue to hold under the alternative choice of stepsize.

Specifically, their proofs mainly rely on three properties: (1) $\mathbf{I} - \gamma_t\mathbf{S}\mathbf{X}^\top\mathbf{X}\mathbf{S}^\top/N \succeq \mathbf{0}$, (2) $\mathbb{P}(\gamma_0 < \gamma/t) \leqslant N^{-ct}$ for all $t \geqslant 1$ and (3) claim (15a) holds. Under the choice $\gamma_0 = \min\{\gamma, 1/[4\,\mathrm{tr}(\mathbf{S}\mathbf{X}^\top\mathbf{X}\mathbf{S}^\top/N)]\}$, the first two properties are satisfied by definition and by the Hanson–Wright inequality (see, e.g., exercise 2.17 in Wainwright (2019)). The third property follows from a similar symmetry property for $\gamma_0(\Gamma) := \min\{1/[4\,\mathrm{tr}(\Gamma\Gamma^\top/N)], \gamma\}$ as used in the proof of claim (15a).

### A.2.3 Proof of Corollary 3.3 and 3.4

These two corollaries follow immediately from combining parts 1–4 of Theorem 3.1 and some basic algebra.

### A.3 Relaxation of Assumption 1

In this section, we show that some conditions in Assumption 1 can be further relaxed. Concretely, we have

(a). The exact alignment of the eigenvectors of the prior and data covariance in Assumption 1D is not necessary. All results in Section 3 remain valid if Assumption 1D is replaced by

**Assumption 1D'** (Approximate source condition). *Let $(\lambda_i, \mathbf{v}_i)_{i>0}$ be the eigenvalues and eigenvectors of $\mathbf{H}$ and let $\mathbf{H}^{\mathbf{w}} = \mathbb{E}[\mathbf{w}^*\mathbf{w}^{*\top}]$. Assume $c\widetilde{\mathbf{H}}^{\mathbf{w}} \preceq \mathbf{H}^{\mathbf{w}} \preceq c'\widetilde{\mathbf{H}}^{\mathbf{w}}$ for some absolute constants $c' \geqslant c > 0$ and $\widetilde{\mathbf{H}}^{\mathbf{w}} \geq 0$ such that*

$$\text{for } i \neq j, \ \ \mathbf{v}_i^\top \widetilde{\mathbf{H}}^{\mathbf{w}} \mathbf{v}_j = 0; \ \ \text{and for } i > 0, \ \ \lambda_i \mathbf{v}_i^\top \widetilde{\mathbf{H}}^{\mathbf{w}} \mathbf{v}_i \asymp i^{-b}, \ \ \text{for some } b > 1.$$

(b). To establish the upper bounds for $\mathsf{Bias}, \mathsf{Var}, \mathsf{Approx}$ in Theorem 3.1 and the upper bounds in Corollary 3.2—3.4, Assumption 1A can be relaxed to

**Assumption 1A'** (sub-Gaussian design). $\mathbf{x} = \mathbf{H}^{1/2}\widetilde{\mathbf{x}}$, *where $\mathbb{E}[\widetilde{\mathbf{x}}\widetilde{\mathbf{x}}^\top] = \mathbf{I}$, and the vector $\widetilde{\mathbf{x}}$ is zero-mean and 1-sub-Gaussian, i.e., $\mathbb{E}[\widetilde{\mathbf{x}}] = \mathbf{0}$ and $\mathbb{E}[e^{\lambda\langle\mathbf{v},\widetilde{\mathbf{x}}\rangle}] \leqslant e^{\lambda^2/2}$ for any unit vector $\mathbf{v}$ and all $\lambda \in \mathbb{R}$.*

We provide some justification of the two relaxations below.

**Justification of (a).** By checking the proof of Theorem 3.1 and its corollaries, it can be seen that Assumption 1D is used to (1) give matching upper and lower bounds on $\mathbb{E}_{\mathbf{w}*}[\|\mathbf{w}_{0:k}^*\|_2^2], \mathbb{E}_{\mathbf{w}*}[\|\mathbf{w}_{k:\infty}^*\|_{\mathbf{H}_{k:\infty}}^2], \mu_i(\mathbf{SHH}^{\mathbf{w}}\mathbf{HS}^\top)$ for any $k \geqslant 0$ and $i \in [M]$ when controlling the approximation and bias error (see Lemma C.5 in Lin et al. (2024) and Lemma B.3); (2) give matching upper and lower bounds on $\mathbb{E}[\|\mathbf{w}^*\|_{\mathbf{H}}^2]$ when controlling the fluctuation error (see Lemma D.5). Under the alternative Assumption 1D', it is readily verified that the same bounds on these quantities can be established up to constant factors. Concretely, suppose there exists some parameter $\widetilde{\mathbf{w}}^*$ with prior $\mathbb{E}[\widetilde{\mathbf{w}}^*\widetilde{\mathbf{w}}^{*\top}] = \widetilde{\mathbf{H}}^{\mathbf{w}}$. Then $\widetilde{\mathbf{w}}^*$ satisfies Assumption 1D and

$$\mathbb{E}_{\mathbf{w}*}[\|\mathbf{w}_{0:k}^*\|_2^2] = \operatorname{tr}(\mathbf{H}_{0:k}^{\mathbf{w}}) \asymp \operatorname{tr}(\widetilde{\mathbf{H}}_{0:k}^{\mathbf{w}}) = \mathbb{E}_{\widetilde{\mathbf{w}}*}[\|\widetilde{\mathbf{w}}_{0:k}^*\|_2^2],$$

$$\mathbb{E}_{\mathbf{w}*}[\|\mathbf{w}_{k:\infty}^*\|_{\mathbf{H}_{k:\infty}}^2] = \operatorname{tr}(\mathbf{H}_{k:\infty}^{1/2}\mathbf{H}_{k:\infty}^{\mathbf{w}}\mathbf{H}_{k:\infty}^{1/2}) \asymp \operatorname{tr}(\mathbf{H}_{k:\infty}^{1/2}\widetilde{\mathbf{H}}_{k:\infty}^{\mathbf{w}}\mathbf{H}_{k:\infty}^{1/2}) = \mathbb{E}_{\widetilde{\mathbf{w}}*}[\|\widetilde{\mathbf{w}}_{k:\infty}^*\|_{\mathbf{H}_{k:\infty}}^2],$$

$$\mu_i(\mathbf{SHH}^{\mathbf{w}}\mathbf{HS}^\top) \overset{(i)}{\asymp} \mu_i(\mathbf{SH}\widetilde{\mathbf{H}}^{\mathbf{w}}\mathbf{HS}^\top),$$

$$\mathbb{E}[\|\mathbf{w}^*\|_{\mathbf{H}}^2] = \operatorname{tr}(\mathbf{H}^{1/2}\mathbf{H}^{\mathbf{w}}\mathbf{H}^{1/2}) \asymp \operatorname{tr}(\mathbf{H}^{1/2}\widetilde{\mathbf{H}}^{\mathbf{w}}\mathbf{H}^{1/2}) = \mathbb{E}_{\widetilde{\mathbf{w}}*}[\|\widetilde{\mathbf{w}}^*\|_{\mathbf{H}}^2],$$

where step (i) follows from the fact that $\mu_i(\mathbf{A}) \leqslant \mu_i(\mathbf{B})$ for all $i$ and any $0 \preceq \mathbf{A} \preceq \mathbf{B}$. Therefore, the proof of Theorem 3.1 and its corollaries goes through under the alternative Assumption 1D'.

**Justification of (b).** In short, for the upper bounds, the relaxation can be made since the Gaussian assumption is mainly used to establish certain concentration bounds (e.g., Bernstein's inequality), which also hold for sub-Gaussian vectors. More specifically, the Gaussian design in Assumption 1A' is used in our proof mainly in three ways: (1) to establish concentration bounds on the sample covariance (e.g., Eq. 10); (2) to allow the use of technical lemmas in Appendix E (e.g., Lemma E.3 and E.4); (3) to control the norm of sketched samples (e.g., to control $B_{\boldsymbol{\nu}}$ in Eq. 20).

Correspondingly, when $\mathbf{x}$ satisfies the alternative Assumption 1A', we can show that (1) the same concentration bounds hold on the sub-Gaussian sample covariance by e.g., Theorem 6.5 in Wainwright (2019); (2) all technical lemmas in Appendix E hold when the Gaussian sketching $\mathbf{S}$ is replaced by a row-wise sub-Gaussian matrix by concentration bounds on quadratic forms of sub-Gaussian vectors (e.g., Theorem 1 in Hsu et al. (2012)), and on sub-Gaussian covariance matrices (e.g., Example 1.5 in Zhivotovskiy (2024)); (3) the norm of sketched samples satisfy the same concentration bounds by e.g., Theorem 6.5 in Wainwright (2019).

On the other hand, for the lower bounds, the Gaussian assumption is still required in order to establish the conditional independence of $\tilde{\epsilon}_i = y_i - \mathbf{x}_i^\top \mathbf{S}^\top \mathbf{v}^*$ and $\mathbf{S}\mathbf{x}_i$ given $(\mathbf{S}, \mathbf{w}^*)$ in the proof of Theorem 3.1 and Lemma D.4.

In addition, we also conduct experiments to check our justification of (b). We generate data $\mathbf{x} = (x_1, \ldots, x_d)^\top$ from the distribution where $x_i$ are independent and

$$\mathbb{P}(x_i = 1) = \mathbb{P}(x_i = -1) = i^{-a}/c_0, \quad \mathbb{P}(x_i = 0) = 1 - 2 \cdot \mathbb{P}(x_i = 1),$$

with $a = 2, b = 1.5$ and $c_0 = 2\sum_{i=1}^d i^{-a}$. Note that $\mathbf{x}$ satisfies Assumption 1A' but not Assumption 1A when $d = \infty$. We run the experiment under the same setting and choice of hyperparameters as in Figure 1(a). Similar to the Gaussian case, in Figure 2, we observe that the excess test error of one-pass SGD and multi-pass SGD both exhibit power-law scaling in the number of effective samples $N_{\mathtt{eff}}$. Moreover, the fitted slopes are both close to the theoretical prediction in Corollary 3.3 ($0.34 \approx 0.33 = (1-b)/b$ for multi-pass SGD and $0.26 \approx 0.25 = (1-b)/a$ for one-pass SGD).

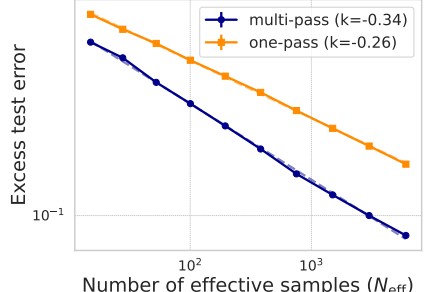

Figure 2: Multi-pass SGD versus one-pass SGD for non-Gaussian design. Multi-pass SGD is run for $L \asymp N^{a/b}$ steps. We use linear functions to fit the excess test error in log-log scale. The fitted exponents $(k)$ are close to the theoretical predictions in Corollary 3.3. The errorbars denote the $\pm 1$ standard deviation of the expected excess test error over 100 i.i.d. samples of $(\mathbf{S}, \mathbf{w}^*)$. Parameters: $\sigma^2 = 1$, $d = 10000$, $M = 1000$, $\gamma = 0.1$.

## B  Bias error

### B.1  An upper bound

**Lemma B.1** (An upper bound on the GD bias term). *Suppose $L_{\mathrm{eff}} \lesssim N^a/\gamma$ and Assumption 1A and 3 hold. Under the notation in Theorem 3.1 and its proof, for any $\mathbf{w}^* \in \mathbb{H}$ and $k \leqslant M/3$ such that $r(\mathbf{H}) \geqslant k + M$, the bias term*

$$\mathsf{Bias}(\mathbf{w}^*) = \mathbb{E}_\mathbf{X} \Big\| \prod_{t=1}^L \Big( \mathbf{I} - \gamma_t \widehat{\boldsymbol{\Sigma}} \Big) \mathbf{v}^* \Big\|_{\boldsymbol{\Sigma}}^2$$

$$\leqslant \frac{c \|\mathbf{w}_{0:k}^*\|_2^2}{L_{\mathrm{eff}} \gamma} \cdot \left[ \frac{\mu_{M/2}(\mathbf{A}_k)}{\mu_M(\mathbf{A}_k)} \right]^2 + B_\mathbf{B} \cdot \|\mathbf{w}_{k:\infty}^*\|_{\mathbf{H}_{k:\infty}}^2$$

*with probability at least $1 - e^{-\Omega(M)}$ over the randomness of $\mathbf{S}$, where $\mathbf{A}_k = \mathbf{S}_{k:\infty} \mathbf{H}_{k:\infty} \mathbf{S}_{k:\infty}$ and*

$$B_\mathbf{B} := c \left( 1 + [(L_{\mathrm{eff}}\gamma)^2 + 1]\Big( \frac{\mathrm{tr}^2(\boldsymbol{\Sigma}_{\tilde{k}:\infty})}{N^2} + \|\boldsymbol{\Sigma}_{\tilde{k}:\infty}\|_2^2 + \frac{\mathrm{tr}(\boldsymbol{\Sigma}_{\tilde{k}:\infty}^2)}{N} + \sqrt{\frac{\mathrm{tr}(\boldsymbol{\Sigma}_{\tilde{k}:\infty}^4)}{N}} \Big) \right)$$

*for some constant $c > 0$ and $\tilde{k} = \lfloor N/2 \rfloor$.[3]*

*Proof of Lemma B.1.* Without loss of generality, we assume the covariance matrix $\mathbf{H} = \mathrm{diag}\{\lambda_1, \lambda_2, \ldots, \lambda_d\}$ where $\lambda_i \geqslant \lambda_j$ for any $i \geqslant j$. Let $(\tilde{\lambda}_1, \tilde{\lambda}_2, \ldots, \tilde{\lambda}_M)$ denote the eigenvalues of $\boldsymbol{\Sigma}$ in non-increasing order. Moreover, we introduce $\mathbf{z}_1, \ldots \mathbf{z}_N \overset{iid}{\sim} \mathcal{N}(0, \mathbf{I}_M/N)$ and write $\mathbf{Z} = (\mathbf{z}_1, \ldots, \mathbf{z}_N)^\top$. It can be verified that $\mathbf{X}\mathbf{S}^\top/\sqrt{N} \overset{d}{=} \mathbf{Z}\boldsymbol{\Sigma}^{1/2}$ conditioned on $\mathbf{S}$. Throughout the proof, by a union bound argument, we w.l.o.g. assume the conditions (1), (2), (3) and (4) in Assumption 3 always hold.

Define $\mathsf{M} := \prod_{t=1}^L (\mathbf{I} - \gamma_t \widehat{\boldsymbol{\Sigma}}) \boldsymbol{\Sigma} \prod_{t=1}^L (\mathbf{I} - \gamma_t \widehat{\boldsymbol{\Sigma}})$ and recall that $\mathbf{v}^* = (\mathbf{S}\mathbf{H}\mathbf{S}^\top)^{-1} \mathbf{S}\mathbf{H}\mathbf{w}^*$. Substituting

$$\mathbf{S}\mathbf{H} = \begin{pmatrix} \mathbf{S}_{0:k}\mathbf{H}_{0:k} & \mathbf{S}_{k:\infty}\mathbf{H}_{k:\infty} \end{pmatrix}$$

into $\mathbf{v}^*$, we have

$$\mathsf{Bias}(\mathbf{w}^*) = \mathbb{E}_\mathbf{X}[\mathbf{v}^{*\top}\mathsf{M}\mathbf{v}^*]$$
$$= \mathbb{E}_\mathbf{X}[\mathbf{w}^{*\top}\mathbf{H}\mathbf{S}^\top(\mathbf{S}\mathbf{H}\mathbf{S}^\top)^{-1}\mathsf{M}(\mathbf{S}\mathbf{H}\mathbf{S}^\top)^{-1}\mathbf{S}\mathbf{H}\mathbf{w}^*]$$
$$\leqslant 2T_1 + 2T_2,$$

---

[3]If $\tilde{k} > M$ then $\boldsymbol{\Sigma}_{\tilde{k}:\infty} := \mathbf{0}$.

where

$$T_1 := \mathbb{E}_{\mathbf{X}}[(\mathbf{w}_{0:k}^*)^\top \mathbf{H}_{0:k} \mathbf{S}_{0:k}^\top (\mathbf{SHS}^\top)^{-1} \mathsf{M} (\mathbf{SHS}^\top)^{-1} \mathbf{S}_{0:k} \mathbf{H}_{0:k} \mathbf{w}_{0:k}^*], \tag{5}$$

$$T_2 := \mathbb{E}_{\mathbf{X}}[(\mathbf{w}_{k:\infty}^*)^\top \mathbf{H}_{k:\infty} \mathbf{S}_{k:\infty}^\top (\mathbf{SHS}^\top)^{-1} \mathsf{M} (\mathbf{SHS}^\top)^{-1} \mathbf{S}_{k:\infty} \mathbf{H}_{k:\infty} \mathbf{w}_{k:\infty}^*]. \tag{6}$$

We will show the following results at the end of the proof. First, with probability $1 - e^{-\Omega(M)}$

$$T_1 \leqslant \frac{c\|\mathbf{w}_{0:k}^*\|_2^2}{L_{\mathtt{eff}}\gamma} \cdot \left[\frac{\mu_{M/2}(\mathbf{A}_k)}{\mu_M(\mathbf{A}_k)}\right]^2 \tag{7a}$$

for some constant $c > 0$. Moreover,

$$T_2 \leqslant B_{\mathbf{B}} \cdot \|\mathbf{w}_{k:\infty}^*\|_{\mathbf{H}_{k:\infty}}^2. \tag{7b}$$

Combining Eq. (7a) and (7b) gives Lemma B.1.

**Proof of claim** (7a). By definition of $T_1$, we have

$$T_1 \leqslant \|\mathbf{H}_{0:k} \mathbf{S}_{0:k}^\top (\mathbf{SHS}^\top)^{-1} \mathsf{M} (\mathbf{SHS}^\top)^{-1} \mathbf{S}_{0:k} \mathbf{H}_{0:k}\|_2 \cdot \|\mathbf{w}_{0:k}^*\|_2^2$$
$$\leqslant \|\mathsf{M}\|_2 \cdot \|(\mathbf{SHS}^\top)^{-1} \mathbf{S}_{0:k} \mathbf{H}_{0:k}\|_2^2 \cdot \|\mathbf{w}_{0:k}^*\|_2^2.$$

for some constant $c > 0$. By Eq. (23) in the proof of Lemma D.1 in Lin et al. (2024), we have

$$\|(\mathbf{SHS}^\top)^{-1} \mathbf{S}_{0:k} \mathbf{H}_{0:k}\|_2 \leqslant c \cdot \frac{\mu_{M/2}(\mathbf{A}_k)}{\mu_M(\mathbf{A}_k)} \tag{8}$$

for some constant $c > 0$ with probability at least $1 - e^{-\Omega(M)}$. Thus, it remains to show

$$\mathbb{E}_{\mathbf{X}}[\|\mathsf{M}\|_2] \leqslant \frac{c}{L_{\mathtt{eff}}\gamma} \tag{9}$$

for some constant $c > 0$.

Let $\lambda > 0$ be a fixed value to be specified later. Note that

$$\|\mathsf{M}\|_2 = \left\|\prod_{t=1}^{L}(\mathbf{I} - \gamma_t \widehat{\boldsymbol{\Sigma}})\boldsymbol{\Sigma}^{1/2}\right\|_2^2 = \left\|\prod_{t=1}^{L}(\mathbf{I} - \gamma_t \widehat{\boldsymbol{\Sigma}})(\widehat{\boldsymbol{\Sigma}} + \lambda\mathbf{I}_M)^{1/2}(\widehat{\boldsymbol{\Sigma}} + \lambda\mathbf{I}_M)^{-1/2}\boldsymbol{\Sigma}^{1/2}\right\|_2^2$$

$$\leqslant \left\|\prod_{t=1}^{L}(\mathbf{I} - \gamma_t \widehat{\boldsymbol{\Sigma}})(\widehat{\boldsymbol{\Sigma}} + \lambda\mathbf{I}_M)^{1/2}\right\|_2^2 \cdot \|(\widehat{\boldsymbol{\Sigma}} + \lambda\mathbf{I}_M)^{-1/2}\boldsymbol{\Sigma}^{1/2}\|_2^2$$

$$\overset{(i)}{\leqslant} \left(\left\|\prod_{t=1}^{L}(\mathbf{I} - \gamma_t \widehat{\boldsymbol{\Sigma}})^2 \widehat{\boldsymbol{\Sigma}}\right\|_2 + \lambda\right) \cdot \|(\widehat{\boldsymbol{\Sigma}} + \lambda\mathbf{I}_M)^{-1/2}\boldsymbol{\Sigma}^{1/2}\|_2^2$$

$$\overset{(ii)}{\leqslant} \left(\frac{c}{L_{\mathtt{eff}}\gamma_0} + \lambda\right) \cdot \|(\widehat{\boldsymbol{\Sigma}} + \lambda\mathbf{I}_M)^{-1/2}\boldsymbol{\Sigma}^{1/2}\|_2^2,$$

where step (i) uses the fact that $\|\mathbf{I} - \gamma_t\widehat{\boldsymbol{\Sigma}}\|_2 \leqslant 1$ by the stepsize assumption and step (ii) follows from the stepsize assumption (2) that $\gamma_t = \gamma_0$ for $t \in [L_{\mathtt{eff}}]$, combined with the fact that $\sup_{x \in [0,1/\gamma_0]} x(1 - \gamma_0 x)^{2L_{\mathtt{eff}}} \leqslant c/(\gamma_0 L_{\mathtt{eff}})$ for some constant $c > 0$.

Recall that we assume $\mathbb{P}(\gamma_0 < \gamma/t) \leqslant N^{-ct}$ for some constant $c > 0$ and all $t \geqslant 1$. Thus, Eq. (9) follows immediately from choosing $\lambda = 1/(L_{\mathtt{eff}}\gamma)$ in the last display, applying Cauchy-Schwartz inequality and Lemma E.1, and noting that $(1 + L_{\mathtt{eff}}^2\gamma^2 \exp(-cN)) \lesssim 1$ for $L_{\mathtt{eff}} \lesssim N^a/\gamma$.

**Proof of claim** (7b). Let $\mathbf{B} = \mathbb{E}_{\mathbf{X}}\left[\boldsymbol{\Sigma}^{-1/2}\prod_{t=1}^{L}(\mathbf{I} - \gamma_t\widehat{\boldsymbol{\Sigma}})\boldsymbol{\Sigma}\prod_{t=1}^{L}(\mathbf{I} - \gamma_t\widehat{\boldsymbol{\Sigma}})\boldsymbol{\Sigma}^{-1/2}\right]$. By definition of $T_2$ in Eq. (6), we have

$$T_2 = \mathbf{w}_{k:\infty}^{*\top}\mathbf{H}_{k:\infty}\mathbf{S}_{k:\infty}^\top \boldsymbol{\Sigma}^{-1/2}\mathbf{B}\boldsymbol{\Sigma}^{-1/2}\mathbf{S}_{k:\infty}\mathbf{H}_{k:\infty}\mathbf{w}_{k:\infty}^*$$

$$\leqslant \|\mathbf{B}\|_2 \cdot \|\mathbf{H}_{k:\infty}^{1/2}\mathbf{S}_{k:\infty}^\top \boldsymbol{\Sigma}^{-1}\mathbf{S}_{k:\infty}\mathbf{H}_{k:\infty}^{1/2}\| \cdot \|\mathbf{w}_{k:\infty}^*\|_{\mathbf{H}_{k:\infty}}^2$$

$$\leqslant \|\mathbf{B}\|_2 \cdot \|\mathbf{w}^*_{k:\infty}\|^2_{\mathbf{H}_{k:\infty}},$$

where the last line follows since

$$\|\mathbf{H}^{1/2}_{k:\infty}\mathbf{S}^\top_{k:\infty}\mathbf{\Sigma}^{-1}\mathbf{S}_{k:\infty}\mathbf{H}^{1/2}_{k:\infty}\|_2 = \|\mathbf{H}^{1/2}_{k:\infty}\mathbf{S}^\top_{k:\infty}(\mathbf{S}_{0:k}\mathbf{H}_{0:k}\mathbf{S}^\top_{0:k} + \mathbf{S}_{k:\infty}\mathbf{H}_{k:\infty}\mathbf{S}^\top_{k:\infty})^{-1}\mathbf{S}_{k:\infty}\mathbf{H}^{1/2}_{k:\infty}\|_2$$
$$\leqslant \|\mathbf{H}^{1/2}_{k:\infty}\mathbf{S}^\top_{k:\infty}\mathbf{A}^{-1}_k\mathbf{S}_{k:\infty}\mathbf{H}^{1/2}_{k:\infty}\|_2 \leqslant 1.$$

In the following, we will show that $\|\mathbf{B}\|_2 \leqslant B_\mathbf{B}$, which immediately yields claim (7b).

Let $\overline{\mathbf{\Sigma}} = \mathbf{X}\mathbf{S}^\top\mathbf{S}\mathbf{X}^\top/N$. To compute $\|\mathbf{B}\|_2$, note that

$$\prod_{t=1}^{L}(\mathbf{I} - \gamma_t\widehat{\mathbf{\Sigma}}) = \prod_{t=1}^{L-1}(\mathbf{I} - \gamma_t\widehat{\mathbf{\Sigma}}) - \gamma_L\widehat{\mathbf{\Sigma}}\prod_{t=1}^{L-1}(\mathbf{I} - \gamma_t\widehat{\mathbf{\Sigma}})$$

$$\overset{(iii)}{=} \prod_{t=1}^{L-1}(\mathbf{I} - \gamma_t\widehat{\mathbf{\Sigma}}) - \frac{1}{N}\mathbf{S}\mathbf{X}^\top\Big[\gamma_L\prod_{t=1}^{L-1}(\mathbf{I} - \gamma_t\overline{\mathbf{\Sigma}})\Big]\mathbf{X}\mathbf{S}^\top$$

$$= \mathbf{I} - \frac{1}{N}\mathbf{S}\mathbf{X}^\top\Big[\sum_{i=0}^{L-1}\gamma_{i+1}\prod_{t=1}^{i}(\mathbf{I} - \gamma_t\overline{\mathbf{\Sigma}})\Big]\mathbf{X}\mathbf{S}^\top =: \mathbf{I} - \mathbf{C},$$

where step (iii) uses $\mathbf{X}\mathbf{S}^\top(\mathbf{I} - \gamma_L\widehat{\mathbf{\Sigma}}) = (\mathbf{I} - \gamma_L\overline{\mathbf{\Sigma}})\mathbf{X}\mathbf{S}^\top$. Recall that $\mathbf{X}\mathbf{S}^\top/\sqrt{N} \overset{d}{=} \mathbf{Z}\mathbf{\Sigma}^{1/2}$ conditioned on $\mathbf{S}$, we can thus rewrite

$$\mathbf{B} = \mathbb{E}_\mathbf{Z}[\mathbf{\Sigma}^{-1/2}(\mathbf{I} - \mathbf{C})\mathbf{\Sigma}(\mathbf{I} - \mathbf{C})\mathbf{\Sigma}^{-1/2}] \preceq 2\mathbf{I} + 2\mathbb{E}[\mathbf{\Sigma}^{-1/2}\mathbf{C}\mathbf{\Sigma}\mathbf{C}\mathbf{\Sigma}^{-1/2}]$$

$$= 2\mathbf{I} + 2\mathbb{E}_\mathbf{Z}\Big[\mathbf{Z}^\top\Big[\sum_{i=0}^{L-1}\gamma_{i+1}\prod_{t=1}^{i}(\mathbf{I} - \gamma_t\overline{\mathbf{\Sigma}})\Big]\mathbf{Z}\mathbf{\Sigma}^2\mathbf{Z}^\top\Big[\sum_{i=0}^{L-1}\gamma_{i+1}\prod_{t=1}^{i}(\mathbf{I} - \gamma_t\overline{\mathbf{\Sigma}})\Big]\mathbf{Z}\Big], \text{ where } \overline{\mathbf{\Sigma}} = \mathbf{Z}\mathbf{\Sigma}\mathbf{Z}^\top.$$

Introduce the shorthand $\mathbf{R}_1 = \sum_{i=0}^{L-1}\gamma_{i+1}\prod_{t=1}^{i}(\mathbf{I} - \gamma_t\overline{\mathbf{\Sigma}})$ and $\mathbf{R}_1(k) := (\mathbf{I} - (\mathbf{I} - \gamma_{kL_{\text{eff}}+1}\overline{\mathbf{\Sigma}})^{L_{\text{eff}}})/\overline{\mathbf{\Sigma}}$ for $k \in [0, \lfloor\log L_{\text{eff}}\rfloor - 1]$. Note that $\|\mathbf{R}_1(k)\|_2 \leqslant L_{\text{eff}} \cdot \gamma_{kL_{\text{eff}}+1}$ since $\sup_{x\in[0,1/\gamma_{kL_{\text{eff}}+1}]}[(1 - (1 - \gamma_{kL_{\text{eff}}+1}x)^{L_{\text{eff}}})/x] = L_{\text{eff}} \cdot \gamma_{kL_{\text{eff}}+1}$. Therefore

$$\|\mathbf{R}_1\|_2 = \Big\|\sum_{i=0}^{L-1}\gamma_{i+1}\prod_{t=1}^{i}(\mathbf{I} - \gamma_t\overline{\mathbf{\Sigma}})\Big\|_2 \leqslant \Big\|\sum_{k=0}^{\lfloor\log L_{\text{eff}}\rfloor-1}\mathbf{R}_1(k)\Big\|_2 \leqslant \sum_{k=0}^{\lfloor\log L_{\text{eff}}\rfloor-1} L_{\text{eff}} \cdot \gamma_{kL_{\text{eff}}+1} \leqslant 2L_{\text{eff}}\gamma,$$

where the last inequality follows from (2) and the definition of $\gamma_0$.

We consider two cases.

Case 1: $M \leqslant N/2$. In this case, we have

$$\mathbf{Z}\mathbf{\Sigma}^2\mathbf{Z}^\top \preceq 5 \cdot (\mathbf{Z}\mathbf{\Sigma}\mathbf{Z}^\top)^2 \tag{10}$$

with probability at least $1 - e^{-\Omega(N)}$ since $\mathbb{P}(\mathbf{Z}^\top\mathbf{Z} \geqslant \mathbf{I}_M/5) \geqslant 1 - e^{-\Omega(N)}$ by concentration of Guassian covariance matrix (see e.g. Theorem 6.1 in Wainwright (2019)). Moreover, since $\text{tr}(\mathbf{\Sigma}^2) \lesssim 1$,

$$\mathbf{Z}^\top\mathbf{R}_1\mathbf{Z}\mathbf{\Sigma}^2\mathbf{Z}^\top\mathbf{R}_1\mathbf{Z} \preceq c \cdot \mathbf{Z}^\top\mathbf{R}_1\mathbf{Z}\mathbf{Z}^\top\mathbf{R}_1\mathbf{Z} \preceq c\|\mathbf{Z}^\top\mathbf{R}_1\mathbf{Z}\|^2_2 \cdot \mathbf{I}_M$$

$$\preceq c(L_{\text{eff}}\gamma)^2\|\mathbf{Z}^\top\mathbf{Z}\|^2_2 \cdot \mathbf{I}_M.$$

Therefore,

$$\mathbb{E}_\mathbf{Z}[\mathbf{Z}^\top\mathbf{R}_1\mathbf{Z}\mathbf{\Sigma}^2\mathbf{Z}^\top\mathbf{R}_1\mathbf{Z}] \preceq c\mathbb{E}_\mathbf{Z}[\mathbf{Z}^\top\mathbf{R}_1\overline{\mathbf{\Sigma}}^2\mathbf{R}_1\mathbf{Z}] + c\mathbb{E}[(L_{\text{eff}}\gamma)^2\|\mathbf{Z}^\top\mathbf{Z}\|^2_2\mathbf{1}_{\{\mathbf{Z}^\top\mathbf{Z}\not\geqslant\mathbf{I}_M/5\}}] \cdot \mathbf{I}_M \tag{11}$$

$$\preceq c\mathbb{E}_\mathbf{Z}[\mathbf{Z}^\top\mathbf{Z}] + c(L_{\text{eff}}\gamma)^2\exp(-c'N) \cdot \mathbf{I}_M \preceq c\mathbf{I}_M$$

for some constant $c, c' > 0$, where the second line uses the fact that $\|\mathbf{R}_1\overline{\mathbf{\Sigma}}\|_2 = \|\mathbf{I} - \prod_{t=1}^{L}(\mathbf{I} - \gamma_t\overline{\mathbf{\Sigma}})\|_2 \leqslant 1$, concentration properties of the empirical covariance matrix $\mathbf{Z}^\top\mathbf{Z}$, and $\mathbb{E}_\mathbf{Z}[\mathbf{Z}^\top\mathbf{Z}] = \mathbf{I}_M$. As a result, $\|\mathbf{B}\|_2 \leqslant 2 + 2\|\mathbb{E}_\mathbf{Z}[\mathbf{Z}^\top\mathbf{R}_1\mathbf{Z}\mathbf{\Sigma}^2\mathbf{Z}^\top\mathbf{R}_1\mathbf{Z}]\|_2 \lesssim 1$.

Case2: $M > N/2$. Let $\tilde{k} = N/2$. W.l.o.g., we assume $\boldsymbol{\Sigma}$ is a diagonal matrix with eigenvalues $\tilde{\lambda}_1, \ldots, \tilde{\lambda}_M$ in non-increasing order. With probability at least $1 - e^{-\Omega(N)}$, we have the decomposition

$$
\begin{aligned}
\mathbf{Z}^\top \mathbf{R}_1 \mathbf{Z} \boldsymbol{\Sigma}^2 \mathbf{Z}^\top \mathbf{R}_1 \mathbf{Z} &\leq 2\mathbf{Z}^\top \mathbf{R}_1 (\mathbf{Z}_{0:\tilde{k}} \boldsymbol{\Sigma}_{0:\tilde{k}}^2 \mathbf{Z}_{0:\tilde{k}}^\top) \mathbf{R}_1 \mathbf{Z} + 2\mathbf{Z}^\top \mathbf{R}_1 (\mathbf{Z}_{\tilde{k}:\infty} \boldsymbol{\Sigma}_{\tilde{k}:\infty}^2 \mathbf{Z}_{\tilde{k}:\infty}^\top) \mathbf{R}_1 \mathbf{Z} \\
&\leq c \mathbf{Z}^\top \mathbf{R}_1 (\mathbf{Z}_{0:\tilde{k}} \boldsymbol{\Sigma}_{0:\tilde{k}} \mathbf{Z}_{0:\tilde{k}}^\top)^2 \mathbf{R}_1 \mathbf{Z} + 2\mathbf{Z}^\top \mathbf{R}_1 (\mathbf{Z}_{\tilde{k}:\infty} \boldsymbol{\Sigma}_{\tilde{k}:\infty}^2 \mathbf{Z}_{\tilde{k}:\infty}^\top) \mathbf{R}_1 \mathbf{Z} \\
&\leq c(\|\mathbf{R}_1 (\mathbf{Z}_{0:\tilde{k}} \boldsymbol{\Sigma}_{0:\tilde{k}} \mathbf{Z}_{0:\tilde{k}}^\top)\|_2^2 + \|\mathbf{Z}_{\tilde{k}:\infty} \boldsymbol{\Sigma}_{\tilde{k}:\infty}^2 \mathbf{Z}_{\tilde{k}:\infty}^\top\|_2) \mathbf{Z}^\top \mathbf{Z} \\
&\leq c(\|\mathbf{R}_1 (\mathbf{Z}_{\tilde{k}:\infty} \boldsymbol{\Sigma}_{\tilde{k}:\infty} \mathbf{Z}_{\tilde{k}:\infty}^\top)\|_2^2 + \|\mathbf{R}_1 \overline{\boldsymbol{\Sigma}}\|_2^2 + \|\mathbf{Z}_{\tilde{k}:\infty} \boldsymbol{\Sigma}_{\tilde{k}:\infty}^2 \mathbf{Z}_{\tilde{k}:\infty}^\top\|_2) \mathbf{Z}^\top \mathbf{Z},
\end{aligned}
$$

where the second line use $\mathbf{Z}_{0:\tilde{k}}^\top \mathbf{Z}_{0:\tilde{k}} \geq \mathbf{I}_k / 5$ with probability at least $1 - e^{-\Omega(N)}$, the last line follows from a triangle inequality. Since $\|\mathbf{R}_1\|_2 \leqslant L_{\texttt{eff}} \gamma$ and $\|\mathbf{R}_1 \overline{\boldsymbol{\Sigma}}\|_2 = \|\mathbf{I} - \prod_{t=1}^L (\mathbf{I} - \gamma_t \overline{\boldsymbol{\Sigma}})\|_2 \leqslant 1$, continuing the calculation, we obtain

$$
\mathbf{Z}^\top \mathbf{R}_1 \mathbf{Z} \boldsymbol{\Sigma}^2 \mathbf{Z}^\top \mathbf{R}_1 \mathbf{Z} \leq c\big((L_{\texttt{eff}} \gamma)^2 \|\mathbf{Z}_{\tilde{k}:\infty} \boldsymbol{\Sigma}_{\tilde{k}:\infty} \mathbf{Z}_{\tilde{k}:\infty}^\top\|_2^2 + 1 + \|\mathbf{Z}_{\tilde{k}:\infty} \boldsymbol{\Sigma}_{\tilde{k}:\infty}^2 \mathbf{Z}_{\tilde{k}:\infty}^\top\|_2\big) \mathbf{Z}^\top \mathbf{Z}. \quad (12)
$$

Since we have by Lemma E.3 that

$$
\|\mathbf{Z}_{\tilde{k}:\infty} \boldsymbol{\Sigma}_{\tilde{k}:\infty} \mathbf{Z}_{\tilde{k}:\infty}^\top\|_2 \leqslant c\left( \frac{\text{tr}(\boldsymbol{\Sigma}_{\tilde{k}:\infty})}{N} + \|\boldsymbol{\Sigma}_{\tilde{k}:\infty}\|_2 + \sqrt{\frac{\text{tr}(\boldsymbol{\Sigma}_{\tilde{k}:\infty}^2)}{N}} \right)
$$

$$
+ c\left( \frac{\|\boldsymbol{\Sigma}_{\tilde{k}:\infty}\|_2}{N} \log \frac{1}{\delta} + \frac{\sqrt{\text{tr}(\boldsymbol{\Sigma}_{\tilde{k}:\infty}^2) \log(1/\delta)}}{N} \right) \quad (13)
$$

with probability at least $1 - \delta$, and $\text{tr}(\boldsymbol{\Sigma}_{\tilde{k}:\infty}^2) \leqslant \text{tr}(\boldsymbol{\Sigma}^2) \lesssim 1$, it can be verified by a standard truncation argument that

$$
\mathbb{E}[\|\mathbf{Z}_{\tilde{k}:\infty} \boldsymbol{\Sigma}_{\tilde{k}:\infty} \mathbf{Z}_{\tilde{k}:\infty}^\top\|_2^2 \cdot \mathbf{Z}^\top \mathbf{Z}] \leq c\left( \frac{\text{tr}^2(\boldsymbol{\Sigma}_{\tilde{k}:\infty})}{N^2} + \|\boldsymbol{\Sigma}_{\tilde{k}:\infty}\|_2^2 + \frac{\text{tr}(\boldsymbol{\Sigma}_{\tilde{k}:\infty}^2)}{N} \right) \cdot \mathbf{I}_M.
$$

A similar bound can be established for $\mathbb{E}[\|\mathbf{Z}_{\tilde{k}:\infty} \boldsymbol{\Sigma}_{\tilde{k}:\infty}^2 \mathbf{Z}_{\tilde{k}:\infty}^\top\|_2]$. Finally, substituting the bounds on the expectations into Eq. (12), we obtain

$$
\begin{aligned}
\|\mathbf{B}\|_2 &\leqslant 2 + 2\|\mathbb{E}[\mathbf{Z}^\top \mathbf{R}_1 \mathbf{Z} \boldsymbol{\Sigma}^2 \mathbf{Z}^\top \mathbf{R}_1 \mathbf{Z}]\|_2 \\
&\leqslant c\left( 1 + [(L_{\texttt{eff}} \gamma)^2 + 1]\left( \frac{\text{tr}^2(\boldsymbol{\Sigma}_{\tilde{k}:\infty})}{N^2} + \|\boldsymbol{\Sigma}_{\tilde{k}:\infty}\|_2^2 + \frac{\text{tr}(\boldsymbol{\Sigma}_{\tilde{k}:\infty}^2)}{N} + \sqrt{\frac{\text{tr}(\boldsymbol{\Sigma}_{\tilde{k}:\infty}^4)}{N}} \right) \right).
\end{aligned}
$$

$\square$

## B.2  A lower bound

**Lemma B.2** (A lower bound on the GD bias term). *Let Assumption 1A and 3 hold. Define* $\mathbf{H}^{\mathbf{w}} := \mathbb{E}[\mathbf{w}^* \mathbf{w}^{*\top}]$ *and* $\boldsymbol{\Sigma}_{\mathbf{w}} := \mathbf{S} \mathbf{H} \mathbf{H}^{\mathbf{w}} \mathbf{H} \mathbf{S}^\top$. *Under the notation in Theorem 3.1 and its proof, the bias term satisfies*

$$
\begin{aligned}
\mathbb{E}_{\mathbf{w}*}[\text{Bias}(\mathbf{w}^*)] &= \mathbb{E}_{\mathbf{w}*} \mathbb{E}\Big\| \prod_{t=1}^L \big( \mathbf{I} - \gamma_t \widehat{\boldsymbol{\Sigma}} \big) \mathbf{v}^* \Big\|_{\boldsymbol{\Sigma}}^2 \\
&\gtrsim \sum_{i=2\bar{t}+1}^M \frac{\mu_{3i}(\boldsymbol{\Sigma}_{\mathbf{w}})}{\mu_i(\boldsymbol{\Sigma})}
\end{aligned}
$$

*with probability at least* $1 - e^{-\Omega(M)}$, *where* $\bar{t} := \mathbb{E}_{\mathbf{X}}[\#\{i \in [M] : \widehat{\lambda}_i L_{\texttt{eff}} \gamma_0 > 1/4\}]$ *and* $(\widehat{\lambda}_i)_{i=1}^M$ *are the eigenvalues of* $\widehat{\boldsymbol{\Sigma}}$.

*Proof of Lemma B.2.* Similar to the proof of Lemma B.1, w.l.o.g., we assume the covariance matrix $\mathbf{H} = \text{diag}\{\lambda_1, \lambda_2, \ldots, \lambda_d\}$ where $\lambda_i \geqslant \lambda_j$ for any $i \geqslant j$. Let $(\tilde{\lambda}_1, \tilde{\lambda}_2, \ldots, \tilde{\lambda}_M)$ denote the eigenvalues of $\boldsymbol{\Sigma}$ in non-increasing order. Moreover, we introduce $\mathbf{z}_1, \ldots \mathbf{z}_N \overset{iid}{\sim} \mathcal{N}(0, \mathbf{I}_M/N)$ and write $\mathbf{Z} = (\mathbf{z}_1, \ldots, \mathbf{z}_N)^\top$. It can be shown that $\mathbf{X} \mathbf{S}^\top / \sqrt{N} \overset{d}{=} \mathbf{Z} \boldsymbol{\Sigma}^{1/2}$ conditioned on $\mathbf{S}$.

Let $C_L := \prod_{t=1}^{L}(\mathbf{I} - \gamma_t \widehat{\boldsymbol{\Sigma}})$. By definition,

$$\text{Bias}(\mathbf{w}^*) = \mathbb{E}_{\mathbf{X}}\left\| \prod_{t=1}^{L} (\mathbf{I} - \gamma_t \widehat{\boldsymbol{\Sigma}})\mathbf{v}^* \right\|_{\boldsymbol{\Sigma}}^2 = \mathbf{v}^{*\top}\mathbb{E}_{\mathbf{X}}[C_L \boldsymbol{\Sigma} C_L]\mathbf{v}^*. \tag{14}$$

Adopt the shorthand $\boldsymbol{\Sigma}_{\mathbf{w}}$ for $\mathbf{SHH^w HS}^\top$. Substituting the definition of $\mathbf{v}^*$ into the expression and noting that $\mathbb{E}[\mathbf{w}^*\mathbf{w}^{*\top}] = \mathbf{H^w}$, we have

$$\begin{aligned}
\mathbb{E}_{\mathbf{w}*}[\text{Bias}(\mathbf{w}^*)] &= \mathbb{E}_{\mathbf{w}*}[\mathbf{v}^{*\top}\mathbb{E}_{\mathbf{X}}[C_L \boldsymbol{\Sigma} C_L]\mathbf{v}^*]\\
&= \text{tr}(\mathbf{HS}^\top(\mathbf{SHS}^\top)^{-1}\mathbb{E}_{\mathbf{X}}[C_L \boldsymbol{\Sigma} C_L](\mathbf{SHS}^\top)^{-1}\mathbf{SHH^w})\\
&\overset{(i)}{\geqslant} \text{tr}(\mathbf{HS}^\top(\mathbf{SHS}^\top)^{-1}\mathbb{E}_{\mathbf{X}}[C_L]\boldsymbol{\Sigma}\mathbb{E}_{\mathbf{X}}[C_L^\top](\mathbf{SHS}^\top)^{-1}\mathbf{SHH^w})\\
&= \mathbb{E}_{\mathbf{X}}[\text{tr}(\boldsymbol{\Sigma}^{-1/2}\mathbb{E}_{\mathbf{X}}[C_L]\boldsymbol{\Sigma}\mathbb{E}_{\mathbf{X}}[C_L^\top]\boldsymbol{\Sigma}^{-1/2}\boldsymbol{\Sigma}^{-1/2}\boldsymbol{\Sigma}_{\mathbf{w}}\boldsymbol{\Sigma}^{-1/2})],
\end{aligned}$$

where step (i) uses the fact that $\mathbb{E}[\mathbf{Y}]\mathbb{E}[\mathbf{Y}^\top] \preceq \mathbb{E}[\mathbf{YY}^\top]$ for any random matrix $\mathbf{Y}$. We claim that

$$\boldsymbol{\Sigma}^{-1/2}\mathbb{E}_{\mathbf{X}}[C_L]\boldsymbol{\Sigma}^{1/2} = \mathbb{E}_{\mathbf{X}}[C_L], \quad \text{and} \tag{15a}$$

$$\mu_{M-i+1}(\mathbb{E}_{\mathbf{X}}[C_L]) \geqslant \frac{1}{2e} \tag{15b}$$

for all $i \in [2\bar{t}+1, M]$, where $\bar{t} := \mathbb{E}_{\mathbf{X}}[\#\{i \in [M] : \widehat{\lambda}_i L\gamma_0 > 1/4\}]$.

The proof of these two claims will be given momentarily. Continuing the calculation using the claims and Von Neumann's trace inequality, we obtain

$$\begin{aligned}
\mathbb{E}_{\mathbf{w}*}\text{Bias}(\mathbf{w}^*) &\geqslant \mathbb{E}_{\mathbf{X}}[\text{tr}(\mathbb{E}_{\mathbf{X}}[C_L]^2\boldsymbol{\Sigma}^{-1/2}\boldsymbol{\Sigma}_{\mathbf{w}}\boldsymbol{\Sigma}^{-1/2})]\\
&\geqslant \sum_{i=1}^{M}\mu_{M-i+1}^2(\mathbb{E}_{\mathbf{X}}[C_L]) \cdot \mu_i(\boldsymbol{\Sigma}^{-1/2}\boldsymbol{\Sigma}_{\mathbf{w}}\boldsymbol{\Sigma}^{-1/2})\\
&\geqslant \sum_{i=2\bar{t}+1}^{M}\mu_{M-i+1}^2(\mathbb{E}_{\mathbf{X}}[C_L]) \cdot \mu_i(\boldsymbol{\Sigma}^{-1/2}\boldsymbol{\Sigma}_{\mathbf{w}}\boldsymbol{\Sigma}^{-1/2}) \gtrsim \sum_{i=2\bar{t}+1}^{M}\mu_i(\boldsymbol{\Sigma}^{-1/2}\boldsymbol{\Sigma}_{\mathbf{w}}\boldsymbol{\Sigma}^{-1/2}).
\end{aligned}$$

Since $\mu_{i+j+1}(XY) \leqslant \mu_{i+1}(X)\mu_{j+1}(Y)$ for all $i,j$ and any matrices $X, Y$ of matching dimensions, we have

$$\mu_i(\boldsymbol{\Sigma}^{-1/2}\boldsymbol{\Sigma}_{\mathbf{w}}\boldsymbol{\Sigma}^{-1/2}) \geqslant \frac{\mu_{2i-1}(\boldsymbol{\Sigma}_{\mathbf{w}}\boldsymbol{\Sigma}^{-1/2})}{\mu_i(\boldsymbol{\Sigma}^{1/2})} \geqslant \frac{\mu_{3i-2}(\boldsymbol{\Sigma}_{\mathbf{w}})}{\mu_i^2(\boldsymbol{\Sigma}^{1/2})} \geqslant \frac{\mu_{3i}(\boldsymbol{\Sigma}_{\mathbf{w}})}{\mu_i(\boldsymbol{\Sigma})}.$$

Combining the last two displays yields the desired result.

**Proof of claim** (15a). Define the learning rate $\gamma_0(\Gamma) = \min\{1/[4\max_j \|\Gamma_{\cdot,j}\|_2^2], \gamma\}$ for any matrix $\Gamma \in \mathbb{R}^{M\times N}$ and define $\gamma_t(\Gamma)$ for all $t \in [L]$ according to (2). Let $\boldsymbol{\Sigma} = \mathbf{U}\tilde{\boldsymbol{\Gamma}}\mathbf{U}^\top$ be the singular value decomposition with $\mathbf{UU}^\top = \mathbf{I}_M$ and $\tilde{\boldsymbol{\Gamma}} = \text{diag}\{\tilde{\lambda}_1, \ldots, \tilde{\lambda}_M\}$ being a diagonal matrix with $\tilde{\lambda}_1 \geqslant \tilde{\lambda}_2 \geqslant \ldots \tilde{\lambda}_M \geqslant 0$. Note that $\mathbf{SX}^\top/\sqrt{N} \overset{d}{=} \mathbf{U}\tilde{\boldsymbol{\Gamma}}^{1/2}\mathbf{Z}^\top$ conditioned on $\mathbf{S}$ and $\mathbf{U}^\top\widehat{\boldsymbol{\Sigma}}\mathbf{U} \overset{d}{=} \tilde{\boldsymbol{\Gamma}}^{1/2}\mathbf{Z}^\top\mathbf{Z}\tilde{\boldsymbol{\Gamma}}^{1/2}$. Therefore

$$\begin{aligned}
\mathbb{E}_{\mathbf{X}}[C_L] &= \mathbb{E}_{\mathbf{X}}\Big[\prod_{t=1}^{L}(\mathbf{I} - \gamma_t(\mathbf{SX}^\top)\widehat{\boldsymbol{\Sigma}})\Big] = \mathbf{U}\mathbb{E}_{\mathbf{X}}\Big[\prod_{t=1}^{L}(\mathbf{I} - \gamma_t(\sqrt{N}\mathbf{U}\tilde{\boldsymbol{\Gamma}}^{1/2}\mathbf{Z}^\top)\mathbf{U}^\top\widehat{\boldsymbol{\Sigma}}\mathbf{U})\Big]\mathbf{U}^\top\\
&= \mathbf{U}\mathbb{E}_{\mathbf{Z}}\Big[\prod_{t=1}^{L}(\mathbf{I} - \gamma_t(\sqrt{N}\tilde{\boldsymbol{\Gamma}}^{1/2}\mathbf{Z}^\top)\tilde{\boldsymbol{\Gamma}}^{1/2}\mathbf{Z}^\top\mathbf{Z}\tilde{\boldsymbol{\Gamma}}^{1/2})\Big]\mathbf{U}^\top.
\end{aligned}$$

Adopt the shorthand notation $\mathcal{U} = \mathbf{Z}\tilde{\boldsymbol{\Gamma}}^{1/2}$ and write $\mathcal{U} = (\boldsymbol{\mu}_1, \ldots, \boldsymbol{\mu}_N)^\top$. It suffices to show (note that $\gamma_k$ is equal to $\gamma_0$ up to some $k$-dependent constant factor)

$$\mathbf{M} := \mathbb{E}_{\mathbf{Z}}[\gamma_0(\sqrt{N}\mathcal{U}^\top)^K \cdot (\mathcal{U}^\top\mathcal{U})^K]$$

is a diagonal matrix for any $K \geqslant 0$. Consider the $kl$-entry $\mathbf{M}_{kl}$. It can be written as the sum of terms of the form $\mu_{i_1,j_1}\mu_{i_2,j_2}\cdots\mu_{i_{2K},j_{2K}}$ with $j_1 = k$, $j_{2K} = l$, $j_{2m} = j_{2m+1}, m \in [K-1]$. When $k \neq l$, there exists some $i \in [N]$ such that $\mu_{i,k}$ appears odd number of times in the product. Since flipping the sign of $\mu_{i,k}$ does not change $\gamma_0(\sqrt{N}\mathcal{U}^\top)$, and $\mu_{i,j}$ are independent symmetric Gaussian variables, it follows that $\mathbb{E}_{\mathbf{Z}}[\gamma_0(\sqrt{N}\mathcal{U}^\top)^K\mu_{i_1,j_1}\mu_{i_2,j_2}\cdots\mu_{i_{2K},j_{2K}}] = 0$. Consequently, we conclude that $\mathbf{M}_{kl} = 0$ for $k \neq l$ and $\mathbf{M}$ is a diagonal matrix.

**Proof of claim** (7b). Let $\widehat{\boldsymbol{\Sigma}} = \widehat{\mathbf{U}}\widehat{\boldsymbol{\Gamma}}\widehat{\mathbf{U}}^\top$ be the singular value decomposition with $\widehat{\boldsymbol{\Gamma}} = \text{diag}\{\widehat{\lambda}_1, \ldots, \widehat{\lambda}_M\}$ and $\widehat{\lambda}_1 \geqslant \ldots \geqslant \widehat{\lambda}_M$. Then we have

$$\prod_{t=1}^{L}(\mathbf{I} - \gamma_t \widehat{\boldsymbol{\Sigma}}) \succeq (\mathbf{I} - 2\gamma_0 \widehat{\boldsymbol{\Sigma}})^{L_{\text{eff}}} = \widehat{\mathbf{U}}(\mathbf{I} - 2\gamma_0 \widehat{\boldsymbol{\Gamma}})^{L_{\text{eff}}}\widehat{\mathbf{U}}^\top \succeq \frac{(\mathbf{I}_M - \widehat{\mathbf{U}}_{0:t}\widehat{\mathbf{U}}_{0:t}^\top)}{e},$$

where $\tilde{t} := \#\{i \in [M] : \widehat{\lambda}_i L_{\text{eff}}\gamma_0 > 1/4\}$. Here, the first inequality follows from $\prod_{k=0}^{\lfloor \log L_{\text{eff}}\rfloor - 1}(\mathbf{I} - \gamma_{kL_{\text{eff}}+1}\widehat{\boldsymbol{\Sigma}}) \succeq \mathbf{I} - 2\gamma_0 \widehat{\boldsymbol{\Sigma}}$ since $(1 - t_1)(1 - t_2) \geqslant 1 - t_1 - t_2$ for all $t_1, t_2 \in [0, 1]$; the second inequality uses $(1 - x)^{L_{\text{eff}}} \geqslant \exp(-2L_{\text{eff}}x) \geqslant e^{-1}$ for $x \in [0, 1/(2L_{\text{eff}})]$. Therefore,

$$\mathbb{E}_{\mathbf{X}}[\mathsf{C}_L] = \mathbb{E}_{\mathbf{X}}(\mathbf{I} - \gamma_0 \widehat{\boldsymbol{\Sigma}})^L \succeq \frac{1}{e}\mathbf{I}_M - \frac{1}{e}\mathbb{E}[\widehat{\mathbf{U}}_{0:t}\widehat{\mathbf{U}}_{0:t}^\top].$$

Since $\text{tr}(\mathbb{E}[\widehat{\mathbf{U}}_{0:t}\widehat{\mathbf{U}}_{0:t}^\top]) = \mathbb{E}[\tilde{t}] = \bar{t}$, it follows that $\mathbb{E}[\widehat{\mathbf{U}}_{0:t}\widehat{\mathbf{U}}_{0:t}^\top]$ has at most $2\mathbb{E}[\tilde{t}]$ eigenvalues greater than $1/2$. Since $X \succeq Y \succeq \mathbf{0}_M$ implies $\mu_i(X) \geqslant \mu_i(Y)$ for all $i \in [M], X, Y \in \mathbb{R}^{M \times M}$ by Weyl's inequality, it follows that

$$\mu_{M-i+1}(\mathbb{E}_{\mathbf{X}}[\mathsf{C}_L]) \geqslant \frac{1}{2e}$$

for all $i \geqslant 2\mathbb{E}_{\mathbf{X}}[\#\{i \in [M] : \widehat{\lambda}_i L\gamma_0 > 1/4\}] + 1$.

$\square$

### B.3 Bias error under the source condition

**Lemma B.3** (Bias bounds under the source condition). *Let Assumption 1 hold, $a > b-1$, and assume $L_{\text{eff}} \lesssim N^a/\gamma$. Under the notation in Theorem 3.1 and its proof, there exist some $(a, b)$-dependent constants $c, c' > 0$ such that when $\gamma \leqslant c/\log N$,*

$$\mathbb{E}_{\mathbf{w}^*}[\mathsf{Bias}(\mathbf{w}^*)] \lesssim \max\{(L_{\text{eff}}\gamma)^{(1-b)/a}, M^{1-b}\},$$
$$\mathbb{E}_{\mathbf{w}^*}[\mathsf{Bias}(\mathbf{w}^*)] \gtrsim (L_{\text{eff}}\gamma)^{(1-b)/a} \text{ when } (L_{\text{eff}}\gamma)^{1/a} \leqslant M/c'.$$

*with probability at least $1 - \exp(-\Omega(M))$ over the randomness of $\mathbf{S}$.*

*Proof of Lemma B.3.* The proof follows from applying Lemma B.1 and Lemma B.2 under Assumption 1. We begin by verifying the conditions required in these two lemmas.

**Verification of conditions (1)–(4) in Assumption 3.** First, by Lemma E.5, we have $\mu_j(\boldsymbol{\Sigma}) \asymp j^{-a}$ with probability at least $1 - \exp(-\Omega(M))$ over the randomness of $\mathbf{S}$. Since $a > 1$, it follows that $\gamma \lesssim 1 \lesssim \min\{1, c/\text{tr}(\boldsymbol{\Sigma})\}$ and $\text{tr}(\boldsymbol{\Sigma}^2) \lesssim 1$. Thus, conditions (1) and (2) in Assumption 3 are satisfied. Moreover, when $L \lesssim N^a$, we have

$$\sum_{i=1}^{M} \frac{\mu_i(\boldsymbol{\Sigma})}{\mu_i(\boldsymbol{\Sigma}) + 1/(L_{\text{eff}}\gamma)}$$
$$\leqslant \#\{i \in [M] : \mu_i(\boldsymbol{\Sigma}) \geqslant 1/(L_{\text{eff}}\gamma)\} + (L_{\text{eff}}\gamma) \cdot \sum_{i:\mu_i(\boldsymbol{\Sigma}) < 1/(L_{\text{eff}}\gamma)} \mu_i(\boldsymbol{\Sigma})$$
$$\lesssim (L_{\text{eff}}\gamma)^{1/a} + (L_{\text{eff}}\gamma) \cdot \sum_{i:i \gtrsim (L_{\text{eff}}\gamma)^{1/a}} \mu_i(\boldsymbol{\Sigma}) \lesssim (L_{\text{eff}}\gamma)^{1/a} + (L_{\text{eff}}\gamma) \cdot \sum_{i:i \gtrsim (L_{\text{eff}}\gamma)^{1/a}} i^{-a}$$
$$\lesssim (L_{\text{eff}}\gamma)^{1/a} \leqslant N/4,$$

where the last inequality follows since we may assume $L_{\text{eff}} \leqslant \tilde{c}N^a/\gamma$ for some constant $\tilde{c} > 0$ sufficiently small. Thus, condition (3) in Assumption 3 is satisfied.

To verify condition (4) in Assumption 3, we introduce $\mathbf{z}_1, \ldots \mathbf{z}_N \overset{iid}{\sim} \mathcal{N}(0, \mathbf{I}_M/N)$ and write $\mathbf{Z} = (\mathbf{z}_1, \ldots, \mathbf{z}_N)^\top$. It can be shown that $\mathbf{X}\mathbf{S}^\top/\sqrt{N} \overset{d}{=} \mathbf{Z}\boldsymbol{\Sigma}^{1/2}$ conditioned on $\mathbf{S}$. Therefore, we have $\text{tr}(\widehat{\boldsymbol{\Sigma}}) \overset{d}{=} \text{tr}(\mathbf{Z}\boldsymbol{\Sigma}\mathbf{Z}^\top)$, where $\mathbf{Z} \in \mathbb{R}^{N \times M}$ is a Gaussian sketching matrix. When $\mu_i(\boldsymbol{\Sigma}) \asymp i^{-a}$ for all

$i \in [M]$ (which happens with probability at least $1 - \exp(-\Omega(M))$ over $\mathbf{S}$ by Lemma E.5), we have by Hansen-Wright inequality (see e.g., exercise 2.17 in Wainwright (2019)) and a union bound that

$$N \max_{i \in [N]} \mathbf{Z}_{i,\cdot} \mathbf{\Sigma} \mathbf{Z}_{i,\cdot}^\top \lesssim \operatorname{tr}(\mathbf{\Sigma}) + \|\mathbf{\Sigma}\|_2 \cdot \log(N/\delta) + \|\mathbf{\Sigma}\|_F \cdot \sqrt{\log(N/\delta)} \lesssim 1 + \log(N/\delta).$$

with probability at least $1 - \delta$ over the randomness of $\bar{\mathbf{Z}}$. Thus, there exist some $a$-dependent constants $c, c' > 0$ such that when $\gamma \leqslant c/\log N$,

$$\mathbb{P}(\gamma_0 < \gamma/t) = \mathbb{P}(t/(4\gamma) < \max_i \|\mathbf{S}\mathbf{x}_i\|_2^2) = \mathbb{P}(t/(4\gamma) < N \max_i \mathbf{Z}_{i,\cdot} \mathbf{\Sigma} \mathbf{Z}_{i,\cdot}^\top) \leqslant N^{-c't}$$

for all $t \geqslant 1$. Therefore, condition (4) is also satisfied.

**The upper bound.** By Lemma E.5, we have $B_\mathbf{B}$ in Lemma B.1 satisfies

$$B_\mathbf{B} \leqslant c \cdot \left(1 + L_{\mathrm{eff}}^2 \cdot (N^{-2a} + N^{-2a} + N^{-2a} + N^{-2a})\right) \leqslant c \cdot (1 + N^{2a-2a}) \lesssim 1$$

with probability at least $1 - \exp(-\Omega(M))$, where the second inequality uses $L_{\mathrm{eff}} \lesssim N^a$. Moreover, we have by Lemma E.6 that $\frac{\mu_{M/2}(\mathbf{A}_k)}{\mu_M(\mathbf{A}_k)} \lesssim 1$ with probability at least $1 - \exp(-\Omega(M))$.

Now, choosing $k = \min\{M/3, (L_{\mathrm{eff}}\gamma)^{1/a}\}$ in Lemma B.1, using Assumption 1D, and taking expectation over $\mathbf{w}^*$ yields

$$\mathbb{E}_{\mathbf{w}*}[\mathsf{Bias}(\mathbf{w}^*)] \lesssim \frac{\max\{k^{a-b+1}, 1\}}{L_{\mathrm{eff}}\gamma} + k^{1-b} \lesssim \max\{(L_{\mathrm{eff}}\gamma)^{(1-b)/a}, M^{1-b}\}$$

with probability at least $1 - \exp(-\Omega(M))$ over the randomness of $\mathbf{S}$.

**The lower bound.** By Lemma B.2, we have

$$\mathbb{E}_{\mathbf{w}*}\mathsf{Bias}(\mathbf{w}^*) \gtrsim \sum_{i=2\bar{t}+1}^M \frac{\mu_{3i}(\mathbf{\Sigma}_\mathbf{w})}{\mu_i(\mathbf{\Sigma})},$$

with probability at least $1 - e^{-\Omega(M)}$, where $\bar{t} = \mathbb{E}_\mathbf{X}[\#\{i \in [M] : \widehat{\lambda}_i L_{\mathrm{eff}} \gamma_0 > 1/4\}]$ and $\{\widehat{\lambda}_i, i \in [M]\}$ are the eigenvalues of $\widehat{\mathbf{\Sigma}}$. Since $\widehat{\mathbf{\Sigma}} \overset{d}{=} \mathbf{Z}\mathbf{\Sigma}\mathbf{Z}^\top$ conditioned on $\mathbf{S}$ by Lemma E.3, when $\mu_i(\mathbf{\Sigma}) \asymp i^{-a}$ for all $i \in [M]$ (which happens with probability at least $1 - \exp(-\Omega(M))$ over $\mathbf{S}$ by Lemma E.5), we have by combining Lemma E.4 and E.3 with $k = N/c$ that

$$
\begin{aligned}
\widehat{\lambda}_{2j-1} &\overset{d}{=} \mu_{2j-1}(\mathbf{Z}\mathbf{\Sigma}\mathbf{Z}^\top) \\
&\leqslant \mu_j(\mathbf{Z}_{0:k}\mathbf{\Sigma}_{0:k}\mathbf{Z}_{0:k}^\top) + \mu_j(\mathbf{Z}_{k:\infty}\mathbf{\Sigma}_{k:\infty}\mathbf{Z}_{k:\infty}^\top) \\
&\lesssim \left(1 + \sqrt{\frac{k + \log(1/\delta)}{N}}\right) \cdot \mu_j(\mathbf{\Sigma}) + \left(N^{-a} + \frac{N + \log(1/\delta)}{N^{a+1}} + \frac{\sqrt{N^{2-2a} + N^{1-2a}\log(1/\delta)}}{N}\right) \\
&\lesssim \left(1 + \sqrt{\frac{\log(1/\delta)}{N}}\right) \cdot j^{-a} + N^{-a}\left(1 + \frac{\log(1/\delta)}{N} + \sqrt{\frac{\log(1/\delta)}{N}}\right) \quad (16)
\end{aligned}
$$

for all $j \leqslant k$ with probability at least $1 - \delta$ over the randomness of $\mathbf{Z}$. Therefore, it can be verified by a standard truncation argument that

$$\bar{t} = \mathbb{E}_\mathbf{X}[\#\{i \in [M] : \widehat{\lambda}_i L_{\mathrm{eff}} \gamma_0 > 1/4\}] \lesssim (L_{\mathrm{eff}}\gamma)^{1/a}.$$

Thus, when $(L_{\mathrm{eff}}\gamma_0)^{1/a} \leqslant M/c$ for some sufficiently large constant $c > 0$, we have

$$\mathbb{E}_{\mathbf{w}*}\mathsf{Bias}(\mathbf{w}^*) \gtrsim \sum_{i=2\bar{t}+1}^M \frac{\mu_{3i}(\mathbf{\Sigma}_\mathbf{w})}{\mu_i(\mathbf{\Sigma})} \gtrsim \sum_{i=c'(L_{\mathrm{eff}}\gamma)^{1/a}}^M \frac{\mu_{3i}(\mathbf{\Sigma}_\mathbf{w})}{\mu_i(\mathbf{\Sigma})} \gtrsim \sum_{i=c'(L_{\mathrm{eff}}\gamma)^{1/a}}^M \frac{i^{-a-b}}{i^{-a}} \gtrsim (L_{\mathrm{eff}}\gamma)^{(1-b)/a}$$

with probability at least $1 - \exp(-\Omega(M))$ over the randomness of $\mathbf{S}$, where the third inequality uses Lemma E.5 (with $\mathbf{H}$ replaced by $\mathbf{H}\mathbf{H}^\mathbf{w}\mathbf{H}$).

$\square$

# C Variance error

## C.1 Upper and lower bounds

**Lemma C.1** (An upper bound on the GD variance term). *Suppose Assumption 1A and 3 hold and $L_{\mathtt{eff}} \lesssim N^a/\gamma$. Under the notation in Theorem 3.1 and its proof, the variance term*

$$\mathsf{Var} := \mathbb{E}[\mathrm{tr}(\mathbf{X}\mathbf{S}^\top \mathbf{V}(\widehat{\boldsymbol{\Sigma}})\boldsymbol{\Sigma}\mathbf{V}(\widehat{\boldsymbol{\Sigma}})\mathbf{S}\mathbf{X}^\top)] \lesssim \frac{D^{\mathsf{U}}}{N}, \quad and \quad \mathsf{Var} \gtrsim \frac{D^{\mathsf{L}}}{N},$$

*where*

$$D^{\mathsf{U}} := \mathbb{E}_{\mathbf{X}}\Big[\#\{i \in [M] : \widehat{\lambda}_i L_{\mathtt{eff}}\gamma_0 > 1/4\} + (L_{\mathtt{eff}}\gamma_0) \sum_{i:\widehat{\lambda}_i L_{\mathtt{eff}}\gamma_0 \leqslant 1/4} \widehat{\lambda}_i\Big],$$

$$D^{\mathsf{L}} := \mathbb{E}_{\mathbf{X}}\Big[(L_{\mathtt{eff}}\gamma_0)^2 \cdot \sum_{i:\widehat{\lambda}_i L_{\mathtt{eff}}\gamma_0 \leqslant 1/4} \mu_i(\boldsymbol{\Sigma}) \cdot \mu_i(\widehat{\boldsymbol{\Sigma}}) + \frac{1}{5} \cdot \sum_{i:\widehat{\lambda}_i L_{\mathtt{eff}}\gamma_0 > 1/4} \frac{\mu_i(\boldsymbol{\Sigma})}{\mu_i(\widehat{\boldsymbol{\Sigma}})}\Big],$$

*and $(\widehat{\lambda}_i)_{i=1}^M$ are the eigenvalues of $\widehat{\boldsymbol{\Sigma}}$.*

*Proof of Lemma C.1.* Note that

$$\mathbf{V}(\widehat{\boldsymbol{\Sigma}}) = \frac{1}{N}\sum_{t=1}^{L}\gamma_t \cdot \prod_{i=t+1}^{L}(\mathbf{I} - \gamma_i\widehat{\boldsymbol{\Sigma}}) = \frac{(\mathbf{I} - \prod_{t=1}^{L}(\mathbf{I} - \gamma_t\widehat{\boldsymbol{\Sigma}}))\widehat{\boldsymbol{\Sigma}}^{-1}}{N}.$$

Adopt the shorthand $\mathsf{V}_L$ for $\mathbf{I} - \prod_{t=1}^{L}(\mathbf{I} - \gamma_t\widehat{\boldsymbol{\Sigma}})$. Reorganizing the terms, we have

$$\mathsf{Var} = N \cdot \mathbb{E}[\mathrm{tr}(\mathbf{V}(\widehat{\boldsymbol{\Sigma}})\boldsymbol{\Sigma}\mathbf{V}(\widehat{\boldsymbol{\Sigma}})\widehat{\boldsymbol{\Sigma}})] = \frac{1}{N} \cdot \mathbb{E}_{\mathbf{X}}[\mathrm{tr}(\boldsymbol{\Sigma}\mathsf{V}_L\widehat{\boldsymbol{\Sigma}}^{-1}\mathsf{V}_L)].$$

Let $\widehat{\lambda}_1, \dots, \widehat{\lambda}_M$ be the eigenvalues of $\widehat{\boldsymbol{\Sigma}}$ in non-increasing order, and let $\lambda > 0$ be some value which will be given later. We now derive an upper bound and a lower bound for the variance $\mathsf{Var}$.

**An upper bound.**  Continuing the calculation, we further have

$$\mathrm{tr}(\boldsymbol{\Sigma}\mathsf{V}_L\widehat{\boldsymbol{\Sigma}}^{-1}\mathsf{V}_L) = \mathrm{tr}(\mathsf{V}_L\widehat{\boldsymbol{\Sigma}}^{-1/2}(\widehat{\boldsymbol{\Sigma}} + \lambda\mathbf{I})^{1/2}[(\widehat{\boldsymbol{\Sigma}} + \lambda\mathbf{I})^{-1/2}\boldsymbol{\Sigma}(\widehat{\boldsymbol{\Sigma}} + \lambda\mathbf{I})^{-1/2}](\widehat{\boldsymbol{\Sigma}} + \lambda\mathbf{I})^{1/2}\widehat{\boldsymbol{\Sigma}}^{-1/2}\mathsf{V}_L)$$

$$\leqslant \|\boldsymbol{\Sigma}^{1/2}(\widehat{\boldsymbol{\Sigma}} + \lambda\mathbf{I})^{-1/2}\|_2^2 \cdot [\mathrm{tr}(\mathsf{V}_L^2 + \lambda\widehat{\boldsymbol{\Sigma}}^{-1}\mathsf{V}_L^2)].$$

Similar to the proof of claim (7b) in Lemma B.2, it can be verified that $\mathsf{V}_L \preceq \mathbf{I} - (\mathbf{I} - 2\gamma_0\widehat{\boldsymbol{\Sigma}})^{L_{\mathtt{eff}}}$ under the condition $\gamma_0 \leqslant 1/[4\max_i \|\mathbf{S}\mathbf{x}_i\|_2^2] \leqslant 1/[4\,\mathrm{tr}(\widehat{\boldsymbol{\Sigma}})]$ and stepsize assumption (2). Since $(1 - (1 - \gamma_0 x)^{L_{\mathtt{eff}}})^2 \leqslant \min\{(xL_{\mathtt{eff}}\gamma_0)^2, 1\}$ for $x \in [0, 1/(2\gamma_0)]$ by Bernoulli's inequality and $\sup_{[0,1/\gamma_0]}[(1 - (1 - \gamma_0 x)^{L_{\mathtt{eff}}})/x] = L_{\mathtt{eff}}\gamma_0$, it follows that

$$\mathrm{tr}(\mathsf{V}_L^2 + \lambda\widehat{\boldsymbol{\Sigma}}^{-1}\mathsf{V}_L^2) \leqslant \sum_{i=1}^{M}\Big[((1 - (1 - 2\gamma_0\widehat{\lambda}_i)^{L_{\mathtt{eff}}})^2 + \frac{\lambda(1 - (1 - 2\gamma_0\widehat{\lambda}_i)^{L_{\mathtt{eff}}})^2}{\widehat{\lambda}_i}\Big]$$

$$\lesssim \sum_{i=1}^{M}\Big[(1 + \lambda L_{\mathtt{eff}}\gamma_0) \cdot \mathbf{1}_{\{\widehat{\lambda}_i L_{\mathtt{eff}}\gamma_0 > 1/4\}}$$

$$+ (\lambda L_{\mathtt{eff}}\gamma_0(\widehat{\lambda}_i L_{\mathtt{eff}}\gamma_0) + (\widehat{\lambda}_i L_{\mathtt{eff}}\gamma_0)^2) \cdot \mathbf{1}_{\{\widehat{\lambda}_i L_{\mathtt{eff}}\gamma_0 \leqslant 1/4\}}\Big].$$

Choosing $\lambda = 1/(L_{\mathtt{eff}}\gamma) \leqslant 1/(L_{\mathtt{eff}}\gamma_0)$ yields

$$\mathrm{tr}(\mathsf{V}_L^2 + \lambda\widehat{\boldsymbol{\Sigma}}^{-1}\mathsf{V}_L^2)$$

$$\lesssim \#\{i \in [M] : \widehat{\lambda}_i L_{\mathtt{eff}}\gamma_0 > 1/4\} + (L_{\mathtt{eff}}\gamma_0)^2 \sum_{i:\widehat{\lambda}_i L_{\mathtt{eff}}\gamma_0 \leqslant 1/4} \widehat{\lambda}_i^2 + (L_{\mathtt{eff}}\gamma_0) \sum_{i:\widehat{\lambda}_i L_{\mathtt{eff}}\gamma_0 \leqslant 1/4} \widehat{\lambda}_i =: \widetilde{D}^{\mathsf{U}}$$

$$\lesssim \#\{i \in [M] : \widehat{\lambda}_i L_{\mathtt{eff}}\gamma_0 > 1/4\} + (L_{\mathtt{eff}}\gamma_0) \sum_{i:\widehat{\lambda}_i L_{\mathtt{eff}}\gamma_0 \leqslant 1/4} \widehat{\lambda}_i =: \widetilde{D}^{\mathsf{U}} \tag{17}$$

Applying Lemma E.1 and noting that $\operatorname{tr}(\mathsf{V}_L^2 + \lambda\widehat{\boldsymbol{\Sigma}}^{-1}\mathsf{V}_L^2) \lesssim N$ as $\mathsf{V}_L$ has at most $N$ non-zero eigenvalues, we obtain

$$\mathbb{E}_{\mathbf{X}}[\operatorname{tr}(\boldsymbol{\Sigma}\mathsf{V}_L\widehat{\boldsymbol{\Sigma}}^{-1}\mathsf{V}_L)] \lesssim \mathbb{E}_{\mathbf{X}}\Big[\#\{i \in [M] : \widehat{\lambda}_i L_{\mathtt{eff}}\gamma_0 > 1/4\} + (L_{\mathtt{eff}}\gamma_0) \sum_{i:\widehat{\lambda}_i L_{\mathtt{eff}}\gamma_0 \leqslant 1/4} \widehat{\lambda}_i\Big].$$

**A lower bound.**    Similarly, by Von-Neumann's trace inequality, we have

$$\operatorname{tr}(\boldsymbol{\Sigma}\mathsf{V}_L\widehat{\boldsymbol{\Sigma}}^{-1}\mathsf{V}_L) \geqslant \sum_{i=1}^{M} \mu_i(\boldsymbol{\Sigma})\mu_{M-i+1}(\mathsf{V}_L^2\widehat{\boldsymbol{\Sigma}}^{-1}) \overset{(i)}{\geqslant} \sum_{i=1}^{M} \mu_i(\boldsymbol{\Sigma})\frac{\mu_{2(M-i)+1}(\mathsf{V}_L^2\widehat{\boldsymbol{\Sigma}}^{-2})}{\mu_{M-i+1}(\widehat{\boldsymbol{\Sigma}}^{-1})}$$

$$\overset{(ii)}{=} \sum_{i=1}^{M} \mu_i(\boldsymbol{\Sigma}) \cdot \mu_i(\widehat{\boldsymbol{\Sigma}}) \cdot \mu_{2(M-i)+1}(\mathsf{V}_L^2\widehat{\boldsymbol{\Sigma}}^{-2}),$$

where step (i) uses $\mu_{i+j+1}(XY) \leqslant \mu_{i+1}(X)\mu_{j+1}(Y)$ for any matrices $X, Y$, and step (ii) uses the fact that $\mu_i(\widehat{\boldsymbol{\Sigma}}) = 1/\mu_{M-i+1}(\widehat{\boldsymbol{\Sigma}}^{-1})$.

Note that $\mathsf{V}_L \geqslant \mathbf{I} - (\mathbf{I} - \gamma_0\widehat{\boldsymbol{\Sigma}})^{L_{\mathtt{eff}}}$. Since $f(x) := (1 - (1 - \gamma_0 x)^{L_{\mathtt{eff}}})^2/x^2$ is a decreasing function on $[0, 1/\gamma_0]$ and (1) $f(x) \geqslant (L_{\mathtt{eff}}\gamma_0)^2/4$ when $L_{\mathtt{eff}}\gamma_0 x \leqslant 1/4$; (2) $f(x) \geqslant 1/(5x^2)$ when $L_{\mathtt{eff}}\gamma_0 x \geqslant 1/4$, we have

$$\operatorname{tr}(\boldsymbol{\Sigma}\mathsf{V}_L\widehat{\boldsymbol{\Sigma}}^{-1}\mathsf{V}_L) \geqslant \frac{(L_{\mathtt{eff}}\gamma_0)^2}{4} \cdot \sum_{i:\widehat{\lambda}_i L_{\mathtt{eff}}\gamma_0 \leqslant 1/4} \mu_i(\boldsymbol{\Sigma}) \cdot \mu_i(\widehat{\boldsymbol{\Sigma}}) + \frac{1}{5} \cdot \sum_{i:\widehat{\lambda}_i L_{\mathtt{eff}}\gamma_0 > 1/4} \frac{\mu_i(\boldsymbol{\Sigma})}{\mu_i(\widehat{\boldsymbol{\Sigma}})}.$$

Taking expectation over $\mathbf{X}$ yields the desired result.

$\square$

## C.2   Variance error under the source condition

**Lemma C.2** (Variance bounds under the source condition). *Let Assumption 1 hold and assume $L_{\mathtt{eff}} \lesssim N^a/\gamma$. Under the notation in Theorem 3.1 and its proof, there exists some $(a, b)$-dependent constant $c > 0$ such that when $\gamma \leqslant c/\log N$,*

$$\mathsf{Var} \asymp \frac{\min\{M, (L_{\mathtt{eff}}\gamma)^{1/a}\}}{N}$$

*with probability at least $1 - \exp(-\Omega(M))$ over the randomness of $\mathbf{S}$.*

*Proof of Lemma C.2.* Similar to the proof of Lemma B.3, we can verify that conditions (1)–(4) in Assumption 3 are satisfied with probability at least $1 - \exp(-\Omega(M))$ over the randomness of $\mathbf{S}$. From the expression of $D^{\mathsf{U}}$, it is straightforward to see that $D^{\mathsf{U}} \leqslant M$. Moreover, applying Lemma C.2, Eq. (16) in the proof of Lemma B.3 and a truncation argument, we can show that

$$D^{\mathsf{U}} = \mathbb{E}_{\mathbf{X}}\Big[\#\{i \in [M] : \widehat{\lambda}_i L_{\mathtt{eff}}\gamma_0 > 1/4\} + (L_{\mathtt{eff}}\gamma_0) \sum_{i:\widehat{\lambda}_i L_{\mathtt{eff}}\gamma_0 \leqslant 1/4} \widehat{\lambda}_i\Big]$$

$$\lesssim (L_{\mathtt{eff}}\gamma)^{1/a} + \mathbb{E}_{\mathbf{X}}\Big[(L_{\mathtt{eff}}\gamma) \cdot \sum_{i:i \gtrsim (L_{\mathtt{eff}}\gamma)^{1/a}} \widehat{\lambda}_i\Big] \lesssim (L_{\mathtt{eff}}\gamma)^{1/a}$$

with probability at least $1 - \exp(-\Omega(M))$ over the randomness of $\mathbf{S}$. Thus, we have obtained $D^{\mathsf{U}} \lesssim \min\{M, (L_{\mathtt{eff}}\gamma)^{1/a}\}$.

For the lower bound, when $(L_{\mathtt{eff}}\gamma)^{1/a} \leqslant M/c$ for some sufficiently large constant $c > 0$, conditioned on $\mathbf{S}$ such that $\mu_j(\boldsymbol{\Sigma}) \asymp j^{-a}$ for $j \in [M]$ (which holds with probability at least $1 - e^{-\Omega(M)}$ by Lemma E.5), we have by Lemma E.8 that $\mu_j(\widehat{\boldsymbol{\Sigma}}) \asymp j^{-a}$ for $j \leqslant \min\{M, N\}/\tilde{c}$ with probability at least $1 - e^{-\Omega(M)}$ for some $\tilde{c} > 0$. Therefore,

$$D^{\mathsf{L}} \geqslant \mathbb{E}_{\mathbf{X}}\Bigg[(L_{\mathtt{eff}}\gamma_0)^2 \cdot \sum_{i:\widehat{\lambda}_i L_{\mathtt{eff}}\gamma_0 \leqslant 1/4} \mu_i(\boldsymbol{\Sigma}) \cdot \mu_i(\widehat{\boldsymbol{\Sigma}})\Bigg]$$

$$\gtrsim (L_{\mathsf{eff}}\gamma)^2 \cdot \sum_{i:i \gtrsim (L_{\mathsf{eff}}\gamma)^{1/a}, i \leqslant \min\{M,N\}/\tilde{c}} i^{-a} \cdot i^{-a} \gtrsim (L_{\mathsf{eff}}\gamma)^{1/a},$$

where the last line follows since we assume $(L_{\mathsf{eff}}\gamma)^{1/a} \leqslant M/c$ for some sufficiently large constant $c > 0$ and $(L_{\mathsf{eff}}\gamma)^{1/a} \lesssim N \lesssim N/c$.

On the other hand, similarly, when $(L_{\mathsf{eff}}\gamma)^{1/a} \geqslant M/c$ for some sufficiently constant $c > 0$, conditioned on $\mathbf{S}$ such that $\mu_j(\mathbf{\Sigma}) \asymp j^{-a}$ for $j \in [M]$ (which holds with probability at least $1 - e^{-\Omega(M)}$ by Lemma E.5), we have by Lemma E.8 that $\mu_j(\widehat{\mathbf{\Sigma}}) \asymp j^{-a}$ for $j \leqslant M$ with probability at least $1/2$. Therefore,

$$D^{\mathsf{L}} \geqslant \mathbb{E}_{\mathbf{X}}\left[\frac{1}{5} \cdot \sum_{i:\hat{\lambda}_i L_{\mathsf{eff}}\gamma_0 > 1/4} \frac{\mu_i(\mathbf{\Sigma})}{\mu_i(\widehat{\mathbf{\Sigma}})}\right] \gtrsim \mathbb{E}_{\mathbf{X}}\left[\frac{1}{5} \cdot \sum_{i:i \leqslant M/c} \frac{\mu_i(\mathbf{\Sigma})}{\mu_i(\widehat{\mathbf{\Sigma}})}\right] \gtrsim M.$$

Putting pieces together yields the desired lower bound.

$\square$

# D  Fluctuation error

## D.1  An upper bound

**Lemma D.1** (An upper bound on the fluctuation error). *For each $i \in [N]$, define the leave-one-out GD process*

$$\boldsymbol{\theta}_t^{(-i)} = (\mathbf{I} - \gamma_t \widehat{\mathbf{\Sigma}}^{(-i)})\boldsymbol{\theta}_{t-1}^{(-i)} + \gamma_t (\mathbf{S}\mathbf{X}^{\top}\mathbf{y})^{(-i)}, \quad \text{with } \boldsymbol{\theta}_0^{(-i)} = \mathbf{0}, \tag{LOO-GD}$$

*where $\widehat{\mathbf{\Sigma}}^{(-i)} := \sum_{j \neq i} \mathbf{S}\mathbf{x}_j \mathbf{x}_j^{\top} \mathbf{S}^{\top}/N$ and $(\mathbf{S}\mathbf{X}^{\top}\mathbf{y})^{(-i)} := \sum_{j \neq i} \mathbf{S}\mathbf{x}_j y_j / N$.*

*Let Assumption 1A, 1B, 3 hold and assume $L_{\mathsf{eff}} \lesssim N^a/\gamma$. Under the notation in Theorem 3.1 and its proof, for any $s \in [0,1], \alpha > 1$, there exists some $(s,\alpha)$-dependent constant $c > 0$ such that the fluctuation error satisfies*

$$\mathbb{E}[\mathsf{Fluc}] = \mathbb{E}_{\mathbf{w}*,(\mathbf{x}_i,y_i)_{i=1}^N, i_t, t \in [L]}\left[\|\mathbf{\Sigma}^{1/2}(\mathbf{v}_L - \boldsymbol{\theta}_L)\|_2^2\right]$$
$$\leqslant c \cdot \mathbb{E}[\mathsf{F} \cdot \mathrm{tr}(\widehat{\mathbf{\Sigma}}^{1/\alpha})] \cdot \gamma^{1/\alpha} L_{\mathsf{eff}}^{1/\alpha - 1},$$

*with probability at least $1 - \exp(-\Omega(M))$ over the randomness of $\mathbf{S}$, where*

$$\mathsf{F} := \widehat{R}\left(\max_{i \in [N]}(\mathbf{x}_i^{\top}\mathbf{S}^{\top}\mathbf{v}*)^2 + \max_{i \in [N]}\tilde{\epsilon}_i^2 + \max_{i \in [N], t \in [L]}(\mathbf{x}_i^{\top}\mathbf{S}^{\top}\boldsymbol{\theta}_t^{(-i)})^2 + \max_{i \in [N]}\|\mathbf{x}_i^{\top}\mathbf{S}^{\top}\|_{\mathbf{\Sigma}^{-s}}^2 \cdot \mathbf{B}_{\mathbf{\Delta}}\right),$$

$$\mathbf{B}_{\mathbf{\Delta}} := a_{\max}^2 \cdot \max_{i \in [N]} \|\mathbf{S}\mathbf{x}_i\|_2^2 \cdot \widehat{R} \cdot \frac{(L_{\mathsf{eff}}\gamma)^{2-s}}{N^2}, \quad \text{and}$$

$$a_{\max} := \max_{i \in [N], t \in [L]} |y_i + \mathbf{x}_i^{\top}\mathbf{S}^{\top}\boldsymbol{\theta}_t^{(-i)}|, \quad \lambda := \frac{1}{L_{\mathsf{eff}}\gamma}, \quad \widehat{R} := \|(\widehat{\mathbf{\Sigma}} + \lambda\mathbf{I})^{-1/2}(\mathbf{\Sigma} + \lambda\mathbf{I})^{1/2}\|_2^2.$$

*Moreover, if $\mu_j(\widehat{\mathbf{\Sigma}}) \asymp j^{-a}$ for $j \leqslant r(\widehat{\mathbf{\Sigma}})$ for some $a > 1$, then*

$$\mathbb{E}_{i_t, t \in [L]}\|\mathbf{\Sigma}^{1/2}(\mathbf{v}_L - \boldsymbol{\theta}_L)\|_2^2 \leqslant c'\mathbb{E}[\mathsf{F}] \cdot \gamma^{1/a} L_{\mathsf{eff}}^{1/a - 1}$$

*for some $a$-dependent constant $c' > 0$.*

*Proof of Lemma D.1.* The proof of this lemma follows from similar ideas as in the proof of Lemma 5 in Pillaud-Vivien et al. (2018), but with a more precise characterization on the magnitude of GD outputs. We start with an overview of the proof. At a high level, to bound the fluctuation error, we express the difference between the multi-pass SGD and GD trajectories, $\mathbf{v}_t - \boldsymbol{\theta}_t$, as a stochastic process (Eq. 18) that fits into the framework of Lemma D.2, which provides an upper bound on the fluctuation error $\mathbb{E}[\|\mathbf{\Sigma}^{1/2}(\mathbf{v}_t - \boldsymbol{\theta}_t)\|^2]$ under certain conditions, up to a mismatch between $\widehat{\mathbf{\Sigma}}$ and $\mathbf{\Sigma}$. We verify that the required conditions hold with appropriate choices of parameters (Eq. 20), which are further bounded using a leave-one-out argument (Lemma D.3). Applying Lemma D.2 with these parameters and a covariance replacement trick (Eq. 21) yields the desired bounds.

We now proceed to the proof. Define $\boldsymbol{\Delta}_t := \mathbf{v}_t - \boldsymbol{\theta}_t$ for $t \in [0, L]$. Recall that $\mathbf{v}^* = (\mathbf{SHS}^\top)^{-1}\mathbf{SHw}^*$ and we have $y_i = (\mathbf{Sx}_i)^\top \mathbf{v}^* + \tilde{\epsilon}_i$ for all $i \in [N]$ with $\tilde{\epsilon}_i$ independent of $\mathbf{Sx}_i$ conditioned on $(\mathbf{S}, \mathbf{w}^*)$ under the Gaussian assumption in Assumption 1A. Moreover, $\mathbb{E}[\tilde{\epsilon}_i | \mathbf{S}, \mathbf{w}^*] = 0$ and $\mathbb{E}[\tilde{\epsilon}_i^2 | \mathbf{S}, \mathbf{w}^*] \leqslant \sigma^2 + \|\mathbf{w}^*\|_\mathbf{H}^2 =: \tilde{\sigma}^2(\mathbf{w}^*)$. By the definition of $\mathbf{v}_t$ and $\boldsymbol{\theta}_t$ in (multi-pass SGD) and (GD), we have

$$\boldsymbol{\Delta}_t = (\mathbf{I} - \gamma_t \mathbf{Sx}_{i_t}\mathbf{x}_{i_t}^\top\mathbf{S}^\top)\boldsymbol{\Delta}_{t-1} + \gamma_t \cdot (\boldsymbol{\xi}_{1,t} + \boldsymbol{\xi}_{2,t}), \tag{18}$$

where

$$\boldsymbol{\Delta}_0 = \mathbf{0}, \ \ \boldsymbol{\xi}_{1,t} := -\Big[\mathbf{Sx}_{i_t}\mathbf{x}_{i_t}^\top\mathbf{S}^\top - \widehat{\boldsymbol{\Sigma}}\Big](\boldsymbol{\theta}_{t-1} - \mathbf{v}^*), \ \ \text{and} \ \ \boldsymbol{\xi}_{2,t} := \mathbf{Sx}_{i_t}\tilde{\epsilon}_{i_t} - \mathbf{SX}^\top\tilde{\epsilon}/N, \ \ t \in [L].$$

Note that conditioned on $\mathbf{w}^*$, $\mathbf{S}$ and the dataset $\mathcal{D} = (\mathbf{x}_i, y_i)_{i=1}^N$, the noise terms $\mathbb{E}[\boldsymbol{\xi}_{1,t} | \mathbf{S}, \mathbf{w}^*, \mathcal{D}] = \mathbb{E}[\boldsymbol{\xi}_{2,t} | \mathbf{S}, \mathbf{w}^*, \mathcal{D}] = 0$. Next, we present the following two results.

**Lemma D.2** (A modified Proposition 1 of Pillaud-Vivien et al. (2018) for the last iterate). *Consider any recursion of the form*

$$\boldsymbol{\mu}_t = (\mathbf{I} - \gamma_t \cdot \boldsymbol{\nu}_t\boldsymbol{\nu}_t^\top)\boldsymbol{\mu}_{t-1} + \gamma_t \cdot \boldsymbol{\xi}_t, \quad \boldsymbol{\mu}_0 = \mathbf{0}, \quad t \in [L], \tag{19}$$

*where the learning rates $(\gamma_t)_{t=1}^L$ are as defined in Theorem 3.1 and Eq. (2), $(\boldsymbol{\nu}_t, \boldsymbol{\xi}_t)_{t=1}^L \in \mathbb{R}^M \times \mathbb{R}^M$ are independent random vectors. Assume that $\mathbb{E}[\boldsymbol{\nu}_t\boldsymbol{\nu}_t^\top] = \boldsymbol{\Sigma}_\nu$, $\mathbb{E}[\boldsymbol{\xi}_t] = \mathbf{0}$, $\mathbb{E}[\boldsymbol{\nu}_t\boldsymbol{\nu}_t^\top\boldsymbol{\nu}_t\boldsymbol{\nu}_t^\top] \preceq B_\nu^2\boldsymbol{\Sigma}_\nu$, $\mathbb{E}[\boldsymbol{\xi}_t\boldsymbol{\xi}_t^\top] \preceq \sigma_\xi^2\boldsymbol{\Sigma}_\nu$, and $\gamma_0 B_\nu^2 \leqslant 1/4$. Then for any $u \in [0, 1]$, we have*

$$\mathbb{E}[\|\boldsymbol{\Sigma}_\nu^{u/2}\boldsymbol{\mu}_L\|_2^2] \leqslant c\sigma_\xi^2 \cdot \gamma \operatorname{tr}(\boldsymbol{\Sigma}_\nu^{1/\alpha})(L_{\mathrm{eff}}\gamma)^{1/\alpha - u}$$

*fot any $\alpha > 1$ and some $\alpha$-dependent constant $c > 0$. Moreover, there exists some $a$-dependent constant $c', \tilde{c} > 1$ such that when $\mu_j(\boldsymbol{\Sigma}_\nu) \asymp j^{-a}$ for $j \leqslant \min\{M, N/\tilde{c}\}$, we have*

$$\mathbb{E}[\|\boldsymbol{\Sigma}_\nu^{u/2}\boldsymbol{\mu}_L\|_2^2] \leqslant c'\sigma_\xi^2 \cdot \gamma(L_{\mathrm{eff}}\gamma)^{1/a - u}$$

*for any $u \in [0, 1]$ and some $a$-dependent constant $c' > 0$.*

See the proof of Lemma D.2 in Section D.4.

**Lemma D.3** (A leave-one-out bound on GD iterates). *Under the assumptions and notation in Lemma D.1, for any $s \in [0, 1]$, there exists some $s$-dependent constant $c > 0$ such that the (GD) updates $(\boldsymbol{\theta}_t)_{t=1}^L$ satisfies*

$$\max_{i \in [N], t \in [L]} (\mathbf{x}_i^\top\mathbf{S}^\top\boldsymbol{\theta}_t)^2 \leqslant c \cdot \Big[\max_{i \in [N], t \in [L]} (\mathbf{x}_i^\top\mathbf{S}^\top\boldsymbol{\theta}_t^{(-i)})^2 + \max_{i \in [N]} \|\mathbf{x}_i^\top\mathbf{S}^\top\|_{\boldsymbol{\Sigma}^{-s}}^2 \cdot \mathbf{B}_\Delta\Big].$$

See the proof of Lemma D.3 in Section D.5.

Let $\boldsymbol{\nu}_t = \mathbf{Sx}_{i_t}$, $\boldsymbol{\xi}_t = \boldsymbol{\xi}_{1,t} + \boldsymbol{\xi}_{2,t}$. We claim that $(\boldsymbol{\nu}_t, \boldsymbol{\xi}_t)$ satisfies the conditions in Lemma D.2 with

$$\boldsymbol{\Sigma}_\nu = \widehat{\boldsymbol{\Sigma}}, \ \ B_\nu = \max_{i \in [N]} \|\mathbf{Sx}_i\|_2, \ \ \sigma_\xi^2 = 2 \max_{i \in [N], t \in [L]} [(\mathbf{x}_i^\top\mathbf{S}^\top(\boldsymbol{\theta}_{t-1} - \mathbf{v}^*))^2 + \tilde{\epsilon}_i^2]. \tag{20}$$

Thus applying Lemma D.2 with $u = 0, 1$ to the stochastic process in (18) and letting $\lambda = \frac{1}{L_{\mathrm{eff}}\gamma}$ yields

$$\mathbb{E}_{i_t, t \in [L]}\|\boldsymbol{\Sigma}^{1/2}(\mathbf{v}_L - \boldsymbol{\theta}_L)\|_2^2 \lesssim \|(\widehat{\boldsymbol{\Sigma}} + \lambda\mathbf{I})^{-1/2}(\boldsymbol{\Sigma} + \lambda\mathbf{I})^{1/2}\|^2 \cdot \mathbb{E}_{i_t, t \in [L]}\|(\widehat{\boldsymbol{\Sigma}} + \lambda\mathbf{I})^{1/2}(\mathbf{v}_L - \boldsymbol{\theta}_L)\|_2^2 \tag{21}$$

$$\lesssim \sigma_\xi^2 \cdot \|(\widehat{\boldsymbol{\Sigma}} + \lambda\mathbf{I})^{-1/2}(\boldsymbol{\Sigma} + \lambda\mathbf{I})^{1/2}\|^2 \cdot \gamma \operatorname{tr}(\widehat{\boldsymbol{\Sigma}}^{1/\alpha})(L_{\mathrm{eff}}\gamma)^{1/\alpha - 1}.$$

Moreover,

$$\sigma_\xi^2 \lesssim \max_{i \in [N]}(\mathbf{x}_i^\top\mathbf{S}^\top\mathbf{v}^*)^2 + \max_{i \in [N]} \tilde{\epsilon}_i^2 + \max_{i \in [N], t \in [L]}(\mathbf{x}_i^\top\mathbf{S}^\top\boldsymbol{\theta}_t)^2$$

$$\lesssim \max_{i \in [N]}(\mathbf{x}_i^\top\mathbf{S}^\top\mathbf{v}^*)^2 + \max_{i \in [N]} \tilde{\epsilon}_i^2 + \max_{i \in [N], t \in [L]}(\mathbf{x}_i^\top\mathbf{S}^\top\boldsymbol{\theta}_t^{(-i)})^2 + \max_{i \in [N]} \|\mathbf{x}_i^\top\mathbf{S}^\top\|_{\boldsymbol{\Sigma}^{-s}}^2 \cdot \mathbf{B}_\Delta,$$

where the second line follows from Lemma D.3. Putting the last two displays together and taking expectation over $(\mathbf{x}_i, y_i)_{i=1}^N$, $\mathbf{w}^*$ yields the first part of Lemma D.1. The second part of Lemma D.1 follows from the same argument by applying the second part of Lemma D.2.

**Proof of claim** (20). Conditioned on $\mathbf{S}$ and $(\mathbf{x}_i, y_i)_{i=1}^N$, when choosing $\boldsymbol{\nu}_t = \mathbf{S}\mathbf{x}_{i_t}$, we have $\mathbb{E}[\boldsymbol{\nu}_t \boldsymbol{\nu}_t^\top] = \mathbb{E}[\mathbf{S}\mathbf{x}_{i_t}\mathbf{x}_{i_t}^\top \mathbf{S}^\top] = \widehat{\boldsymbol{\Sigma}}$, and $\mathbb{E}[\boldsymbol{\nu}_t \boldsymbol{\nu}_t^\top \boldsymbol{\nu}_t \boldsymbol{\nu}_t^\top] \leq \mathbb{E}[\max_{i \in [N]} \|\boldsymbol{\nu}_i\|^2 \boldsymbol{\nu}_i \boldsymbol{\nu}_i^\top] \leq \max_{i \in [N]} \|\mathbf{S}\mathbf{x}_i\|_2^2 \cdot \widehat{\boldsymbol{\Sigma}} = B_{\boldsymbol{\nu}}^2 \cdot \widehat{\boldsymbol{\Sigma}}$. Thus we may let $\boldsymbol{\Sigma}_{\boldsymbol{\nu}} = \widehat{\boldsymbol{\Sigma}}$ and $B_{\boldsymbol{\nu}} = \max_{i \in [N]} \|\mathbf{S}\mathbf{x}_i\|_2$. In this case, we have $\gamma_0 B_{\boldsymbol{\nu}}^2 \leq 1/4$ by the assumption that $\gamma_0 \leq 1/[4\max_{i \in [N]} \|\mathbf{S}\mathbf{x}_i\|_2^2]$. It remains to bound $\sigma_\xi^2$ in (20). Note that

$$
\begin{aligned}
\mathbb{E}[\boldsymbol{\xi}_t \boldsymbol{\xi}_t^\top] &\leq 2\mathbb{E}[\boldsymbol{\xi}_{1,t}\boldsymbol{\xi}_{1,t}^\top] + 2\mathbb{E}[\boldsymbol{\xi}_{2,t}\boldsymbol{\xi}_{2,t}^\top] \\
&\leq 2\mathbb{E}[\boldsymbol{\nu}_i \boldsymbol{\nu}_i^\top (\boldsymbol{\theta}_{t-1} - \mathbf{v}^*)(\boldsymbol{\theta}_{t-1} - \mathbf{v}^*)^\top \boldsymbol{\nu}_i \boldsymbol{\nu}_i^\top] + 2\mathbb{E}[\mathbf{S}\mathbf{x}_{i_t}^\top \tilde{\epsilon}_{i_t}(\mathbf{S}\mathbf{x}_{i_t}^\top \tilde{\epsilon}_{i_t})^\top] \\
&\leq 2\max_{i \in [N]}(\mathbf{x}_i^\top \mathbf{S}^\top (\boldsymbol{\theta}_{t-1} - \mathbf{v}^*))^2 \cdot \boldsymbol{\Sigma}_{\boldsymbol{\nu}} + 2\max_{i \in [N]} \tilde{\epsilon}_i^2 \cdot \boldsymbol{\Sigma}_{\boldsymbol{\nu}},
\end{aligned}
$$

where the second line uses Jensen's inequality. Therefore, we can set

$$
\sigma_\xi^2 = 2\max_{i \in [N], t \in [L]}[(\mathbf{x}_i^\top \mathbf{S}^\top (\boldsymbol{\theta}_{t-1} - \mathbf{v}^*))^2 + \tilde{\epsilon}_i^2]
$$

and the conditions required by Lemma D.2 are satisfied.

$\square$

## D.2 A lower bound

**Lemma D.4** (A lower bound on the fluctuation error). *Let Assumption 1A, 1B, 3 hold and assume $L_{\mathrm{eff}} \lesssim N^a/\gamma$. Under the notation in Theorem 3.1 and its proof, with probability at least $1 - \exp(-\Omega(M))$ over the randomness of $\mathbf{S}$,*

$$
\begin{aligned}
\mathbb{E}_{(\mathbf{x}_i, y_i)_{i \in [N]}, i_t, t \in [L]}[\mathsf{Fluc}] &= \mathbb{E}_{(\mathbf{x}_i, y_i)_{i \in [N]}, i_t, t \in [L]}[\|\boldsymbol{\Sigma}^{1/2}(\mathbf{v}_L - \boldsymbol{\theta}_L)\|_2^2] \\
&\gtrsim (\sigma^2 + \|\mathbf{w}^*\|_{\mathbf{H}}^2) \cdot \mathbb{E}_{(\mathbf{x}_i)_{i \in [N]}}\Big[\frac{B_{\boldsymbol{\xi}}\gamma_0 L_{\mathrm{eff}}\gamma_0}{10} \cdot \sum_{i > \tilde{t}} \mu_i(\widehat{\boldsymbol{\Sigma}}) \cdot \mu_i(\boldsymbol{\Sigma})\Big],
\end{aligned}
$$

*where $B_{\boldsymbol{\xi}} := \max\{\frac{N-1}{N} - \frac{4\gamma_0 L_{\mathrm{eff}}}{N}B_{\boldsymbol{\nu}}^2, 0\}$, $B_{\boldsymbol{\nu}} := \max_{i \in [N]} \|\mathbf{S}\mathbf{x}_i\|_2$ and $\tilde{t} := \#\{i \in [M] : \widehat{\lambda}_i L_{\mathrm{eff}}\gamma_0 > 1/8\}$, and $(\widehat{\lambda}_i)_{i=1}^M$ are the eigenvalues of $\widehat{\boldsymbol{\Sigma}}$.*

*Proof of Lemma D.4.* Define $\boldsymbol{\Delta}_t := \mathbf{v}_t - \boldsymbol{\theta}_t$ for $t \in [L]$. Similar to the proof of Lemma D.1, conditioned on $\mathbf{S}$ and $\mathbf{w}^*$, we have

$$
\begin{aligned}
\boldsymbol{\Delta}_t &= (\mathbf{I} - \gamma_t \mathbf{S}\mathbf{x}_{i_t}\mathbf{x}_{i_t}^\top \mathbf{S}^\top)\boldsymbol{\Delta}_{t-1} + \gamma_t \cdot (\boldsymbol{\xi}_{1,t} + \boldsymbol{\xi}_{2,t}) \\
&= \sum_{i=1}^L \gamma_i \cdot \prod_{j=i+1}^L (\mathbf{I} - \gamma_j \mathbf{S}\mathbf{x}_{i_j}\mathbf{x}_{i_j}^\top \mathbf{S}^\top)^\top (\boldsymbol{\xi}_{1,i} + \boldsymbol{\xi}_{2,i}) \quad (22)
\end{aligned}
$$

where

$$
\boldsymbol{\Delta}_0 = \mathbf{0}, \quad \boldsymbol{\xi}_{1,t} := -\Big[\mathbf{S}\mathbf{x}_{i_t}\mathbf{x}_{i_t}^\top \mathbf{S}^\top - \widehat{\boldsymbol{\Sigma}}\Big](\boldsymbol{\theta}_{t-1} - \mathbf{v}^*), \quad \text{and} \quad \boldsymbol{\xi}_{2,t} := \mathbf{S}\mathbf{x}_{i_t}\tilde{\epsilon}_{i_t} - \mathbf{S}\mathbf{X}^\top \tilde{\boldsymbol{\epsilon}}/N, \ t \in [L],
$$

and $\tilde{\epsilon}_i$ are i.i.d $\mathcal{N}(0, \tilde{\sigma}^2(\mathbf{w}^*))$ independent of $\mathbf{S}\mathbf{x}_i$ conditioned on $\mathbf{S}$ and $\mathbf{w}^*$, where $\tilde{\sigma}^2(\mathbf{w}^*) := \sigma^2 + \|\mathbf{w}^*\|_{\mathbf{H}}^2$. Let $B_{\boldsymbol{\nu}} := \max_{i \in [N]} \|\mathbf{S}\mathbf{x}_i\|_2$. We claim that

$$
\mathbb{E}_{(\tilde{\epsilon}_i)_{i \in [N]}}\mathbb{E}_{i_t, t \in [L]}[(\boldsymbol{\xi}_{1,i} + \boldsymbol{\xi}_{2,i})(\boldsymbol{\xi}_{1,i} + \boldsymbol{\xi}_{2,i})^\top] \geq \tilde{\sigma}^2(\mathbf{w}^*)B_{\boldsymbol{\xi}} \cdot \widehat{\boldsymbol{\Sigma}}, \quad (23)
$$

and we have $B_{\boldsymbol{\xi}} \geq 1/2$ when $(L_{\mathrm{eff}}\gamma)B_{\boldsymbol{\nu}}^2/N \leq 1/3$. The proof of this claim is deferred to the end of the proof.

Since $\boldsymbol{\xi}_{1,t}$ and $\boldsymbol{\xi}_{2,t}$ are zero-mean noise, conditioned on $\mathbf{S}$, $\mathbf{w}^*$ and $(\mathbf{x}_i, y_i)_{i=1}^N$, we have

$$
\begin{aligned}
&\mathbb{E}_{i_t, t \in [L]}\mathbb{E}_{(\tilde{\epsilon}_k)_{k \in [N]}}[\|\boldsymbol{\Sigma}^{1/2}(\mathbf{v}_L - \boldsymbol{\theta}_L)\|_2^2] \\
&= \sum_{i=1}^L \gamma_i^2 \cdot \mathbb{E}_{(\tilde{\epsilon}_k)_{k \in [N]}}\mathbb{E}_{i_t, t \in [L]}[\mathrm{tr}(\prod_{j=i+1}^L (\mathbf{I} - \gamma_j \mathbf{S}\mathbf{x}_{i_j}\mathbf{x}_{i_j}^\top \mathbf{S}^\top)\boldsymbol{\Sigma}\prod_{j=i+1}^L (\mathbf{I} - \gamma_j \mathbf{S}\mathbf{x}_{i_j}\mathbf{x}_{i_j}^\top \mathbf{S}^\top)^\top (\boldsymbol{\xi}_{1,i} + \boldsymbol{\xi}_{2,i})(\boldsymbol{\xi}_{1,i} + \boldsymbol{\xi}_{2,i})^\top)]
\end{aligned}
$$

$$\geq \tilde{\sigma}^2(\mathbf{w}^*)B_{\boldsymbol{\xi}} \cdot \sum_{i=1}^{L} \gamma_i^2 \mathbb{E}_{i_t, t \in [L]}[\mathrm{tr}(\prod_{j=i+1}^{L}(\mathbf{I} - \gamma_j \mathbf{S}\mathbf{x}_{i_j}\mathbf{x}_{i_j}^\top \mathbf{S}^\top)\boldsymbol{\Sigma} \prod_{j=i+1}^{L}(\mathbf{I} - \gamma_j \mathbf{S}\mathbf{x}_{i_j}\mathbf{x}_{i_j}^\top \mathbf{S}^\top)^\top \widehat{\boldsymbol{\Sigma}})]$$

$$\geq \tilde{\sigma}^2(\mathbf{w}^*)B_{\boldsymbol{\xi}} \cdot \sum_{i=1}^{L} \gamma_i^2 \, \mathrm{tr}(\boldsymbol{\Sigma}\widehat{\boldsymbol{\Sigma}} \prod_{j=i+1}^{L}(\mathbf{I} - 2\gamma_j \widehat{\boldsymbol{\Sigma}})),$$

where the last line follows from the fact that $\mathbb{E}_{i_j}[(\mathbf{I} - \gamma_j \mathbf{S}\mathbf{x}_{i_j}\mathbf{x}_{i_j}^\top \mathbf{S}^\top)P(\widehat{\boldsymbol{\Sigma}})(\mathbf{I} - \gamma_j \mathbf{S}\mathbf{x}_{i_j}\mathbf{x}_{i_j}^\top \mathbf{S}^\top)] \geq P(\widehat{\boldsymbol{\Sigma}})(\mathbf{I} - 2\gamma_j \widehat{\boldsymbol{\Sigma}})$ for any polynomial $P$. Continuing from the last line, we have

$$\mathbb{E}_{i_t, t \in [L]}\mathbb{E}_{(\tilde{\boldsymbol{\epsilon}}_k)_{k \in [N]}}[\|\boldsymbol{\Sigma}^{1/2}(\mathbf{v}_L - \boldsymbol{\theta}_L)\|_2^2]$$

$$\geq \tilde{\sigma}^2(\mathbf{w}^*)B_{\boldsymbol{\xi}} \cdot \mathrm{tr}\left(\widehat{\boldsymbol{\Sigma}} \sum_{i=1}^{L} \gamma_i^2 \prod_{j=i+1}^{L}(\mathbf{I} - 2\gamma_j \widehat{\boldsymbol{\Sigma}})\boldsymbol{\Sigma}\right)$$

$$= \tilde{\sigma}^2(\mathbf{w}^*)B_{\boldsymbol{\xi}} \cdot \sum_{k=0}^{\lfloor \log L - 1 \rfloor} \gamma_{L_{\mathrm{eff}}k+1}^2 \cdot \mathrm{tr}\left(\widehat{\boldsymbol{\Sigma}}\frac{\mathbf{I} - (\mathbf{I} - 2\gamma_{L_{\mathrm{eff}}k+1}\widehat{\boldsymbol{\Sigma}})^{L_{\mathrm{eff}}}}{2\gamma_{L_{\mathrm{eff}}k+1}\widehat{\boldsymbol{\Sigma}}} \prod_{j=k+1}^{\lfloor \log L - 1 \rfloor}(\mathbf{I} - 2\gamma_{L_{\mathrm{eff}}j+1}\widehat{\boldsymbol{\Sigma}})^{L_{\mathrm{eff}}} \boldsymbol{\Sigma}\right)$$

$$\geq \tilde{\sigma}^2(\mathbf{w}^*)B_{\boldsymbol{\xi}} \cdot \gamma_0 \, \mathrm{tr}\left((\mathbf{I} - (\mathbf{I} - 2\gamma_0\widehat{\boldsymbol{\Sigma}})^{L_{\mathrm{eff}}})(\mathbf{I} - 4\gamma_0\widehat{\boldsymbol{\Sigma}})^{L_{\mathrm{eff}}} \boldsymbol{\Sigma}\right),$$

where the last line uses $\prod_{j=0}^{\lfloor \log L - 1 \rfloor}(1 - 2\gamma_{L_{\mathrm{eff}}j+1}x)^{L_{\mathrm{eff}}} \geqslant (1 - 4\gamma_0 x)^{L_{\mathrm{eff}}}$ for $x \in [0, 1/(2\gamma_0)]$ by the stepsize definition (2). Since $\mu_{i+j+1}(XY) \leqslant \mu_{i+1}(X)\mu_{j+1}(Y)$ for any $i, j$ and matrices $X, Y$ of matching dimensions, it follows that for any $j \geqslant 1$

$$\mu_j((\mathbf{I} - (\mathbf{I} - 2\gamma_0\widehat{\boldsymbol{\Sigma}})^{L_{\mathrm{eff}}})(\mathbf{I} - 4\gamma_0\widehat{\boldsymbol{\Sigma}})^{L_{\mathrm{eff}}} \boldsymbol{\Sigma})$$

$$\geqslant \frac{\mu_{3j-2}((\mathbf{I} - (\mathbf{I} - 2\gamma_0\widehat{\boldsymbol{\Sigma}})^{L_{\mathrm{eff}}})(\mathbf{I} - 4\gamma_0\widehat{\boldsymbol{\Sigma}})^{L_{\mathrm{eff}}}/\widehat{\boldsymbol{\Sigma}})}{\mu_j(\widehat{\boldsymbol{\Sigma}}^{-1})\mu_j(\boldsymbol{\Sigma}^{-1})}$$

$$= \mu_{M-j}(\widehat{\boldsymbol{\Sigma}}) \cdot \mu_{M-j}(\boldsymbol{\Sigma}) \cdot \mu_{3j-2}\left(\frac{(\mathbf{I} - (\mathbf{I} - 2\gamma_0\widehat{\boldsymbol{\Sigma}})^{L_{\mathrm{eff}}})(\mathbf{I} - 4\gamma_0\widehat{\boldsymbol{\Sigma}})^{L_{\mathrm{eff}}}}{\widehat{\boldsymbol{\Sigma}}}\right).$$

Since $f(x) = (1 - (1 - 2\gamma_0 x)^{L_{\mathrm{eff}}})(1 - 4\gamma_0 x)^{L_{\mathrm{eff}}}/x$ satisfies $f(x) \geqslant L_{\mathrm{eff}}\gamma_0/10$ for $x \in [0, 1/(8\gamma_0 L_{\mathrm{eff}})]$, it follows that $\frac{(\mathbf{I} - (\mathbf{I} - 2\gamma_0\widehat{\boldsymbol{\Sigma}})^{L_{\mathrm{eff}}})(\mathbf{I} - 4\gamma_0\widehat{\boldsymbol{\Sigma}})^{L_{\mathrm{eff}}}}{\widehat{\boldsymbol{\Sigma}}}$ has at most $\tilde{t} = \#\{i \in [M] : \widehat{\lambda}_i L_{\mathrm{eff}}\gamma_0 > 1/8\}$ eigenvalues that are less than $L_{\mathrm{eff}}\gamma_0/10$. Therefore, we have

$$\mathrm{tr}\left((\mathbf{I} - (\mathbf{I} - 2\gamma_0\widehat{\boldsymbol{\Sigma}})^{L_{\mathrm{eff}}})(\mathbf{I} - 4\gamma_0\widehat{\boldsymbol{\Sigma}})^{L_{\mathrm{eff}}} \boldsymbol{\Sigma}\right) = \sum_{j=1}^{M} \mu_j\left((\mathbf{I} - (\mathbf{I} - 2\gamma_0\widehat{\boldsymbol{\Sigma}})^{L_{\mathrm{eff}}})(\mathbf{I} - 4\gamma_0\widehat{\boldsymbol{\Sigma}})^{L_{\mathrm{eff}}} \boldsymbol{\Sigma}\right)$$

$$\geqslant \frac{L_{\mathrm{eff}}\gamma_0}{10} \cdot \sum_{i > \tilde{t}} \mu_i(\widehat{\boldsymbol{\Sigma}}) \cdot \mu_i(\boldsymbol{\Sigma}).$$

Putting pieces together and taking expectation over $(\mathbf{x}_i)_{i \in [N]}$, we obtain

$$\mathbb{E}_{(\mathbf{x}_i)_{i \in [N]}}\mathbb{E}_{(\tilde{\boldsymbol{\epsilon}}_k)_{k \in [N]}}[\|\boldsymbol{\Sigma}^{1/2}(\mathbf{v}_L - \boldsymbol{\theta}_L)\|_2^2] \gtrsim \tilde{\sigma}^2(\mathbf{w}^*) \cdot \mathbb{E}_{(\mathbf{x}_i)_{i \in [N]}}\left[\frac{B_{\boldsymbol{\xi}}\gamma_0 L_{\mathrm{eff}}\gamma_0}{10} \cdot \sum_{i > \tilde{t}} \mu_i(\widehat{\boldsymbol{\Sigma}}) \cdot \mu_i(\boldsymbol{\Sigma})\right].$$

**Proof of claim** (23). By Eq. (4) in the proof of Theorem 3.1, we have

$$\boldsymbol{\theta}_t - \mathbf{v} = -\prod_{i=1}^{t}\left(\mathbf{I} - \gamma_i \widehat{\boldsymbol{\Sigma}}\right)\mathbf{v}^* + \mathbf{V}_t(\widehat{\boldsymbol{\Sigma}})\mathbf{S}\mathbf{X}^\top \tilde{\boldsymbol{\epsilon}},$$

where

$$\mathbf{V}_t(\widehat{\boldsymbol{\Sigma}}) := \frac{1}{N}\sum_{i=1}^{t} \gamma_i \cdot \prod_{j=i+1}^{t}(\mathbf{I} - \gamma_j \widehat{\boldsymbol{\Sigma}}) = \frac{\mathbf{I} - \prod_{i=1}^{t}(\mathbf{I} - \gamma_i \widehat{\boldsymbol{\Sigma}})}{N\widehat{\boldsymbol{\Sigma}}}.$$

Let

$$\boldsymbol{\xi}_i^s := -(\mathbf{S}\mathbf{x}_{i_t}\mathbf{x}_{i_t}^\top \mathbf{S}^\top - \widehat{\boldsymbol{\Sigma}})\mathbf{V}_{t-1}(\widehat{\boldsymbol{\Sigma}})\mathbf{S}\mathbf{X}^\top \tilde{\boldsymbol{\epsilon}} + (\mathbf{S}\mathbf{x}_{i_t}\tilde{\epsilon}_{i_t} - \mathbf{S}\mathbf{X}^\top \tilde{\boldsymbol{\epsilon}}/N)$$

$$= \boldsymbol{\xi}_{1,i} + \boldsymbol{\xi}_{2,i} + (\mathbf{S}\mathbf{x}_{i_t}\mathbf{x}_{i_t}^\top\mathbf{S}^\top - \widehat{\boldsymbol{\Sigma}})\prod_{i=1}^{t}(\mathbf{I} - \gamma_i\widehat{\boldsymbol{\Sigma}})\mathbf{v}^*.$$

Since $\mathbb{E}_{i_t,t\in[L]}[\boldsymbol{\xi}_i^s] = 0$, it can be verified that

$$\mathbb{E}_{\boldsymbol{\epsilon}}\mathbb{E}_{i_t,t\in[L]}[(\boldsymbol{\xi}_{1,i} + \boldsymbol{\xi}_{2,i})(\boldsymbol{\xi}_{1,i} + \boldsymbol{\xi}_{2,i})^\top]$$

$$\geqslant \mathbb{E}_{\boldsymbol{\epsilon}}\mathbb{E}_{i_t,t\in[L]}[\boldsymbol{\xi}_i^s\boldsymbol{\xi}_i^{s\top}]$$

$$\geq \tilde{\sigma}^2(\mathbf{w}^*)\cdot\Big[\frac{N-1}{N}\widehat{\boldsymbol{\Sigma}} - (\mathbf{S}\mathbf{x}_{i_t}\mathbf{x}_{i_t}^\top\mathbf{S}^\top - \widehat{\boldsymbol{\Sigma}})\Big[\frac{\mathbf{I} - \prod_{i=1}^t(\mathbf{I} - \gamma_i\widehat{\boldsymbol{\Sigma}})^2}{N\widehat{\boldsymbol{\Sigma}}}\Big](\mathbf{S}\mathbf{x}_{i_t}\mathbf{x}_{i_t}^\top\mathbf{S}^\top - \widehat{\boldsymbol{\Sigma}})^\top\Big]$$

$$\geq \tilde{\sigma}^2(\mathbf{w}^*)\cdot\Big[\frac{N-1}{N}\widehat{\boldsymbol{\Sigma}} - (\mathbf{S}\mathbf{x}_{i_t}\mathbf{x}_{i_t}^\top\mathbf{S}^\top - \widehat{\boldsymbol{\Sigma}})\Big[\frac{\mathbf{I} - (\mathbf{I} - 2\gamma_0\widehat{\boldsymbol{\Sigma}})^{2L_{\text{eff}}}}{N\widehat{\boldsymbol{\Sigma}}}\Big](\mathbf{S}\mathbf{x}_{i_t}\mathbf{x}_{i_t}^\top\mathbf{S}^\top - \widehat{\boldsymbol{\Sigma}})^\top\Big].$$

Since $\sup_{x\in[0,1/(2\tilde{\gamma})]}(1 - (1 - 2\tilde{\gamma}x)^{2L_{\text{eff}}})/x \leqslant 4\tilde{\gamma}L_{\text{eff}}$ and $\mathbb{E}_{i_t}[(\mathbf{S}\mathbf{x}_{i_t}\mathbf{x}_{i_t}^\top\mathbf{S}^\top - \widehat{\boldsymbol{\Sigma}})^2] \leq \mathbb{E}_{i_t}[(\mathbf{S}\mathbf{x}_{i_t}\mathbf{x}_{i_t}^\top\mathbf{S}^\top)^2] \leq B_{\boldsymbol{\nu}}^2\widehat{\boldsymbol{\Sigma}}$, we further have

$$\mathbb{E}_{\boldsymbol{\epsilon}}\mathbb{E}_{i_t,t\in[L]}[(\boldsymbol{\xi}_{1,i} + \boldsymbol{\xi}_{2,i})(\boldsymbol{\xi}_{1,i} + \boldsymbol{\xi}_{2,i})^\top] \geq \tilde{\sigma}^2(\mathbf{w}^*)\cdot\Big[\frac{N-1}{N}\widehat{\boldsymbol{\Sigma}} - \frac{4\gamma_0 L_{\text{eff}}}{N}\mathbb{E}_{i_t}[(\mathbf{S}\mathbf{x}_{i_t}\mathbf{x}_{i_t}^\top\mathbf{S}^\top - \widehat{\boldsymbol{\Sigma}})^2]\Big]$$

$$\geq \tilde{\sigma}^2(\mathbf{w}^*)\cdot\Big(\frac{N-1}{N} - \frac{4\gamma_0 L_{\text{eff}}}{N}B_{\boldsymbol{\nu}}^2\Big)\widehat{\boldsymbol{\Sigma}}$$

$$\geq \tilde{\sigma}^2(\mathbf{w}^*)\cdot\widehat{\boldsymbol{\Sigma}}/2$$

when $(L_{\text{eff}}\gamma)B_{\boldsymbol{\nu}}^2/N \leqslant 1/3$.

$\square$

### D.3 Fluctuation error under the source condition

**Lemma D.5** (Fluctuation error under the source condition). *Under the notation and assumptions in Theorem 3.1 and suppose that $L_{\text{eff}} \lesssim N^{(1-\varepsilon)a}/\gamma$ for some small constant $\varepsilon \in (0,1]$. For any $s \in [0, 1 - 1/a)$, there exists some $(s,\varepsilon,a)$-dependent constant $c > 0$ such that the* (multi-pass SGD) *process satisfies*

$$\mathbb{E}[\mathsf{Fluc}] \leqslant c\gamma\log N \cdot \Big[1 + \frac{\log^2 N(L_{\text{eff}}\gamma)^{2-s}}{N^2}\Big](L_{\text{eff}}\gamma)^{1/a-1}.$$

*with probability at least $1 - \exp(-\Omega(M))$ over the randomness of $\mathbf{S}$. Consequently, choosing $s = 1 - 1/(a(1 - \varepsilon/2))$ yields*

$$\mathbb{E}[\mathsf{Fluc}] \lesssim \gamma\log N \cdot \Big[1 + \frac{\log^2 N(L_{\text{eff}}\gamma)^{1/(a(1-\varepsilon/2))+1}}{N^2}\Big](L_{\text{eff}}\gamma)^{1/a-1}$$

$$\leqslant c' \cdot \gamma\log N \cdot \Big[(L_{\text{eff}}\gamma)^{1/a-1} + \frac{(L_{\text{eff}}\gamma)^{1/a}}{N}\Big]$$

*for some $(\varepsilon,a)$-dependent constant $c' > 0$ with probability at least $1 - \exp(-\Omega(M))$.*

*Moreover, assume in addition that $L_{\text{eff}} \lesssim N/\gamma$. Then with probability at least $1 - \exp(-\Omega(M))$ over the randomness of $\mathbf{S}$, we have*

$$\mathbb{E}[\mathsf{Fluc}] \geqslant c''\gamma(L_{\text{eff}}\gamma)^{1/a-1}$$

*for some $a$-dependent constant $c'' > 0$.*

*Proof of Lemma D.5.* The proof follows from instantiating Lemma D.1 and D.4 under the source condition. We start by establishing concentration bounds on some quantities that appear in the bounds in Lemma D.1 and D.4.

First, note that we have for any $s \in [0, 1 - 1/a)$, conditioned on $\mathbf{S}$ and $\mathbf{w}^*$, with probability at least $1 - \delta$ over $(\mathbf{x}_i, y_i)_{i=1}^{N}$,

$$\max_{i\in[N]}(\mathbf{x}_i^\top\mathbf{S}^\top\mathbf{v}^*)^2 \lesssim \|\mathbf{w}^*\|_{\mathbf{H}}^2\log(N/\delta), \tag{24a}$$

$$\max_{i\in[N]} \tilde{\epsilon}_i^2 \lesssim (\sigma^2 + \|\mathbf{w}^*\|_{\mathbf{H}}^2) \log(N/\delta), \tag{24b}$$

$$\max_{i\in[N]} \|\mathbf{x}_i^\top \mathbf{S}^\top\|_{\mathbf{\Sigma}^{-s}}^2 \lesssim \mathrm{tr}(\mathbf{\Sigma}^{1-s}) + \log(N/\delta) + \sqrt{\mathrm{tr}(\mathbf{\Sigma}^{2-2s})\log(N/\delta)} \lesssim \log(N/\delta), \tag{24c}$$

where Eq. (24a) and (24b) follow from a union bound on concentration inequalities for Gaussian random variables; Eq. (24c) uses Hanson-Wright inequality and Lemma E.5.

Moreover, we will show that conditioned on $\mathbf{S}$, $\mathbf{w}^*$ and $\boldsymbol{\theta}_t^{(-i)}$, $\mathbf{x}_i^\top \mathbf{S}^\top \boldsymbol{\theta}_t^{(-i)}$ is a zero-mean random Gaussian variable with covariance

$$\mathbb{E}[(\mathbf{x}_i^\top \mathbf{S}^\top \boldsymbol{\theta}_t^{(-i)})^2 \mid \mathbf{S}, \mathbf{w}^*, \boldsymbol{\theta}_t^{(-i)}] = \boldsymbol{\theta}_t^{(-i)\top} \mathbf{\Sigma} \boldsymbol{\theta}_t^{(-i)} \lesssim (T_\mathbf{B} + T_\mathbf{V}) \cdot (\sigma^2 + \|\mathbf{w}^*\|_{\mathbf{H}}^2), \tag{25}$$

where

$$T_\mathbf{B} := \begin{cases} 1 + \log(1/\delta)/N + \mathsf{t}(\delta) \cdot (L_{\mathtt{eff}}\gamma)^2 \cdot (1 + \log(1/\delta)/N)^2 & \text{when } M \leqslant N/2, \\ \mathbf{B}_\mathbf{B} \cdot (1 + \log(1/\delta)/N)^2 + 1 & \text{when } M > N/2, \end{cases}$$

$$T_\mathbf{V} := \max_{i\in[N]} \|(\widehat{\mathbf{\Sigma}}^{(-i)} + \lambda\mathbf{I})^{-1/2}(\mathbf{\Sigma} + \lambda\mathbf{I})^{1/2}\|^2 \cdot \max_{i\in[N]} \tilde{\epsilon}_i^2 \cdot \widetilde{D}^{\mathsf{U}}/N$$

with $\mathbf{B}_\mathbf{B}$ defined in Lemma B.1, $\widetilde{D}^{\mathsf{U}}$ defined in Eq. (17), and $\mathsf{t}(\delta) := \mathbb{1}_{\{\log(1/\delta) \gtrsim N\}}$. Thus, we have by a union bound that

$$\max_{i\in[N],t\in[L]} (\mathbf{x}_i^\top \mathbf{S}^\top \boldsymbol{\theta}_t^{(-i)})^2 \lesssim \max_{i\in[N],t\in[L]} \boldsymbol{\theta}_t^{(-i)\top} \mathbf{\Sigma} \boldsymbol{\theta}_t^{(-i)} \cdot \log(NL/\delta)$$

$$\lesssim (T_\mathbf{B} + T_\mathbf{V}) \cdot (\sigma^2 + \|\mathbf{w}^*\|_{\mathbf{H}}^2) \cdot \log(N/\delta) \tag{26}$$

with probability at least $1 - \delta$ over the randomness of $(\mathbf{x}_i, y_i)_{i=1}^N$ conditioned on $\mathbf{S}$ and $\mathbf{w}^*$. Moreover, we note that $\mu_j(\mathbf{\Sigma}) \asymp j^{-a}$ for $j \in [M]$ with probability at least $1 - \exp(-\Omega(M))$ by Lemma E.5, and conditioned on $\mathbf{S}$ and $\mathbf{w}^*$, we have $\mu_j(\widehat{\mathbf{\Sigma}}) \asymp j^{-a}$ for $j \leqslant \min\{M, N/c\}$ and $\mu_j(\widehat{\mathbf{\Sigma}}) \lesssim j^{-a}$ otherwise with probability at least $1 - \exp(-\Omega(N))$ by Lemma E.8.

**Proof of the upper bound.** Therefore, substituting Eq. (24a)—(24c) and (26) into the expression in $a_{\max}$, $\mathbf{B}_\mathbf{\Delta}$ and $\mathsf{F}$, applying Eq. (16) to bound $\widetilde{D}^{\mathsf{U}}$ (and $\mathrm{tr}(\widehat{\mathbf{\Sigma}}^{1/\alpha})$ for some $\alpha = a + \varepsilon > a$), using part 2 of Lemma D.1[4] and taking expectation w.r.t. $(\mathbf{x}_i, y_i)_{i=1}^N$ conditioned on $\mathbf{S}$ and $\mathbf{w}^*$, it can be verified that

$$\mathbb{E}[\mathsf{Fluc} \mid \mathbf{S}, \mathbf{w}^*] \lesssim (\sigma^2 + \|\mathbf{w}^*\|_{\mathbf{H}}^2) \cdot \gamma \log N \cdot \left[1 + \frac{\log^2 N (L_{\mathtt{eff}}\gamma)^{2-s}}{N^2}\right] \cdot (L_{\mathtt{eff}}\gamma)^{1/a-1}.$$

Taking expectation w.r.t. $\mathbf{w}^*$ yields the desired result.

**Proof of the lower bound.** Setting $s = 0$ in Eq. (24c), we have $B_{\boldsymbol{\xi}} \geqslant 1/2$ when $\gamma L_{\mathtt{eff}} B_{\boldsymbol{\xi}}/N \geqslant 1/3$, which happens with probability at least $1 - N^{-c_1/c_2}$ for some constant $c_1 > 0$ when $\gamma \leqslant c_2/\log N$ for some $c_2 > 0$. Moreover, by the concentration properties on $\mu_j(\widehat{\mathbf{\Sigma}})$ and $\mu_j(\mathbf{\Sigma})$ in the previous discussion, the assumptions on $\gamma$, and a union bound, conditioned on $\mathbf{S}$ such that $\mu_j(\mathbf{\Sigma}) \asymp j^{-a}$ for $j \in [M]$ (which happens with probability at least $1 - e^{-\Omega(M)}$), we have with probability at least $1/2$ over the randomness of $(\mathbf{x}_i, y_i)_{i=1}^N$ that

$$\mu_j(\widehat{\mathbf{\Sigma}}) \asymp j^{-a} \quad \text{for} \quad j \leqslant \min\{M, N/c\}, \quad \text{and}$$

$$\tilde{t} \lesssim (L_{\mathtt{eff}}\gamma)^{1/a}, \quad B_{\boldsymbol{\xi}} \geqslant 1/2, \quad \gamma_0 = \gamma.$$

Thus, when $(L_{\mathtt{eff}}\gamma)^{1/a} \leqslant M/\tilde{c} \leqslant \min\{M, N/c\}$ for some $\tilde{c}, c > 0$ sufficiently large, we have by Lemma D.4 that

$$\mathbb{E}[\mathsf{Fluc}] \gtrsim \mathbb{E}_{(\mathbf{x}_i)_{i\in[N]}}\left[\frac{B_{\boldsymbol{\xi}}\gamma_0 L_{\mathtt{eff}}\gamma_0}{10} \cdot \sum_{i>\tilde{t}} \mu_i(\widehat{\mathbf{\Sigma}}) \cdot \mu_i(\mathbf{\Sigma})\right]$$

---

[4]More specifically, we apply part 2 of Lemma D.1 on the event with probability at least $1 - \exp(-\Omega(N))$ where conditions on $\widehat{\mathbf{\Sigma}}$ specified in Lemma E.8 hold, and apply part 1 of Lemma D.1 for some $\alpha = a + \varepsilon > 1$ otherwise.

$$\gtrsim L_{\text{eff}}\gamma^2 \cdot \sum_{i=\tilde{t}+1}^{\min\{M,N/c\}} i^{-2a}$$

$$\gtrsim L_{\text{eff}}\gamma^2 \cdot \tilde{t}^{1-2a} \gtrsim \gamma(L_{\text{eff}}\gamma)^{1/a-1}$$

with probability at least $1 - e^{-\Omega(M)}$ over the randomness of $\mathbf{S}$.

**Proof of claim** (25). Note that

$$\boldsymbol{\theta}_t^{(-i)\top}\boldsymbol{\Sigma}\boldsymbol{\theta}_t^{(-i)} \lesssim \mathbf{v}^{*\top}\boldsymbol{\Sigma}\mathbf{v}^* + (\boldsymbol{\theta}_t^{(-i)} - \mathbf{v}^*)^\top\boldsymbol{\Sigma}(\boldsymbol{\theta}_t^{(-i)} - \mathbf{v}^*)$$

$$\overset{(i)}{\lesssim} \|\mathbf{v}^*\|_{\boldsymbol{\Sigma}}^2 + \|\prod_{i=1}^t (\mathbf{I} - \gamma_i\widehat{\boldsymbol{\Sigma}}^{(-i)})\mathbf{v}^*\|_{\boldsymbol{\Sigma}}^2 + \|\mathbf{V}(\widehat{\boldsymbol{\Sigma}}^{(-i)})(\mathbf{S}\mathbf{X}^\top\tilde{\boldsymbol{\epsilon}})^{(-i)}\|_{\boldsymbol{\Sigma}}^2.$$

where step (i) follows from the decomposition

$$\boldsymbol{\theta}_t^{(-i)} - \mathbf{v}^* = \prod_{t=1}^t \left(\mathbf{I} - \gamma_t\widehat{\boldsymbol{\Sigma}}^{(-i)}\right)(\boldsymbol{\theta}_0 - \mathbf{v}^*) + \mathbf{V}(\widehat{\boldsymbol{\Sigma}}^{(-i)})(\mathbf{S}\mathbf{X}^\top\tilde{\boldsymbol{\epsilon}})^{(-i)}$$

$$= -\prod_{t=1}^t \left(\mathbf{I} - \gamma_t\widehat{\boldsymbol{\Sigma}}^{(-i)}\right)\mathbf{v}^* + \mathbf{V}(\widehat{\boldsymbol{\Sigma}}^{(-i)})(\mathbf{S}\mathbf{X}^\top\tilde{\boldsymbol{\epsilon}})^{(-i)}$$

as similar to Eq. (4), where $(\mathbf{S}\mathbf{X}^\top\tilde{\boldsymbol{\epsilon}})^{(-i)} := \sum_{j\neq i}\mathbf{S}\mathbf{x}_j\tilde{\epsilon}_j/N$ and

$$\mathbf{V}(\widehat{\boldsymbol{\Sigma}}^{(-i)}) := \frac{1}{N}\sum_{i=1}^t \gamma_i \cdot \prod_{j=i+1}^t (\mathbf{I} - \gamma_j\widehat{\boldsymbol{\Sigma}}^{(-i)}) = \frac{\mathbf{I} - \prod_{i=1}^t(\mathbf{I} - \gamma_i\widehat{\boldsymbol{\Sigma}}^{(-i)})}{N\widehat{\boldsymbol{\Sigma}}^{(-i)}}.$$

Let $\bar{\mathbf{w}}^* := \boldsymbol{\Sigma}^{1/2}\mathbf{v}^*$. Note that $(\boldsymbol{\theta}_t^{(-i)})_{t=1}^L$ can be viewed as a (GD) process on $(\mathbf{x}_j, y_j)_{j\neq i}$ with stepsize $(N-1)\gamma_t/N$.

Following the proof of Lemma B.1 (Eq. 11 and 12), it can be verified that, conditioned on $\mathbf{S}$ and $\mathbf{w}^*$,

$$\|\prod_{i=1}^t (\mathbf{I} - \gamma_i\widehat{\boldsymbol{\Sigma}}^{(-i)})\mathbf{v}^*\|_{\boldsymbol{\Sigma}}^2 \lesssim \begin{cases} B_{\mathsf{F},1} \overset{d}{=} \|\mathbf{Z}\bar{\mathbf{w}}^*\|_2^2 + (L_{\text{eff}}\gamma)^2\|\mathbf{Z}^\top\mathbf{Z}\|_2^2 \mathbf{1}_{\{\mathbf{Z}^\top\mathbf{Z}\not\preceq\mathbf{I}_M/5\}} \cdot \|\bar{\mathbf{w}}^*\|_2^2 & \text{when } M \leqslant N/2, \\ B_{\mathsf{F},2} \overset{d}{=} B_{\mathsf{F},3} \cdot \|\mathbf{Z}\bar{\mathbf{w}}^*\|_2^2 + \|\bar{\mathbf{w}}^*\|_2^2 & \text{when } M > N/2, \end{cases}$$

where $\mathbf{Z} \in \mathbb{R}^{(N-1)\times M}$ has i.i.d $\mathcal{N}(0, 1/N)$ entries and

$$B_{\mathsf{F},3} := (L_{\text{eff}}\gamma)^2\|\mathbf{Z}_{\tilde{k}:\infty}\boldsymbol{\Sigma}_{\tilde{k}:\infty}\mathbf{Z}_{\tilde{k}:\infty}^\top\|_2^2 + 1 + \|\mathbf{Z}_{\tilde{k}:\infty}\boldsymbol{\Sigma}_{\tilde{k}:\infty}^2\mathbf{Z}_{\tilde{k}:\infty}^\top\|_2$$

with $\tilde{k} = N/2$. In addition, we have $\|\bar{\mathbf{w}}^*\|_2^2 \leqslant \|\mathbf{v}^*\|_{\boldsymbol{\Sigma}}^2 \leqslant \|\mathbf{w}^*\|_{\mathbf{H}}^2$ and $\|\mathbf{Z}\bar{\mathbf{w}}^*\|_2^2 \leqslant \|\mathbf{v}^*\|_{\boldsymbol{\Sigma}}^2 \cdot (2 + \log(1/\delta)/N)$ with probability at least $1 - \delta$ by concentration properties of chi-squared random variables. Therefore, putting pieces together, applying Lemma E.5, Eq. (13) and concentration properties of Gaussian covariance matrices (see e.g., Theorem 6.1 in Wainwright (2019)), we obtain with probability at least $1 - \delta$ conditioned on $\mathbf{S}$ and $\mathbf{w}^*$,

$$\|\prod_{i=1}^t (\mathbf{I} - \gamma_i\widehat{\boldsymbol{\Sigma}}^{(-i)})\mathbf{v}^*\|_{\boldsymbol{\Sigma}}^2$$

$$\lesssim \begin{cases} \|\mathbf{w}^*\|_{\mathbf{H}}^2 \cdot (2 + \log(1/\delta)/N + \mathsf{t}(\delta) \cdot (L_{\text{eff}}\gamma)^2 \cdot (2 + \log(1/\delta)/N)^2) & \text{when } M \leqslant N/2, \\ \|\mathbf{w}^*\|_{\mathbf{H}}^2 \cdot (\mathbf{B}_{\mathbf{B}} \cdot (1 + \log(1/\delta)/N)^2 + 1) & \text{when } M > N/2, \end{cases} \tag{27}$$

where $\mathsf{t}(\delta) := \mathbf{1}_{\{\log(1/\delta)\gtrsim N\}}$ and $\mathbf{B}_{\mathbf{B}}$ is defined in Lemma B.1, with probability at least $1 - e^{-\Omega(M)}$ over the randomness of $\tilde{\mathbf{S}}$.

Adopt the shorthand $\mathsf{V}_t^{(-i)}$ for $\mathbf{I} - \prod_{t=1}^L(\mathbf{I} - \gamma_t\widehat{\boldsymbol{\Sigma}})$ and $R_i$ for $\|(\widehat{\boldsymbol{\Sigma}}^{(-i)} + \lambda\mathbf{I})^{-1/2}(\boldsymbol{\Sigma}^{(-i)} + \lambda\mathbf{I})^{1/2}\|_2$.

Similarly, following the proof of the upper bound in Lemma C.1, we have (choosing $\lambda = 1/(L_{\text{eff}}\gamma)$)

$$\|\mathbf{V}(\widehat{\boldsymbol{\Sigma}}^{(-i)})(\mathbf{S}\mathbf{X}^\top\tilde{\boldsymbol{\epsilon}})^{(-i)}\|_{\boldsymbol{\Sigma}}^2$$

$$\lesssim \frac{\max_{i\in[N]}\tilde{\epsilon}_i^2\cdot}{N}\cdot\operatorname{tr}(\mathsf{V}_t^{(-i)}\mathbf{\Sigma}\mathsf{V}_t^{(-i)}\widehat{\mathbf{\Sigma}}^{-1})$$

$$\leqslant R_i\cdot\frac{\max_{i\in[N]}\tilde{\epsilon}_i^2\cdot}{N}\cdot\operatorname{tr}(\mathsf{V}_t^{(-i),2}+\lambda\widehat{\mathbf{\Sigma}}^{-1}\mathsf{V}_t^{(-i),2})$$

$$\overset{(ii)}{\lesssim}\frac{R_i}{N}\cdot\max_{i\in[N]}\tilde{\epsilon}_i^2\cdot\widetilde{D}_i^{\mathsf{U}}\overset{(iii)}{\lesssim}\frac{R_i}{N}\cdot\max_{i\in[N]}\tilde{\epsilon}_i^2\cdot\widetilde{D}^{\mathsf{U}}\tag{28}$$

where $\widetilde{D}^{\mathsf{U}}$ is defined in Eq. (17) and

$$\widetilde{D}_i^{\mathsf{U}}:=\#\{j\in[M]:\widehat{\lambda}_j^{(-i)}L_{\mathtt{eff}}\gamma_0>1/4\}+(L_{\mathtt{eff}}\gamma_0)\sum_{j:\widehat{\lambda}_j^{(-i)}L_{\mathtt{eff}}\gamma_0\leqslant1/4}\widehat{\lambda}_j^{(-i)},$$

and $\widehat{\lambda}_j^{(-i)}$ is the $j$-th largest eigenvalue of $\widehat{\mathbf{\Sigma}}^{(-i)}$. Here, step (ii) uses Eq. (17) and step (iii) uses the fact that $\widehat{\lambda}_j^{(-i)}\leqslant\widehat{\lambda}_j$ for all $j\in[M]$ since $\widehat{\mathbf{\Sigma}}^{(-i)}\preceq\widehat{\mathbf{\Sigma}}$.

$\square$

## D.4 Proof of Lemma D.2

The proof of Lemma D.2 follows from similar ideas as in the proof of Proposition 1 of Pillaud-Vivien et al. (2018). We first state a few lemmas that contribute to the proof. These lemmas are modified versions of the lemmas in Pillaud-Vivien et al. (2018), but we provide their proofs here for completeness.

**Lemma D.6** (Semi-stochastic SGD; Lemma 1 in Pillaud-Vivien et al. (2018)). *Under the notation and assumptions in Lemma D.2, consider any stochastic process $\widetilde{\boldsymbol{\mu}}_t=(\mathbf{I}-\gamma_t\mathbf{\Sigma}_{\boldsymbol{\nu}})\widetilde{\boldsymbol{\mu}}_{t-1}+\gamma_t\cdot\widetilde{\boldsymbol{\xi}}_t$ with $\widetilde{\boldsymbol{\mu}}_0=\mathbf{0},t\in[L]$ and $(\widetilde{\boldsymbol{\xi}}_t)_{t=1}^L$ such that $\mathbb{E}[\widetilde{\boldsymbol{\xi}}_t]=\mathbf{0}$ and $\mathbb{E}[\widetilde{\boldsymbol{\xi}}_t\widetilde{\boldsymbol{\xi}}_t^\top]\preceq\widetilde{\sigma}_\xi^2\mathbf{\Sigma}_{\boldsymbol{\nu}}$. Then for any $u\in[0,1]$, we have*

$$\mathbb{E}[\|\mathbf{\Sigma}_{\boldsymbol{\nu}}^{u/2}\widetilde{\boldsymbol{\mu}}_L\|_2^2]\leqslant c\cdot\widetilde{\sigma}_\xi^2\gamma_0\operatorname{tr}(\mathbf{\Sigma}_{\boldsymbol{\nu}}^{1/\alpha})\cdot(L_{\mathtt{eff}}\gamma_0)^{1/\alpha-u}$$

*for any $\alpha>1$ and some $\alpha$-dependent constant $c>0$. Moreover, there exists some $a$-dependent constant $c',\tilde{c}>1$ such that when $\mu_j(\mathbf{\Sigma}_{\boldsymbol{\nu}})\asymp j^{-a}$ for $j\leqslant\min\{M,N/\tilde{c}\}$, we have*

$$\mathbb{E}[\|\mathbf{\Sigma}_{\boldsymbol{\nu}}^{u/2}\widetilde{\boldsymbol{\mu}}_L\|_2^2]\leqslant c'\widetilde{\sigma}_\xi^2\cdot\gamma_0(L_{\mathtt{eff}}\gamma_0)^{1/a-u}$$

*for any $u\in[0,1]$.*

See the proof of Lemma D.6 in Section D.4.1.

Following the ideas in (Pillaud-Vivien et al., 2018; Aguech et al., 2000), we introduce a sequence of stochastic processes $(\widetilde{\boldsymbol{\mu}}_t^k)_{t=0}^L$ that connects the SGD process in (19) to the semi-stochastic SGD in Lemma D.6. Namely, for $k\geqslant0$, we define

$$\widetilde{\boldsymbol{\mu}}_t^k=(\mathbf{I}-\gamma_t\mathbf{\Sigma}_{\boldsymbol{\nu}})\widetilde{\boldsymbol{\mu}}_{t-1}^k+\gamma_t\cdot\boldsymbol{\xi}_t^k,\quad\widetilde{\boldsymbol{\mu}}_0^k=\mathbf{0},\quad t\in[L],\tag{29}$$

where $\boldsymbol{\xi}_t^0:=\boldsymbol{\xi}_t$ and $\boldsymbol{\xi}_t^k:=(\mathbf{\Sigma}_{\boldsymbol{\nu}}-\boldsymbol{\nu}_t\boldsymbol{\nu}_t^\top)\widetilde{\boldsymbol{\mu}}_{t-1}^{k-1}$ for $k\geqslant1$. It can be verified that

$$\boldsymbol{\mu}_t-\sum_{i=0}^k\widetilde{\boldsymbol{\mu}}_t^i=(\mathbf{I}-\gamma_t\boldsymbol{\nu}_t\boldsymbol{\nu}_t^\top)\Big(\boldsymbol{\mu}_{t-1}-\sum_{i=0}^{k-1}\widetilde{\boldsymbol{\mu}}_{t-1}^i\Big)+\gamma_t\cdot\boldsymbol{\xi}_t^{k+1}.$$

**Lemma D.7** (Bounds on the covariance; Lemma 2 in Pillaud-Vivien et al. (2018)). *Under the notation and assumptions in Lemma D.2 and its proof, for any $k\geqslant0$, we have*

$$\mathbb{E}[\boldsymbol{\xi}_t^k\boldsymbol{\xi}_t^{k\top}]\preceq\sigma_\xi^2\gamma_0^kB_{\boldsymbol{\nu}}^{2k}\cdot\mathbf{\Sigma}_{\boldsymbol{\nu}}\quad\text{and}\quad\mathbb{E}[\widetilde{\boldsymbol{\mu}}_t^k\widetilde{\boldsymbol{\mu}}_t^{k\top}]\preceq\sigma_\xi^2\gamma_0^{k+1}B_{\boldsymbol{\nu}}^{2k}\cdot\mathbf{I}.$$

See the proof of Lemma D.7 in Section D.4.2.

**Lemma D.8** (SGD recursion; Lemma 3 in Pillaud-Vivien et al. (2018)). *Under the notation and assumptions in Lemma D.2, consider any stochastic process $\widehat{\boldsymbol{\mu}}_t=(\mathbf{I}-\gamma_t\boldsymbol{\nu}_t\boldsymbol{\nu}_t^\top)\widehat{\boldsymbol{\mu}}_{t-1}+\gamma_t\cdot\widehat{\boldsymbol{\xi}}_t$, with $\widehat{\boldsymbol{\mu}}_0=\mathbf{0},t\in[L]$ and $(\widehat{\boldsymbol{\xi}}_t)_{t=1}^L$ such that $\mathbb{E}[\widehat{\boldsymbol{\xi}}_t]=\mathbf{0}$ and $\mathbb{E}[\widehat{\boldsymbol{\xi}}_t\widehat{\boldsymbol{\xi}}_t^\top]\preceq\widehat{\sigma}_\xi^2\mathbf{\Sigma}_{\boldsymbol{\nu}}$. Then*

$$\mathbb{E}[\|\mathbf{\Sigma}_{\boldsymbol{\nu}}^{u/2}\widehat{\boldsymbol{\mu}}_L\|_2^2]\leqslant2\widehat{\sigma}_\xi^2\cdot\gamma_0^2B_{\boldsymbol{\nu}}^{2u}\operatorname{tr}(\mathbf{\Sigma}_{\boldsymbol{\nu}})L_{\mathtt{eff}}.$$

*for any $u\in[0,1]$.*

See the proof of Lemma D.8 in Section D.4.3.

With these lemmas at hand, we are ready to prove Lemma D.2. Performing a decomposition as in the proof of Proposition 1 in Pillaud-Vivien et al. (2018) and using Lemma D.6 on $\widetilde{\boldsymbol{\mu}}_t^i$ for $i \in [0, k]$ and D.8 on $\boldsymbol{\mu}_L - \sum_{i=0}^{k} \widetilde{\boldsymbol{\mu}}_L^i$, we find

$$
(\mathbb{E}[\|\boldsymbol{\Sigma}_{\boldsymbol{\nu}}^{u/2}\boldsymbol{\mu}_L\|_2^2])^{1/2}
$$

$$
\leqslant \sum_{i=0}^{k} (\mathbb{E}[\|\boldsymbol{\Sigma}_{\boldsymbol{\nu}}^{u/2}\widetilde{\boldsymbol{\mu}}_L^i\|_2^2])^{1/2} + (\mathbb{E}[\|\boldsymbol{\Sigma}_{\boldsymbol{\nu}}^{u/2}(\boldsymbol{\mu}_L - \sum_{i=0}^{k}\widetilde{\boldsymbol{\mu}}_L^i)\|_2^2])^{1/2}
$$

$$
\lesssim \sum_{i=0}^{k} (\sigma_\xi^2 \gamma_0^i B_{\boldsymbol{\nu}}^{2i} \cdot \gamma_0 \operatorname{tr}(\boldsymbol{\Sigma}_{\boldsymbol{\nu}}^{1/\alpha}) L_{\texttt{eff}}^{1/\alpha - u})^{1/2} + (\sigma_\xi^2 \gamma_0^{k+1} B_{\boldsymbol{\nu}}^{2k+2+2u} \cdot \gamma_0^2 \operatorname{tr}(\boldsymbol{\Sigma}_{\boldsymbol{\nu}}) L_{\texttt{eff}})^{1/2}
$$

$$
\leqslant (\sigma_\xi^2 \cdot \gamma_0 \operatorname{tr}(\boldsymbol{\Sigma}_{\boldsymbol{\nu}}^{1/\alpha})(L_{\texttt{eff}}\gamma_0)^{1/\alpha - u})^{1/2} \cdot \sum_{i=0}^{k} (\gamma_0 B_{\boldsymbol{\nu}}^2)^{i/2} + (\sigma_\xi^2 \gamma_0^{k+3} B_{\boldsymbol{\nu}}^{2k+2+2u} \cdot \operatorname{tr}(\boldsymbol{\Sigma}_{\boldsymbol{\nu}}) L_{\texttt{eff}})^{1/2}
$$

$$
\leqslant 2(\sigma_\xi^2 \cdot \gamma_0 \operatorname{tr}(\boldsymbol{\Sigma}_{\boldsymbol{\nu}}^{1/\alpha})(L_{\texttt{eff}}\gamma_0)^{1/\alpha - u})^{1/2} + (\sigma_\xi^2 \gamma_0^{k+3} B_{\boldsymbol{\nu}}^{2k+2+2u} \cdot \operatorname{tr}(\boldsymbol{\Sigma}_{\boldsymbol{\nu}}) L_{\texttt{eff}})^{1/2},
$$

where the last inequality follows as $\gamma_0 B_{\boldsymbol{\nu}}^2 \leqslant 1/4$ by the assumption in Lemma D.2. Finally, letting $k \to \infty$ and noting that $\sigma_\xi^2 \gamma_0^{k+3} B_{\boldsymbol{\nu}}^{2k+2+2u} \cdot \operatorname{tr}(\boldsymbol{\Sigma}_{\boldsymbol{\nu}}) L_{\texttt{eff}} \xrightarrow{k\to\infty} 0$, we obtain the desired result. The second part of Lemma D.2 follows from similar arguments and therefore we omit the proof.

### D.4.1 Proof of Lemma D.6

By definition of $\widetilde{\boldsymbol{\mu}}_t$, we have

$$
\widetilde{\boldsymbol{\mu}}_L = \sum_{t=1}^{L} \gamma_t \cdot \prod_{i=t+1}^{L} (\mathbf{I} - \gamma_i \boldsymbol{\Sigma}_{\boldsymbol{\nu}})\boldsymbol{\xi}_t.
$$

Thus,

$$
\mathbb{E}[\|\boldsymbol{\Sigma}_{\boldsymbol{\nu}}^{u/2}\widetilde{\boldsymbol{\mu}}_L\|_2^2]
$$

$$
= \mathbb{E}[\|\boldsymbol{\Sigma}_{\boldsymbol{\nu}}^{u/2} \sum_{t=1}^{L} \gamma_t \cdot \prod_{i=t+1}^{L} (\mathbf{I} - \gamma_i \boldsymbol{\Sigma}_{\boldsymbol{\nu}})\boldsymbol{\xi}_t\|_2^2] = \sum_{t=1}^{L} \gamma_t^2 \cdot \operatorname{tr}(\mathbb{E}[\boldsymbol{\Sigma}_{\boldsymbol{\nu}}^u \prod_{i=t+1}^{L} (\mathbf{I} - \gamma_i \boldsymbol{\Sigma}_{\boldsymbol{\nu}})^2 \boldsymbol{\xi}_t \boldsymbol{\xi}_t^\top])
$$

$$
\leqslant \widetilde{\sigma}_\xi^2 \cdot \sum_{t=1}^{L} \gamma_t^2 \cdot \operatorname{tr}(\prod_{i=t+1}^{L} (\mathbf{I} - \gamma_i \boldsymbol{\Sigma}_{\boldsymbol{\nu}})^2 \boldsymbol{\Sigma}_{\boldsymbol{\nu}}^{1+u})
$$

$$
= \widetilde{\sigma}_\xi^2 \cdot \sum_{k=0}^{\lfloor \log L \rfloor - 1} \gamma_{L_{\texttt{eff}}k+1}^2 \cdot \operatorname{tr}\left(\boldsymbol{\Sigma}_{\boldsymbol{\nu}}^{1+u} \cdot \frac{\mathbf{I} - (\mathbf{I} - \gamma_{L_{\texttt{eff}}k+1}\boldsymbol{\Sigma}_{\boldsymbol{\nu}})^{2L_{\texttt{eff}}}}{2\gamma_{L_{\texttt{eff}}k+1}\boldsymbol{\Sigma}_{\boldsymbol{\nu}} - (\gamma_{L_{\texttt{eff}}k+1}\boldsymbol{\Sigma}_{\boldsymbol{\nu}})^2} \cdot \prod_{j=k+1}^{\lfloor \log L \rfloor - 1} (\mathbf{I} - \gamma_{L_{\texttt{eff}}j+1}\boldsymbol{\Sigma}_{\boldsymbol{\nu}})^{2L_{\texttt{eff}}}\right)
$$

$$
\leqslant \widetilde{\sigma}_\xi^2 \cdot \sum_{k=0}^{\lfloor \log L \rfloor - 2} \gamma_{L_{\texttt{eff}}k+1} \operatorname{tr}(\boldsymbol{\Sigma}_{\boldsymbol{\nu}}^u \cdot (\mathbf{I} - (\mathbf{I} - \gamma_{L_{\texttt{eff}}k+1}\boldsymbol{\Sigma}_{\boldsymbol{\nu}})^{2L_{\texttt{eff}}})(\mathbf{I} - \gamma_{L_{\texttt{eff}}(k+1)+1}\boldsymbol{\Sigma}_{\boldsymbol{\nu}})^{2L_{\texttt{eff}}})
$$

$$
+ \operatorname{tr}(\widetilde{\sigma}_\xi^2 \cdot \frac{\gamma_0}{L} \boldsymbol{\Sigma}_{\boldsymbol{\nu}}^u \cdot (\mathbf{I} - (\mathbf{I} - \gamma_0 \boldsymbol{\Sigma}_{\boldsymbol{\nu}}/L)^{2L_{\texttt{eff}}})), \tag{30}
$$

where the first inequality uses $\mathbb{E}[\boldsymbol{\xi}_t \boldsymbol{\xi}_t^\top] \preceq \widetilde{\sigma}_\xi^2 \boldsymbol{\Sigma}_{\boldsymbol{\nu}}$.

**Part 1 of Lemma D.6.** Comtinuing the calculation in Eq. (30), we have

$$
\mathbb{E}[\|\boldsymbol{\Sigma}_{\boldsymbol{\nu}}^{u/2}\widetilde{\boldsymbol{\mu}}_L\|_2^2]
$$

$$
\leqslant \widetilde{\sigma}_\xi^2 \cdot \operatorname{tr}\left[\sum_{k=0}^{\lfloor \log L \rfloor - 2} \frac{\gamma_{L_{\texttt{eff}}k+1}}{(2\gamma_{L_{\texttt{eff}}(k+1)+1}L_{\texttt{eff}})^u} \cdot (\mathbf{I} - (\mathbf{I} - \gamma_{L_{\texttt{eff}}k+1}\boldsymbol{\Sigma}_{\boldsymbol{\nu}})^{2L_{\texttt{eff}}}) + \frac{\gamma_0}{L}\boldsymbol{\Sigma}_{\boldsymbol{\nu}} \cdot (\mathbf{I} - (\mathbf{I} - \gamma_0 \boldsymbol{\Sigma}_{\boldsymbol{\nu}}/L)^{2L_{\texttt{eff}}})\right]
$$

$$
\lesssim \widetilde{\sigma}_\xi^2 \cdot \left[\sum_{k=0}^{\lfloor \log L \rfloor - 2} \frac{\gamma_{L_{\texttt{eff}}k+1}^{1-u}}{L_{\texttt{eff}}^u} \cdot \operatorname{tr}((\gamma_{L_{\texttt{eff}}k+1}L_{\texttt{eff}}\boldsymbol{\Sigma}_{\boldsymbol{\nu}})^{1/\alpha}) + \frac{\gamma_0}{L} \cdot \operatorname{tr}((\gamma_0 \boldsymbol{\Sigma}_{\boldsymbol{\nu}})^{1/\alpha})\right]
$$

$$\lesssim \widetilde{\sigma}_\xi^2 \cdot \gamma_0^{1-u+1/\alpha} \operatorname{tr}(\boldsymbol{\Sigma}_{\boldsymbol{\nu}}^{1/\alpha}) L_{\texttt{eff}}^{1/\alpha-u} \lesssim \widetilde{\sigma}_\xi^2 \cdot \gamma_0 \operatorname{tr}(\boldsymbol{\Sigma}_{\boldsymbol{\nu}}^{1/\alpha})(L_{\texttt{eff}}\gamma_0)^{1/\alpha-u},$$

where the first inequality uses $\sup_{x\in[0,1/\gamma_0]} x^u(1-\gamma_0 x)^{2L_{\texttt{eff}}} \leqslant 1/[2\gamma_0 L_{\texttt{eff}}]^u$ for any $u \in [0,1]$, the second inequality follows from $1-(1-\gamma_0 x)^{2L_{\texttt{eff}}} \leqslant (2\gamma_0 L_{\texttt{eff}} x)^{1/\alpha}$ for any $\alpha > 1$ and $x \in [0,1/\gamma_0]$ by Bernoulli's inequality, and the last inequality follows from the stepsize definition (2). This gives the first part of Lemma D.6.

**Part 2 of Lemma D.6.** Similarly, continuing the calculation in Eq. (30) and noting that $\sup_{x\in[0,1/\gamma]}[1-(1-\gamma x)^{2L_{\texttt{eff}}}]/x \leqslant 2\gamma L_{\texttt{eff}}$, we obtain

$$\mathbb{E}[\|\boldsymbol{\Sigma}_{\boldsymbol{\nu}}^{u/2}\widetilde{\boldsymbol{\mu}}_L\|_2^2] \leqslant \widetilde{\sigma}_\xi^2 \cdot \operatorname{tr}\Big[ \sum_{k=0}^{\lfloor \log L\rfloor-2} 2L_{\texttt{eff}}\gamma_{L_{\texttt{eff}}k+1}^2 \boldsymbol{\Sigma}_{\boldsymbol{\nu}}^{1+u}\cdot (\mathbf{I}-\gamma_{L_{\texttt{eff}}(k+1)+1}\boldsymbol{\Sigma}_{\boldsymbol{\nu}})^{2L_{\texttt{eff}}} + \frac{2\gamma_0^2}{L}\boldsymbol{\Sigma}_{\boldsymbol{\nu}}^{1+u}\Big]. \tag{31}$$

Denote the eigenvalues of $\boldsymbol{\Sigma}_{\boldsymbol{\nu}}$ by $\widehat{\lambda}_1 \geqslant \widehat{\lambda}_2 \geqslant \ldots \geqslant \widehat{\lambda}_N$ (let $\widehat{\lambda}_j = 0$ for $j > M$). Choose $\iota = (L_{\texttt{eff}}\gamma)^{1/a}$. When $M \leqslant N/\tilde{c}$, we have $\mu_j(\boldsymbol{\Sigma}_{\boldsymbol{\nu}}) \asymp j^{-a}$ for $j \leqslant M$ and otherwise 0. When $M > N/\tilde{c}$, we have $\iota \leqslant N/\tilde{c}$, $\widehat{\lambda}_j \asymp j^{-a}$ for $j \leqslant N/\tilde{c}$ and otherwise $\widehat{\lambda}_j \lesssim j^{-a}$ by monotonicity of $\widehat{\lambda}_j$. In both cases, we have $\widehat{\lambda}_j \asymp j^{-a}$ (or $\widehat{\lambda}_j = 0$) for $j \leqslant \iota$ and $\widehat{\lambda}_j \lesssim j^{-a}$ for $j > \iota$.

Since $f(x) > f(0)$ for any $x \in [0,1/\tilde{\gamma}]$ and $f(x) = x^{1+u}(1-\tilde{\gamma}x/2)^{2L_{\texttt{eff}}}$ for any $u \in [0,1]$, to obtain an upper bound on $\operatorname{tr}(\boldsymbol{\Sigma}_{\boldsymbol{\nu}}^{1+u}\cdot(\mathbf{I}-\tilde{\gamma}\boldsymbol{\Sigma}_{\boldsymbol{\nu}}/2)^{2L_{\texttt{eff}}})$, we can w.l.o.g. assume $\widehat{\lambda}_j \asymp j^{-a}$ for $j \leqslant \iota$ and $\widehat{\lambda}_j \lesssim j^{-a}$ for $j > \iota$. Under this assumption, for any $\tilde{\gamma} \in [0,1/(4\widehat{\lambda}_1)]$,

$$\operatorname{tr}(\boldsymbol{\Sigma}_{\boldsymbol{\nu}}^{1+u}\cdot(\mathbf{I}-\tilde{\gamma}\boldsymbol{\Sigma}_{\boldsymbol{\nu}}/2)^{2L_{\texttt{eff}}})$$

$$\lesssim \sum_{j=1}^N \widehat{\lambda}_j^{1+u}\cdot(\mathbf{I}-\tilde{\gamma}\widehat{\lambda}_j/2)^{2L_{\texttt{eff}}}$$

$$\lesssim \sum_{j>\iota} \widehat{\lambda}_j^{1+u}\cdot(\mathbf{I}-\tilde{\gamma}\widehat{\lambda}_j/2)^{2L_{\texttt{eff}}} + \sum_{k=0}^\infty \sum_{j\in[\iota/2^{k+1},\iota/2^k)} \widehat{\lambda}_j^{1+u}\cdot(\mathbf{I}-\tilde{\gamma}\widehat{\lambda}_j/2)^{2L_{\texttt{eff}}}$$

$$\lesssim \iota^{1-(1+u)a} + \sum_{k=0}^{\lfloor \log \iota\rfloor+1} \sum_{j\in[\iota/2^{k+1},\iota/2^k)} \widehat{\lambda}_j^{1+u}\cdot(1-\frac{2^{ka}}{2L_{\texttt{eff}}})^{2L_{\texttt{eff}}}$$

$$\lesssim \iota^{1-(1+u)a} + \sum_{k=0}^\infty (\iota/2^k)^{1-(1+u)a}\cdot e^{-2^{ka}} \lesssim \iota^{1-(1+u)a}\cdot(1+\sum_{k=0}^\infty 2^{k((1+u)a-1)}e^{-2^{ka}})$$

$$\lesssim \iota^{1-(1+u)a} = (L_{\texttt{eff}}\tilde{\gamma})^{1-(1+u)a}, \tag{32}$$

and $\gamma_0^2\operatorname{tr}(\boldsymbol{\Sigma}_{\boldsymbol{\nu}}^{1+u})/L \lesssim \gamma^2/L_{\texttt{eff}}$. Substituting these into Eq. (31) yields

$$\mathbb{E}[\|\boldsymbol{\Sigma}_{\boldsymbol{\nu}}^{u/2}\widetilde{\boldsymbol{\mu}}_L\|_2^2] \lesssim \widetilde{\sigma}_\xi^2 \cdot \Big[ \sum_{k=0}^{\lfloor \log L\rfloor-2} L_{\texttt{eff}}\gamma_{L_{\texttt{eff}}k+1}^2\cdot(L_{\texttt{eff}}\gamma_{L_{\texttt{eff}}k+1})^{1/a-u-1} + \gamma_0^2/L_{\texttt{eff}}\Big]$$

$$\lesssim \widetilde{\sigma}_\xi^2 \cdot \gamma_0 \cdot (L_{\texttt{eff}}\gamma_0)^{1/a-u}.$$

### D.4.2 Proof of Lemma D.7

We prove this lemma by induction. When $k = 0$, we have $\mathbb{E}[\boldsymbol{\xi}_t^0\boldsymbol{\xi}_t^{0\top}] = \mathbb{E}[\boldsymbol{\xi}_t\boldsymbol{\xi}_t^\top] \preceq \sigma_\xi^2\boldsymbol{\Sigma}_{\boldsymbol{\nu}}$ and

$$\mathbb{E}[\widetilde{\boldsymbol{\mu}}_t^0\widetilde{\boldsymbol{\mu}}_t^{0\top}] = \sum_{i=1}^t \gamma_i^2 \prod_{j=i+1}^t (\mathbf{I}-\gamma_j\boldsymbol{\Sigma}_{\boldsymbol{\nu}})\mathbb{E}[\boldsymbol{\xi}_i\boldsymbol{\xi}_i^\top] \prod_{j=i+1}^t (\mathbf{I}-\gamma_j\boldsymbol{\Sigma}_{\boldsymbol{\nu}})$$

$$\preceq \sigma_\xi^2\gamma_0 \sum_{i=1}^t \gamma_t \cdot \prod_{j=i+1}^t (\mathbf{I}-\gamma_j\boldsymbol{\Sigma}_{\boldsymbol{\nu}})\boldsymbol{\Sigma}_{\boldsymbol{\nu}} \preceq \sigma_\xi^2\gamma_0\Big(\mathbf{I}-\prod_{i=1}^t(\mathbf{I}-\gamma_i\boldsymbol{\Sigma}_{\boldsymbol{\nu}})\Big) \lesssim \sigma_\xi^2\gamma_0\mathbf{I}.$$

Now, assume the lemma holds for some $k \geqslant 0$, we show that it also holds for $k + 1$. For $\boldsymbol{\xi}_t^{k+1}$, we have

$$\mathbb{E}[\boldsymbol{\xi}_t^{k+1}\boldsymbol{\xi}_t^{k+1\top}] \preceq \mathbb{E}[(\boldsymbol{\Sigma}_{\boldsymbol{\nu}}-\boldsymbol{\nu}_t\boldsymbol{\nu}_t^\top)\mathbb{E}[\widetilde{\boldsymbol{\mu}}_{t-1}^k\widetilde{\boldsymbol{\mu}}_{t-1}^{k\top}](\boldsymbol{\Sigma}_{\boldsymbol{\nu}}-\boldsymbol{\nu}_t\boldsymbol{\nu}_t^\top)] \preceq \sigma_\xi^2\gamma_0^{k+1}B_{\boldsymbol{\nu}}^{2k}\cdot\mathbb{E}[(\boldsymbol{\Sigma}_{\boldsymbol{\nu}}-\boldsymbol{\nu}_t\boldsymbol{\nu}_t^\top)^2]$$

$$\leqslant \sigma_\xi^2 \gamma_0^{k+1} B_\nu^{2k} \cdot \mathbb{E}[\nu_t \nu_t^\top \nu_t \nu_t^\top] \leqslant \sigma_\xi^2 \gamma_0^{k+1} B_\nu^{2(k+1)} \cdot \Sigma_\nu.$$

For $\widetilde{\mu}_t^{k+1}$, we have

$$\mathbb{E}[\widetilde{\mu}_t^{k+1} \widetilde{\mu}_t^{k+1\top}] = \sum_{i=1}^t \gamma_i^2 \prod_{j=i+1}^t (\mathbf{I} - \gamma_j \Sigma_\nu) \mathbb{E}[\xi_i^{k+1} \xi_i^{k+1\top}] \prod_{j=i+1}^t (\mathbf{I} - \gamma_j \Sigma_\nu)$$

$$\leqslant \sigma_\xi^2 \gamma_0^{k+2} B_\nu^{2(k+1)} \sum_{i=1}^t \gamma_t \cdot \prod_{j=i+1}^t (\mathbf{I} - \gamma_j \Sigma_\nu) \Sigma_\nu \leqslant \sigma_\xi^2 \gamma_0^{k+2} B_\nu^{2(k+1)} \cdot \mathbf{I}.$$

This completes the induction.

### D.4.3 Proof of Lemma D.8

By definition of $\widehat{\mu}_t$, we have

$$\mathbb{E}[\|\Sigma_\nu^{u/2} \widehat{\mu}_L\|_2^2] = \mathbb{E}[\|\Sigma_\nu^{u/2} \sum_{t=1}^L \gamma_t \cdot \prod_{i=t+1}^L (\mathbf{I} - \gamma_i \nu_i \nu_i^\top) \widehat{\xi}_t\|_2^2]$$

$$= \sum_{t=1}^L \gamma_t^2 \cdot \operatorname{tr}\left(\mathbb{E}\left[\Sigma_\nu^u \prod_{i=t+1}^L (\mathbf{I} - \gamma_i \nu_i \nu_i^\top) \widehat{\xi}_t \widehat{\xi}_t^\top \prod_{i=t+1}^L (\mathbf{I} - \gamma_i \nu_i \nu_i^\top)\right]\right)$$

$$\leqslant \widehat{\sigma}_\xi^2 \sum_{t=1}^L \gamma_t^2 \cdot \operatorname{tr}\left(\Sigma_\nu^u \prod_{i=t+1}^L (\mathbf{I} - \gamma_i \nu_i \nu_i^\top) \Sigma_\nu \prod_{i=t+1}^L (\mathbf{I} - \gamma_i \nu_i \nu_i^\top)\right)$$

$$\leqslant \widehat{\sigma}_\xi^2 \cdot \sum_{t=1}^L \gamma_t^2 \operatorname{tr}(\Sigma_\nu) \|\Sigma_\nu\|_2^u \leqslant 2\widehat{\sigma}_\xi^2 B_\nu^{2u} \cdot \gamma_0^2 \operatorname{tr}(\Sigma_\nu) L_{\texttt{eff}},$$

where the last inequality follows since $\Sigma_\nu^2 \leqslant \mathbb{E}[\nu_t \nu_t^\top \nu_t \nu_t^\top] \leqslant B_\nu^2 \Sigma_\nu$ and $\sum_{t=1}^L \gamma_t^2 \leqslant 2 L_{\texttt{eff}} \gamma_0^2$.

### D.5 Proof of Lemma D.3

Let $\Delta_t^{(-i)} := \theta_t - \theta_t^{(-i)}$. For any $i \in [N], t \in [L]$, we have

$$(\mathbf{x}_i^\top \mathbf{S}^\top \theta_t)^2 \lesssim 2(\mathbf{x}_i^\top \mathbf{S}^\top \theta_t^{(-i)})^2 + 2(\mathbf{x}_i^\top \mathbf{S}^\top \Delta_t^{(-i)})^2 \leqslant 2(\mathbf{x}_i^\top \mathbf{S}^\top \theta_t^{(-i)})^2 + 2\|\mathbf{x}_i^\top \mathbf{S}^\top\|_{\Sigma^{-s}}^2 \cdot \|\Delta_t^{(-i)}\|_{\Sigma^s}^2$$

$$\lesssim \max_{i \in [N], t \in [L]} (\mathbf{x}_i^\top \mathbf{S}^\top \theta_t^{(-i)})^2 + \max_{i \in [N]} \|\mathbf{x}_i^\top \mathbf{S}^\top\|_{\Sigma^{-s}}^2 \cdot \max_{i \in [N], t \in [L]} \|\Delta_t^{(-i)}\|_{\Sigma^s}^2$$

It remains to bound $\max_{i \in [N], t \in [L]} \|\Delta_t^{(-i)}\|_{\Sigma^s}^2$. Adopt the shorthand notation $a_t^{(-i)} := y_i + \mathbf{x}_i^\top \mathbf{S}^\top \theta_{t-1}^{(-i)}$ and recell $a_{\max} = \max_{i \in [N], t \in [L]} |a_t^{(-i)}|$. By taking the difference between the (GD) process and the (LOO-GD) process, we have

$$\Delta_t^{(-i)} = (\mathbf{I} - \gamma_t \widehat{\Sigma}) \Delta_{t-1}^{(-i)} + \frac{\gamma_t a_t^{(-i)}}{N} \mathbf{S} \mathbf{x}_i = \underbrace{\left[\sum_{i=1}^t \frac{\gamma_i a_t^{(-i)}}{N} \cdot \prod_{j=i+1}^t (\mathbf{I} - \gamma_j \widehat{\Sigma})\right]}_{:= \mathsf{V}_{i,t}} \mathbf{S} \mathbf{x}_i.$$

Therefore, for $\lambda = 1/(L_{\texttt{eff}} \gamma)$,

$$\|\Delta_t^{(-i)}\|_{\Sigma^s}^2 = \operatorname{tr}(\mathbf{x}_i^\top \mathbf{S}^\top \mathsf{V}_{i,t} \Sigma^s \mathsf{V}_{i,t}^\top \mathbf{S} \mathbf{x}_i)$$

$$= \operatorname{tr}(\mathbf{x}_i^\top \mathbf{S}^\top \mathsf{V}_{i,t} \Sigma^s [\Sigma + \lambda \mathbf{I}]^{1/2} [\Sigma + \lambda \mathbf{I}]^{-1/2} \Sigma^s [\Sigma + \lambda \mathbf{I}]^{-1/2} [\Sigma + \lambda \mathbf{I}]^{1/2} \mathsf{V}_{i,t} \mathbf{S} \mathbf{x}_i)$$

$$\leqslant \sup_{x \geqslant 0} \frac{x^s}{x + \lambda} \cdot \operatorname{tr}(\mathbf{x}_i^\top \mathbf{S}^\top \mathsf{V}_{i,t} (\Sigma + \lambda \mathbf{I}) \mathsf{V}_{i,t} \mathbf{S} \mathbf{x}_i)$$

$$\lesssim \lambda^{s-1} \cdot \|(\widehat{\Sigma} + \lambda \mathbf{I})^{-1/2} (\Sigma + \lambda \mathbf{I})^{1/2}\|^2 \cdot \operatorname{tr}(\mathbf{x}_i^\top \mathbf{S}^\top \mathsf{V}_{i,t} (\widehat{\Sigma} + \lambda \mathbf{I}) \mathsf{V}_{i,t} \mathbf{S} \mathbf{x}_i).$$

Note that

$$\mathsf{V}_{i,t} (\widehat{\Sigma} + \lambda \mathbf{I}) \mathsf{V}_{i,t} \preceq \mathsf{V}_{\max} (\widehat{\Sigma} + \lambda \mathbf{I}) \mathsf{V}_{\max},$$

where

$$\mathsf{V}_{\max} := \sum_{i=1}^{t} \frac{\gamma_i a_{\max}}{N} \cdot \prod_{j=i+1}^{t} (\mathbf{I} - \gamma_j \widehat{\boldsymbol{\Sigma}}) = a_{\max} \cdot \frac{\mathbf{I} - \prod_{i=1}^{t}(\mathbf{I} - \gamma_i \widehat{\boldsymbol{\Sigma}})}{N\widehat{\boldsymbol{\Sigma}}}.$$

Adopt the shorthand notation $\mathsf{V}_t = \mathbf{I} - \prod_{i=1}^{t}(\mathbf{I} - \gamma_i \widehat{\boldsymbol{\Sigma}})$. Choosing $\lambda = 1/(L_{\texttt{eff}}\gamma)$ in the last display and taking the supremum over $t \in [L], i \in [N]$, we obtain

$$\max_{i \in [N], t \in [L]} \|\boldsymbol{\Delta}_t^{(-i)}\|_{\widehat{\boldsymbol{\Sigma}}^s}^2 \lesssim \frac{\|(\widehat{\boldsymbol{\Sigma}} + \lambda\mathbf{I})^{-1/2}(\boldsymbol{\Sigma} + \lambda\mathbf{I})^{1/2}\|^2}{(L_{\texttt{eff}}\gamma)^{s-1}} \cdot \frac{a_{\max}^2}{N^2} \cdot \mathrm{tr}(\mathbf{x}_i^\top \mathbf{S}^\top \widehat{\boldsymbol{\Sigma}}^{-1} \mathsf{V}_t (\widehat{\boldsymbol{\Sigma}} + \lambda\mathbf{I}) \mathsf{V}_t \widehat{\boldsymbol{\Sigma}}^{-1} \mathbf{S} \mathbf{x}_i)$$

$$\lesssim \frac{\|(\widehat{\boldsymbol{\Sigma}} + \lambda\mathbf{I})^{-1/2}(\boldsymbol{\Sigma} + \lambda\mathbf{I})^{1/2}\|^2}{(L_{\texttt{eff}}\gamma)^{s-1}} \cdot \frac{a_{\max}^2}{N^2} \cdot \max_{i \in [N]} \|\mathbf{S}\mathbf{x}_i\|_2^2 \cdot \|\widehat{\boldsymbol{\Sigma}}^{-1} \mathsf{V}_t (\widehat{\boldsymbol{\Sigma}} + \lambda\mathbf{I}) \mathsf{V}_t \widehat{\boldsymbol{\Sigma}}^{-1}\|_2.$$

Moreover, we have

$$\|\widehat{\boldsymbol{\Sigma}}^{-1} \mathsf{V}_t (\widehat{\boldsymbol{\Sigma}} + \lambda\mathbf{I}) \mathsf{V}_t \widehat{\boldsymbol{\Sigma}}^{-1}\| \leqslant \|\mathsf{V}_t^2 \widehat{\boldsymbol{\Sigma}}^{-1}\| + \lambda \cdot \|\widehat{\boldsymbol{\Sigma}}^{-1} \mathsf{V}_t\|^2 \overset{(i)}{\leqslant} L_{\texttt{eff}}\gamma,$$

where step (i) follows from $\|\mathsf{V}_t\| \leqslant 1$ and $\sup_{x \in [0, 1/\gamma]}(1 - \prod_{i=1}^{t}(1 - \gamma_i x)) \lesssim L_{\texttt{eff}}\gamma$ by the stepsize definition (2). Combining the last two displays, we find

$$\max_{i \in [N], t \in [L]} \|\boldsymbol{\Delta}_t^{(-i)}\|_{\widehat{\boldsymbol{\Sigma}}^s}^2 \lesssim a_{\max}^2 \cdot \max_{i \in [N]} \|\mathbf{S}\mathbf{x}_i\|_2^2 \cdot \|(\widehat{\boldsymbol{\Sigma}} + \lambda\mathbf{I})^{-1/2}(\boldsymbol{\Sigma} + \lambda\mathbf{I})^{1/2}\|^2 \cdot \frac{(L_{\texttt{eff}}\gamma)^{2-s}}{N^2}.$$

This completes the proof.

# E Auxiliary lemmas

In this section, we provide some auxiliary lemmas that are used in the proofs.

## E.1 General concentration bounds

**Lemma E.1.** *Let $\nu_1, \nu_2, \ldots, \nu_N$ be i.i.d. samples from $\mathcal{N}(0, \Sigma)$ for some $\Sigma \in \mathbb{R}^{p \times p}$. Let $\widehat{\Sigma} = \sum_{i=1}^N \nu_i \nu_i^\top / N$. Assume that $\sum_{i=1}^p \frac{\mu_i(\Sigma)}{\mu_i(\Sigma) + \lambda} \leqslant N/4$. Then with probability at least $1 - e^{-\Omega(N)}$*

$$\|(\widehat{\Sigma} + \lambda \mathbf{I}_p)^{-1/2} \Sigma^{1/2}\|_2 \leqslant \|(\widehat{\Sigma} + \lambda \mathbf{I}_p)^{-1/2} (\Sigma + \lambda \mathbf{I}_p)^{1/2}\|_2 \leqslant 3.$$

*Moreover, the expectation $\mathbb{E}\|(\widehat{\Sigma} + \lambda \mathbf{I}_p)^{-1/2} (\Sigma + \lambda \mathbf{I}_p)^{1/2}\|_2^4 \leqslant 100 + \exp(-cN)\|\Sigma\|_2^2 / \lambda^2$ for some constant $c > 0$.*

*Proof of Lemma E.1.* Adopt the shorthand notation $\Sigma_\lambda = \Sigma + \lambda \mathbf{I}_p, \widehat{\Sigma}_\lambda = \widehat{\Sigma} + \lambda \mathbf{I}_p$. By some basic algebra, we have

$$\|(\widehat{\Sigma} + \lambda \mathbf{I}_p)^{-1/2} \Sigma^{1/2}\|_2^2 \leqslant \|(\widehat{\Sigma} + \lambda \mathbf{I}_p)^{-1/2} (\Sigma + \lambda \mathbf{I}_p)^{1/2}\|_2^2 = \|\Sigma_\lambda^{1/2} \widehat{\Sigma}_\lambda^{-1} \Sigma_\lambda^{1/2}\|_2$$
$$= \|(\mathbf{I}_p - \Sigma_\lambda^{-1/2}(\Sigma - \widehat{\Sigma})\Sigma_\lambda^{-1/2})^{-1}\|_2. \tag{33}$$

Let $B = \Sigma_\lambda^{-1/2} \Sigma^{1/2}$. Then we have $\|B\|_2 \leqslant 1$ and $\operatorname{tr}(BB^\top) = \sum_{i=1}^p \frac{\mu_i(\Sigma)}{\mu_i(\Sigma) + \lambda} \leqslant N/4$ by assumption. Therefore, by Theorem 4 and 5 in Koltchinskii and Lounici (2017)

$$\|\Sigma_\lambda^{-1/2}(\Sigma - \widehat{\Sigma})\Sigma_\lambda^{-1/2}\|_2 \leqslant \|B\|_2^2 \cdot \max\left\{\sqrt{\frac{\operatorname{tr}(BB^\top)}{N}}, \frac{\operatorname{tr}(BB^\top)}{N}\right\} + c\sqrt{\frac{t}{N}} \cdot \|B\|_2^2$$
$$\leqslant \sqrt{\frac{\operatorname{tr}(BB^\top)}{N}} + c\sqrt{\frac{t}{N}} \leqslant \frac{1}{2} + c\sqrt{\frac{t}{N}}.$$

with probability at least $1 - e^{-t}$ for any $t \in [1, N]$. Choosing $t = N/c'$ for some sufficiently large constant $c' > 0$ yields $\|\Sigma_\lambda^{-1/2}(\Sigma - \widehat{\Sigma})\Sigma_\lambda^{-1/2}\|_2 \leqslant 2/3$ with probability at least $1 - e^{-\Omega(N)}$. Combining this with Eq. (33) yields the first part of Lemma E.1.

To establish the bound in expectation, we first use Eq. (33) to obtain an always-valid upper bound

$$\|(\widehat{\Sigma} + \lambda \mathbf{I}_p)^{-1/2} (\Sigma + \lambda \mathbf{I}_p)^{1/2}\|_2^2 \leqslant \frac{1}{\mu_{\min}(\mathbf{I}_p - \Sigma_\lambda^{-1/2}\Sigma\Sigma_\lambda^{-1/2})} = \frac{\lambda + \|\Sigma\|_2}{\lambda}.$$

Combining this with the first part of Lemma E.1, we obtain

$$\mathbb{E}\|(\widehat{\Sigma} + \lambda \mathbf{I}_p)^{-1/2} (\Sigma + \lambda \mathbf{I}_p)^{1/2}\|_2^4 \leqslant 100 + \frac{\exp(-cN)}{\lambda^2} \cdot \|\Sigma\|_2^2$$

for some constant $c > 0$. □

In the next three lemmas, we let $(\lambda_i)_{i=1}^d$ denote the eigenvalues of $\mathbf{H}$ in non-increasing order.

**Lemma E.2** (Lemma G.1 in Lin et al. (2024)). *Let $\mathbf{S} \in \mathbb{R}^{M \times d}$ be a random sketching matrix with i.i.d. entries $\mathbf{S}_{ij} \sim \mathcal{N}(0, 1/M)$.[5] Then there exists some absolute constant $c > 1$ such that for any $M \geqslant 1$ and $0 \leqslant k \leqslant M$, with probability at least $1 - e^{-\Omega(M)} - e^{-\Omega(k)}$, we have*

$$\text{for every } j \leqslant M, \quad \left|\tilde{\lambda}_j - \left(\lambda_j + \frac{\sum_{i>k}\lambda_i}{M}\right)\right| \leqslant c\left(\sqrt{\frac{k}{M}}\lambda_j + \lambda_{k+1} + \sqrt{\frac{\sum_{i>k}\lambda_i^2}{M}}\right).$$

*Consequently, if $k \leqslant M/c^2$, then*

$$\text{for every } j \leqslant M, \quad \left|\tilde{\lambda}_j - \left(\lambda_j + \frac{\sum_{i>k}\lambda_i}{M}\right)\right| \leqslant \frac{1}{2}\left(\lambda_j + \frac{\sum_{i>k}\lambda_i}{M}\right) + c_1\lambda_{k+1},$$

*where $c_1 = c + 2c^2$.*

---

[5]$d$ can be $+\infty$.

**Lemma E.3** (Tail concentration; Lemma G.2 in Lin et al. (2024) and Lemma 26 in Bartlett et al. (2020)). *Let $\mathbf{S} \in \mathbb{R}^{M \times d}$ be a random sketching matrix with i.i.d. entries $\mathbf{S}_{ij} \sim \mathcal{N}(0, 1/M)$. For any $k \geqslant 0$, with probability at least $1 - \delta$, we have*

$$\left\| \mathbf{S}_{k:\infty} \mathbf{H}_{k:\infty} \mathbf{S}_{k:\infty}^\top - \frac{\sum_{i>k} \lambda_i}{M} \cdot \mathbf{I}_M \right\|_2 \lesssim \frac{1}{M} \left( \lambda_{k+1} \left( M + \log \frac{1}{\delta} \right) + \sqrt{\sum_{i>k} \lambda_i^2 \left( M + \log \frac{1}{\delta} \right)} \right).$$

*In particular, with probability at least $1 - e^{-\Omega(M)}$, we have*

$$\left\| \mathbf{S}_{k:\infty} \mathbf{H}_{k:\infty} \mathbf{S}_{k:\infty}^\top - \frac{\sum_{i>k} \lambda_i}{M} \cdot \mathbf{I}_M \right\|_2 \lesssim \lambda_{k+1} + \sqrt{\frac{\sum_{i>k} \lambda_i^2}{M}}.$$

*Furthermore, the minimum eigenvalue of $\mathbf{S}_{k:\infty} \mathbf{H}_{k:\infty} \mathbf{S}_{k:\infty}^\top$ satisfies*

$$\mu_{\min} \left( \mathbf{S}_{k:\infty} \mathbf{H}_{k:\infty} \mathbf{S}_{k:\infty}^\top \right) \gtrsim \lambda_{k+2M}$$

*with probability at least $1 - e^{-\Omega(M)}$.*

**Lemma E.4** (Head concentration; Lemma G.3 in Lin et al. (2024)). *Let $\mathbf{S} \in \mathbb{R}^{M \times d}$ be a random sketching matrix with i.i.d. entries $\mathbf{S}_{ij} \sim \mathcal{N}(0, 1/M)$. For any $k \geqslant 1$, with probability at least $1 - \delta$, we have*

$$\text{for every } j \leqslant k, \quad \left| \mu_j \left( \mathbf{S}_{0:k} \mathbf{H}_{0:k} \mathbf{S}_{0:k}^\top \right) - \lambda_j \right| \lesssim \sqrt{\frac{k + \log(1/\delta)}{M}} \lambda_j.$$

*In particular, with probability at least $1 - e^{-\Omega(k)}$,*

$$\text{for every } j \leqslant k, \quad \left| \mu_j \left( \mathbf{S}_{0:k} \mathbf{H}_{0:k} \mathbf{S}_{0:k}^\top \right) - \lambda_j \right| \lesssim \sqrt{\frac{k}{M}} \lambda_j.$$

### E.2   Concentration bounds under power-law spectrum

**Lemma E.5** (Eigenvalues of $\mathbf{SHS}^\top$ under power-law spectrum; Lemma G.4 in Lin et al. (2024)). *Let Assumption 1C hold. There exist some $a$-dependent constants $c_2 > c_1 > 0$ such that*

$$c_1 j^{-a} \leqslant \mu_j(\mathbf{SHS}^\top) \leqslant c_2 j^{-a}$$

*with probability at least $1 - e^{-\Omega(M)}$.*

**Lemma E.6** (Ratio of eigenvalues of $\mathbf{S}_{k:\infty} \mathbf{H}_{k:\infty} \mathbf{S}_{k:\infty}^\top$ under power-law spectrum; Lemma G.5 in Lin et al. (2024)). *Let Assumption 1C hold. There exists some $a$-dependent constant $c > 0$ such that for any $k \geqslant 0$, the ratio between the $M/2$-th and $M$-th eigenvalues of $\mathbf{S}_{k:\infty} \mathbf{H}_{k:\infty} \mathbf{S}_{k:\infty}^\top$ satisfies*

$$\frac{\mu_{M/2}(\mathbf{S}_{k:\infty} \mathbf{H}_{k:\infty} \mathbf{S}_{k:\infty}^\top)}{\mu_M(\mathbf{S}_{k:\infty} \mathbf{H}_{k:\infty} \mathbf{S}_{k:\infty}^\top)} \leqslant c$$

*with probability at least $1 - e^{-\Omega(M)}$.*

**Lemma E.7** (Bounds on Approx under the source condition; Lemma C.5 in Lin et al. (2024)). *Suppose Assumption 1 is in force. Then with probability at least $1 - e^{-\Omega(M)}$ over $\mathbf{S}$,*

$$M^{1-b} \lesssim \mathbb{E}_{\mathbf{w}*}[\mathsf{Approx}] \lesssim M^{1-b}.$$

*Here, the hidden constants only depend on $(a, b)$ in Assumption 1.*

**Lemma E.8** (Eigenvalues of $\widehat{\mathbf{\Sigma}}$ under power-law spectrum). *Suppose $\mathbf{\Sigma} = \mathbf{SHS}^\top$ satisfies $\mu_j(\mathbf{\Sigma}) \asymp j^{-a}$ for $j \in [M]$. Then for some $a$-dependent constants $c, c_1, c_2 > 0$, $\widehat{\mathbf{\Sigma}} = \frac{1}{N} \sum_{i=1}^N \mathbf{S} \mathbf{x}_i \mathbf{x}_i^\top \mathbf{S}^\top$ satisfies*

$$c_1 j^{-a} \leqslant \mu_j(\widehat{\mathbf{\Sigma}}) \leqslant c_2 j^{-a} \text{ for all } j \leqslant \min\{M, N/c\}, \quad \text{and}$$

$$\mu_j(\widehat{\mathbf{\Sigma}}) \leqslant c_2 j^{-a} \text{ for all } j \in (\min\{M, N/c\}, \min\{M, N\}]$$

*with probability at least $1 - e^{-\Omega(N)}$ over the randomness of $(\mathbf{x}_i)_{i=1}^N$ conditioned on $\mathbf{S}$.*

*Proof of Lemma E.8.* Note that $\mathbf{SX}^\top/\sqrt{N} \overset{d}{=} \mathbf{\Sigma}^{1/2}\mathbf{Z}^\top$, where $\mathbf{Z} \in \mathbb{R}^{M \times N}$ has i.i.d. entries $\mathbf{Z}_{ij} \sim \mathcal{N}(0, 1/N)$ conditioned on $\mathbf{S}$. Thus, $\mu_j(\widehat{\mathbf{\Sigma}}) = \mu_j(\mathbf{Z\Sigma Z}^\top)$ for $j \leqslant \min\{M, N\}$. Let $(\widehat{\lambda}_i)_{i=1}^N$ denote the eigenvalues of $\mathbf{Z\Sigma Z}^\top$ in non-increasing order. Using Lemma E.2 with $k = N/c$ for some sufficiently large constant $c$ and noting that $\sum_{i>k} i^{-a} \lesssim k^{1-a}$, we have

$$\frac{1}{2} \cdot (j^{-a} + \tilde{c}_1 N^{-a}) - \tilde{c}_2 \cdot N^{-a} \leqslant \widehat{\lambda}_j \leqslant \frac{3}{2} \cdot (j^{-a} + \tilde{c}_1 N^{-a}) + \tilde{c}_2 \cdot N^{-a}$$

for every $j \leqslant \min\{M, N/c\}$ for some constants $\tilde{c}_i, i \in [2]$ with probability at least $1 - e^{-\Omega(N)}$. Therefore, for all $j \leqslant \min\{M, N/\tilde{c}\}$ for some sufficiently large constant $\tilde{c} > 1$, we have

$$\widehat{\lambda}_j \in [\tilde{c}_3 j^{-a}, \tilde{c}_4 j^{-a}]$$

with probability at least $1 - e^{-\Omega(N)}$ for some constants $\tilde{c}_3, \tilde{c}_4 > 0$. For $j \in (\min\{M, N/\tilde{c}\}, \min\{M, N\}]$, by monotonicity of the eigenvalues, we have

$$\widehat{\lambda}_j \leqslant \widehat{\lambda}_{\lfloor \min\{M, N/\tilde{c}\}\rfloor} \leqslant \tilde{c}_4 \left(\left\lfloor \min\{M, N/\tilde{c}\}\right\rfloor\right)^{-a} \leqslant \tilde{c}_5 \min\{M, N\}^{-a} \leqslant \tilde{c}_5 j^{-a}$$

for some sufficiently large constant $\tilde{c}_5 > \tilde{c}_4$ with probability at least $1 - e^{-\Omega(N)}$. $\qquad\square$

