# OpenReview forum: "Improved Scaling Laws in Linear Regression via Data Reuse"
_NeurIPS.cc/2025/Conference — NeurIPS 2025 poster_

### Official Review · Reviewer_6YBu · 2025-06-18

**Clarity:** 3
**Significance:** 3
**Originality:** 2
**Rating:** 4
**Confidence:** 3

**Summary:**

In this paper, the authors study the behavior of multi-pass SGD on task of linear regression with sketching. They
assume that the inputs follow the Gaussian distribution $N(0, H)$ with the eigenvalues of $H$ following a power law with
exponent $a$, the ground truth direction is $w^\*$, and covariance matrix of the prior on $w^*$ share the eigenvectors
with $H$ and the eigenvalues follow a power law with exponent $b$. Under these assumptions, they show that the test
loss of multi-pass SGD satisfies the scaling law $M^{1-b} + L^{(1-b)/a}$ with $M$ being the sketching dimension and
$L \lesssim N^{a/b}$ the number of steps, while the scaling law of single-pass SGD is $M^{1-b} + N^{(1-b)/a}$, where
$N$ is the number of samples. Namely, multi-pass SGD is provably more sample-efficient than single-pass SGD (in this setting).

**Questions:**

See the weakness section. In particular, I'd like to know the following.
* Is it true that in the isotropic prior case, the bounds are essentially the same as the ones in Lin et al. (2024), or I am missing something here?
* What will happen if, instead of assuming the data distribution and the prior share the same eigenvectors, we only assume
  they are correlated?

**Ethical Concerns:**

["NO or VERY MINOR ethics concerns only"]

**Final Justification:**

My original concern was that the contribution of this paper over [1-2] seems to be small, but I was convinced during the rebuttal that this is not the case. In addition, the authors also clarified that certain strong conditions are not needed for some of their results. Therefore, I raised my score from 3 to 4.

**Limitations:**

Yes

**Quality:**

3

**Strengths And Weaknesses:**

### Strengths

* This is a well-written paper and is easy to follow. The setting and results are clearly stated and described.
* The analysis of multi-pass SGD is rather rare in this field, probably due to the additional dependence between iterations
  introduced by the reuse of samples (most of the existing results about GD or online/single-pass SGD). It is nice to
  have a result which proves that at least in certain scenarios, the reused samples behave similar to fresh samples and
  multi-pass SGD can beat one-pass SGD in terms of sample complexity.
* The authors also prove lower bounds on the errors, showing their bounds are not vacuous, and also verify their claims with
  experiments.

### Weaknesses

My main concern with this paper is that its contribution over [1-2] appears to be limited.

When the prior of the ground truth is isotropic, we have $a = b$, and in this case, the results (cf. Corollary 3.3) in this
paper do not give any improvement over the results in [1].

In the regime where improvement can be obtained, the authors need to assume that the data distribution and the prior of
the ground truth share the same eigenvectors and the eigenvalues of the prior follow another power law, which is a rather
strong assumption (especially the first part). Moreover, the benefit of reusing samples in linear regression has been
studied in [2] and the results/observations are similar. To be fair, [2] does not analyze the dependence on the model
size, but that part is covered in [1].


[1] Licong Lin, Jingfeng Wu, Sham M Kakade, Peter L Bartlett, and Jason D Lee. Scaling laws in linear regression: Compute, parameters, and data. 2024

[2] Loucas Pillaud-Vivien, Alessandro Rudi, Francis Bach. Statistical Optimality of Stochastic Gradient Descent on Hard Learning Problems through Multiple Passes. 2018

---

> ### Author Rebuttal · Authors · 2025-07-31
>
> We appreciate Reviewer 6YBu for their careful review and valuable feedback. We hope to address all questions below.
>
> ---
>
> > W1. My main concern with this paper is that its contribution over [1-2] appears to be limited.
>
> A1. We respectively disagree that our analysis has limited novelty.
> For multi-pass SGD, we generalize the results of Pillaud-Vivien et al. (2018) [2] in two key aspects.
> First, the upper bound on the risk of multi-pass SGD in Theorem 1 of [2] is restricted to the regime $\mu a \leq b \leq a$ for some $\mu \in (0,1]$. (It can be verified that our parameters $(a, b)$ equal  $(\alpha, 2\alpha r + 1)$ in  their work, and note that choosing $\mu = 0$ in their Assumption A3 leads to $\kappa = \infty$, making the bound vacuous.) In contrast, our upper bounds in Corollaries 3.3 and 3.4 apply to a strictly broader regime $a>b-1$.
> Second, [2] provides no lower bounds for the risk of multi-pass SGD. In contrast, we establish matching lower bounds in the regime $a>b-1$, verifying the tightness of our upper bounds. We believe these generalizations are not possible without introducing technical innovations. Note: the condition $b > a$ in lines 219–221 is a typo and should be revised to $b > a > b - 1$. Similarly, lines 129 and 169 should include the condition $a > b - 1$.
>
>
> Compared to the analysis in [1] for one-pass SGD, the non-commutativity between the empirical covariance and population covariance matrices introduces additional technical challenges, requiring new proof arguments.  For example, unlike for the bias upper bound (lines 267-270), the covariance replacement trick in Lemma 7 of [2] does not apply when establishing the bias lower bound (Lemma B.2). Instead, we derive the bound by showing that a specific function of the empirical covariance commutes with the population covariance in expectation.
>
> ---
>
> > Q1. Is it true that in the isotropic prior case, the bounds are essentially the same as the ones in Lin et al. (2024), or I am missing something here?
>
> A2.
> Yes, the bounds are the same in the regime where $a = b$. In fact, the rate  $O(N^{(1-a)/a})$ in this case   is minimax optimal for a class of linear regression problems with the same spectral conditions [2], and hence cannot be improved.
>
>
> ---
>
> > Q2. What will happen if, instead of assuming the data distribution and the prior share the same eigenvectors, we only assume they are correlated?
>
> A3. First, we emphasize that our general upper and lower bounds on the bias, variance, and fluctuation (Lemmas B.1, B.2, C.1, D.1, D.4) do not require either the power-law assumption or the alignment between the eigenvectors of the data distribution and the prior. These bounds hold under Gaussian design with an arbitrary data spectrum, as long as certain regularity conditions (Assumption 3) are satisfied.
>
> The alignment of eigenvectors is assumed only to derive exact rates from the general bounds under the power-law setting. In fact, our main results in Section 3 still hold when the prior covariance $\mathbf{H}^{\mathbf{w}} := \mathbb{E}[\mathbf{w}^* \mathbf{w}^{*\top}]$ is bounded above and below (in Loewner order, up to constant factors) by a matrix $\widetilde{\mathbf{H}}^{\mathbf{w}}$ that has a power-law spectrum and share the same eigenvectors as the data covariance $\mathbf{H}$ as in Assumption 1D.
>
> ---
> [1]. Lin, Licong, et al. "Scaling laws in linear regression: Compute, parameters, and data." Advances in Neural Information Processing Systems 37 (2024): 60556-60606.
>
> [2]. Pillaud-Vivien, Loucas, Alessandro Rudi, and Francis Bach. "Statistical optimality of stochastic gradient descent on hard learning problems through multiple passes." Advances in Neural Information Processing Systems 31 (2018).

---

> > ### Comment · Reviewer_6YBu · 2025-08-03
> >
> > Thank you for the clarification. Now I see the contribution of this paper over [1-2]. I'll raise my score to 4

---

### Official Review · Reviewer_b6Gz · 2025-06-23

**Clarity:** 3
**Significance:** 3
**Originality:** 3
**Rating:** 4
**Confidence:** 2

**Summary:**

The paper studies mulpi-pass SGD applied to linear regression. Under power-law spectrum assumptions of the features and the true model parameter, the authors prove scaling laws of the different components of the test error. The main contribution is extending the previous analysis to account for data reuse.

**Questions:**

1. Could you provide a high-level overview of how the fluctuation error is bounded?
2. Would the analysis/results extend to SGD with l2 regularization?

**Ethical Concerns:**

["NO or VERY MINOR ethics concerns only"]

**Final Justification:**

My previous concerns about clarity, originality, and significance are adequately addressed by the rebuttal. While this work focuses on the linear case, the upper bound can be extended to sub-Gaussian design. In addition, the lower bound is a novel contribution. I still believe the limitation to the linear case restricts the impact of works like this. I am maintaining my low confidence score since I am not super familiar with related works. Also, I did not verify the correctness of the proofs, but the agreement with experiments is a promising sign.

**Limitations:**

yes

**Paper Formatting Concerns:**

None.

**Quality:**

3

**Strengths And Weaknesses:**

- Quality: the submission appears technically sound. The theoretical results are accompanied by experiments. However, some proofs are a bit hard to follow and I am not able to verify their correctness.
- Clarity: the main paper is pretty clearly written. However, the proofs in the appendix sometimes are not that clear and detailed. E.g., from line 846-line 849, too many things are done at the same time. Also, nested lemmas (e.g., lemma D.2 and D.3 are stated in the proof of lemma D.1) makes the flow hard to follow. The clarity can be improved by providing a better high-level proof sketch of bounding the fluctuation error. (If I understand correctly, this is the key difference from the results in [1]).
- Significance: The submission advances our understanding of scaling laws under data-reuse. While this is somewhat significant, the main limitation is that the analysis is limited to linear models. In addition, the Gaussian design matrix is a pretty strong assumption.
- Originality: The submission extends previous results on scaling laws to multi-pass SGD. The proofs follow closely along the lines of [0].

Typos:
- Second line of equation after 521, missing a S?


[0] Loucas Pillaud-Vivien, Alessandro Rudi, and Francis Bach. Statistical optimality of stochastic gradient descent on hard learning problems through multiple passes
[1] Licong Lin, Jingfeng Wu, Sham M Kakade, Peter L Bartlett, and Jason D Lee. Scaling laws in linear regression: Compute, parameters, and data. arXiv preprint arXiv:2406.08466, 2024

---

> ### Author Rebuttal · Authors · 2025-07-31
>
> We  appreciate Reviewer b6Gz for their thorough review and constructive feedback on our work. Below, we hope to answer all questions raised in the review.
>
> ---
> > W1 and Q1. Clarity: the main paper is pretty clearly written. However, the proofs in the appendix sometimes are not that clear and detailed. E.g., from line 846-line 849, too many things are done at the same time. Also, nested lemmas (e.g., lemma D.2 and D.3 are stated in the proof of lemma D.1) makes the flow hard to follow. The clarity can be improved by providing a better high-level proof sketch of bounding the fluctuation error. (If I understand correctly, this is the key difference from the results in [1]).
>
> A1. We thank the reviewer for the constructive comments on improving the clarity of our proof. We will include additional details and provide a high-level proof sketch for bounding the fluctuation error in the paper. For completeness, we present the sketch here.
>
> To bound the fluctuation error, we express the difference between the multi-pass SGD and GD trajectories, $\mathbf{v}_t - \boldsymbol{\theta}_t$, as a stochastic process (Eq. 17) that fits into the framework of Lemma D.2, which provides a bound on the fluctuation error $\mathbb{E}[\\|\mathbf{\Sigma}^{1/2}(\mathbf{v}_t - \boldsymbol{\theta}_t)\\|^2]$ under certain conditions, up to a mismatch between $\widehat{\mathbf{\Sigma}}$ and $\mathbf{\Sigma}$. We verify that the required conditions hold with appropriate choices of parameters (Eq. 19), which are further bounded using a leave-one-out argument (Lemma D.3). Applying Lemma D.2 with these parameters and a covariance replacement trick (line 776) yields the bounds stated in Lemma D.1.
>
> Moreover, we emphasize that, in addition to bounding the fluctuation error, generalizing the one-pass SGD results from [1] to GD in our setting introduces additional challenges. Please see our response to your W3 below for further discussion.
>
> ---
>
> > W2. Significance: The submission advances our understanding of scaling laws under data-reuse. While this is somewhat significant, the main limitation is that the analysis is limited to linear models. In addition, the Gaussian design matrix is a pretty strong assumption.
>
> A2.  Extending the results to nonlinear settings such as logistic regression or two-layer neural networks would be an interesting direction for future work. We will comment on this in the paper.
>  That said, we believe our contribution, deriving finite-sample matching upper and lower bounds for multi-pass SGD in sketched linear models, has made substantial contributions to the theoretical understanding of scaling laws.
>
>
> In fact, the Gaussian design is not necessary for all the upper bounds in Section 3.
>  Namely, it can be shown that the Gaussian design assumption (Assumption 1A) can be relaxed to the following: $\mathbf{x}= \mathbf{H}^{1/2}\mathbf{\widetilde x}$, where $\mathbb{E}[\mathbf{\widetilde x}\mathbf{\widetilde x}^\top]=\mathbf{I}$, and the vector $\mathbf{\widetilde x}$ is symmetric and  1-sub-Gaussian, i.e.,  $\mathbf{\widetilde x}\overset{d}{=}-\mathbf{\widetilde x}$ and $\mathbb{E}[e^{\lambda\langle \mathbf{v},\mathbf{\widetilde x}\rangle}]\leq e^{\lambda^2/2}$ for any unit vector $\mathbf{v}$. In short, the relaxation can be made since the Gaussian assumption is mainly used to establish certain concentration bounds (e.g., Bernstein’s inequality), which also hold for sub-Gaussian vectors. On the other hand, for the lower bounds, the  Gaussian assumption is still required in order to establish the conditional independence in line 521 and 754.  We are happy to provide a more detailed explanation, should the reviewer have any concerns regarding the relaxation.
>
> In addition, we run simulations to verify the claim. We generate data $\mathbf{x}=(x_1,\ldots,x_d)^\top$ from the distribution where $x_i$ are independent and $\mathbb{P}(x_i=1)=\mathbb{P}(x_i=-1)\propto i^{a-b}, \mathbb{P}(x_i=0)=1-2\mathbb{P}(x_i=1)$ with $a=2,b=1.5$ instead of Gaussian. Under the same setting and choices of hyperparameters as in Figure 1(a), similar to the Gaussian case, we observe that excess test error of one-pass SGD and multi-pass SGD both exhibit power-law scaling in the number of effective steps ($L_{\mathsf{eff}})$. Moreover, the fitted slopes are both close to the theoretical prediction ($0.34\approx0.33=(1-b)/b$ for multi-pass SGD and  $0.26\approx0.25=(1-a)/b$ for one-pass SGD).
>
> > W3. Originality: The submission extends previous results on scaling laws to multi-pass SGD. The proofs follow closely along the lines of [0].
>
> A3. We respectively disagree that our analysis has limited originality.
> For multi-pass SGD, we extend the results in [0] in two key aspects.
> First, the upper bound on the risk of multi-pass SGD in Theorem 1 of [0] is restricted to the regime $\mu a \leq b \leq a$ for some $\mu \in (0,1]$. (It can be verified that our parameters $(a, b)$ equal  $(\alpha, 2\alpha r + 1)$ in [0], and note that choosing $\mu = 0$ in their Assumption A3 leads to $\kappa = \infty$, making the bound vacuous.) In contrast, our upper bounds in Corollaries 3.3 and 3.4 apply to a strictly broader regime $a>b-1$.
> Second, [0] provides no lower bounds for the risk of multi-pass SGD. In contrast, we establish matching lower bounds in the regime $a>b-1$, verifying the tightness of our upper bounds. We believe these generalizations are not possible without introducing technical innovations. Note: the condition $b > a$ in lines 219–221 is a typo and should be revised to $b > a > b - 1$. Similarly, lines 129 and 169 should include the condition $a > b - 1$.
>
>
>
> More specifically, as mentioned in our response to your Q1 above,  for the fluctuation error of multi-pass SGD, we follow the standard practice as in [0] and the early work by Aguech et al., 2000 [2]  to express the difference between the multi-pass SGD and GD trajectories as a stochastic process (Eq. 17). We then bound the fluctuation error through controlling the accumulated error of the stochastic process using Lemma D.2 and Lemma D.3, which involves a novel leave-one-out argument to control the model parameters.
>  Note that our Lemma D.2 also generalizes the corresponding results in [0], as the analysis in [0] does not apply when $a=b$ for infinite-dimensional problems. In addition, we derive a novel lower bound on the fluctuation error that matches the upper bound up to logarithmic factors in certain regimes (Lemma D.4 and D.5).
>
>
> ---
>
>
> >Q2. Would the analysis/results extend to SGD with l2 regularization?
>
> A4.
> This is an interesting question. We believe part of our results extend to multi-pass SGD with l2-regularization. We will comment on this as an interesting future direction. On the other hand, we would like to point out that, mult-pass SGD already provides sufficient implicit regularization, in which the achieved excess risk is minimax optimal for $a$ and $b$ such that $a>b>2$, (see [0]). Therefore, adding an l2-regularization might not provide additional benefits.
>
> ---
>
> > Other: typos: Second line of equation after 521, missing a S?
>
> Thanks for pointing this out! We have fixed it.
>
> ---
>
>
>
>
>
>
>
>
>
>
> [0]. Loucas Pillaud-Vivien, Alessandro Rudi, and Francis Bach. Statistical optimality of stochastic gradient descent on hard learning problems through multiple passes
>
> [1]. Licong Lin, Jingfeng Wu, Sham M Kakade, Peter L Bartlett, and Jason D Lee. Scaling laws in linear regression: Compute, parameters, and data. arXiv preprint arXiv:2406.08466, 2024
>
> [2]. R. Aguech, E. Moulines, and P. Priouret. On a perturbation approach for the analysis of stochastic tracking algorithms. SIAM J. Control and Optimization, 39(3):872–899, 2000.

---

> > ### Comment · Reviewer_b6Gz · 2025-08-05
> >
> > Thank you for your response. Most of my concerns are adequately addressed. I will raise my scores accordingly.

---

### Official Review · Reviewer_inkZ · 2025-06-28

**Clarity:** 3
**Significance:** 3
**Originality:** 3
**Rating:** 5
**Confidence:** 3

**Summary:**

The paper derives test error bounds when training an M-dimensional linear model on N samples with sketched features using multi-pass SGD of $\Theta(M^{1-b}+L^{(1-b)/a})$ for $L$ iterations, under the standard assumption on data covariance having a power-law spectrum of degree $a$, and the true parameter having a prior with an aligned power-law spectrum of degree $b-a$. This bound holds for $L\leq N^{a/b}$ and improves upon the prior result of $\Theta(M^{1-b}+N^{(1-b)/a})$, for one-pass SGD ($N<L$).

**Questions:**

Some stylistic suggestions to further improve readability:

- The notation paragraph can be moved to Section 2.
- In Corollary 3.3, the part after ‘In contrast’ should be part of the text and not the result statement, since it only discusses a result from prior work.
- In Fig. 1, the last part in the caption (values of M and N for the subfigures) can be included in the respective subcaptions.

In Section 5, the proof overview can be improved by including some more details. The headings in line 258 and 259 seem unnecessary and this part can be made more concise. In the paragraph discussing the main technical difficulty, currently, the details are helpful to contrast the technique with prior work, but some of the statements, such as the one about Lemma D.3, and the one about a ‘novel lower bound’, are not very informative. It would be helpful to include some mathematical expressions for these to make the proof overview easier to follow.

**Ethical Concerns:**

["NO or VERY MINOR ethics concerns only"]

**Final Justification:**

I maintain my recommendation to accept the paper based on the aforementioned strengths.

**Limitations:**

The paper does not have a limitations section. The authors can include a paragraph to discuss some limitations of the work.

**Quality:**

3

**Strengths And Weaknesses:**

This work makes an interesting and important contribution to the recent line of work on theory of scaling laws in simple settings. In particular, prior work considers one-pass SGD while this work considers multi-pass SGD, and shows that data reuse can be helpful to some extent in improving test performance.

The paper is very well-written overall. As someone who has not read other papers on provable scaling laws for simple settings, the paper contains sufficient background information, the authors thoroughly and clearly contrast their contributions, and (parts of) the proof technique with other related works, which was helpful, and discuss well the implications of their result.

---

> ### Author Rebuttal · Authors · 2025-07-31
>
> We  are grateful to Reviewer inkZ for their careful reading of our work and valuable feedback. Please find our responses to the comments below.
>
> ---
>
> > Q1. Some stylistic suggestions to further improve readability:
> >* The notation paragraph can be moved to Section 2.
> >* In Corollary 3.3, the part after ‘In contrast’ should be part of the text and not the result statement, since it only discusses a result from prior work.
> >* In Fig. 1, the last part in the caption (values of M and N for the subfigures) can be included in the respective subcaptions.
>
>  A1. We sincerely appreciate your stylistic suggestions and will revise our paper accordingly.
>
> ---
> > Q2.
> In Section 5, the proof overview can be improved by including some more details. The headings in line 258 and 259 seem unnecessary and this part can be made more concise. In the paragraph discussing the main technical difficulty, currently, the details are helpful to contrast the technique with prior work, but some of the statements, such as the one about Lemma D.3, and the one about a ‘novel lower bound’, are not very informative. It would be helpful to include some mathematical expressions for these to make the proof overview easier to follow.
>
>  A2. Thanks for the suggestions! We will provide a more detailed proof overview in Section 5.
>
> ---
> > Q3.
> The paper does not have a limitations section. The authors can include a paragraph to discuss some limitations of the work.
>
> A3. We will expand our discussion of limitations in the conclusion section. One limitation of the paper is the assumption that the eigenvectors of the prior and data distributions are aligned (implied by Assumption 1D). While this assumption cannot be fully removed without affecting the error rate, it would be interesting to investigate what alternative rates  are achieved when the eigenvectors are not aligned. Another limitation is that our results are currently restricted to linear models; extending them to nonlinear settings such as logistic regression or two-layer neural networks remains an open question.

---

> > ### Comment · Reviewer_inkZ · 2025-08-03
> >
> > Thank you for your response. I will maintain my score.

---

### Official Review · Reviewer_Y8gi · 2025-06-30

**Clarity:** 4
**Significance:** 3
**Originality:** 3
**Rating:** 5
**Confidence:** 2

**Summary:**

PAPER SUMMARY:
This paper provides a theoretical analysis of the effect of multiple SGD passes on scaling laws for linear regression.

Neural scaling laws successfully predict test accuracy as a function of model and data size. Empirically, they proved successful and in the past few years several theoretical explantations of this phenomenon have been proposed.
Recently, the theoretical work of Lin et al., (2024) characterized scaling laws of one-pass SGD for linear regression.
Empirically,  Muennighoff et al. (2023) showed that scaling laws hold for multi-pass SGD as if new data was used, granted that the number of passes is small.

This work explains the latter phenomenon by refining the analysis of Lin et al., (2024) to include multiple passes. In particular, this work retrieves the bounds of Lin et al., (2024) for one pass, and shows that multiple passes behave as if fresh data was used, as long as the number of passes is small enough.

Along with standard assumption in the analysis of GD for linear regression, this work's analysis assumes that the data covariance's and parameter's prior eigenspaces are aligned and both spectra follow a power law, as in Lin et al., (2024).

**Questions:**

No.

**Ethical Concerns:**

["NO or VERY MINOR ethics concerns only"]

**Final Justification:**

I confirm my evaluation and recommend acceptance.

**Limitations:**

Yes.

**Paper Formatting Concerns:**

No.

**Quality:**

4

**Strengths And Weaknesses:**

Despite not being an expert in this field, I believe that this work makes progress towards explaining neural scaling laws.
Indeed, this work explains an important phenomenon already observed empirically, extending and strengthening the illuminating result from Lin et al., (2024).

---

> ### Author Rebuttal · Authors · 2025-07-31
>
> We thank reviewer Y8gi for their positive feedback on our work. We are glad that the reviewer found our work to make progress towards understanding neural scaling laws and to help explain empirical observations such as those in Muennighoff et al. (2023).

---

### Official Review · Reviewer_6mRL · 2025-07-07

**Clarity:** 3
**Significance:** 2
**Originality:** 2
**Rating:** 4
**Confidence:** 3

**Summary:**

The paper analyzes the scaling of test error under SGD on squared loss (regression) for linear models acting on sketched inputs.
The paper considers the setting of Gaussian inputs with a power-law spectrum and a specified source condition quantifying the alignment along the target. The main result shows that upon the re-use of data, i.e., under multi-pass SGD, the number of samples can be replaced by the number of iterations in the scaling law (for a sufficient number of samples). The result thus shows that re-using data provably reduces sample complexity over online SGD, or equivalently, leads to improved scaling laws.

**Questions:**

* Could you provide brief motivations for the introduction of the sketching matrix and the assumption $M<<d$?
* Lines 151-153 appear unclear without looking into Koltchinskii and Lounici (2017). Could you provide a short, self-contained summary of the source of the condition on the initial step-size?
* The parameter b is unspecified in the abstract and only defined later in the source condition.
* What happens when $a<b$?
* What are the central challenges introduced by the introduction of the sketching matrix while bounding the fluctuation error (lines 274-276)?
* What motivates the choice of the geometrically-decaying step-size schedule?

**Ethical Concerns:**

["NO or VERY MINOR ethics concerns only"]

**Final Justification:**

The paper’s contributions are valuable with minor concerns regarding novelty and the scope of the setting.

**Limitations:**

The work is primarily theoretical and does not pose direct societal risks. The paper would benefit from a more detailed discussion of the limitations of their model listed under "weaknesses".

**Paper Formatting Concerns:**

I haven't noticed any major formatting issues in this paper.

**Quality:**

3

**Strengths And Weaknesses:**

#  Strengths
- Understanding scaling laws is a timely question of great interest in the community.
- The paper is well-written with clear statements of the assumptions and theorems. The risk decomposition in (3) and the proof sketch improve the comprehensibility of the results and the proofs.
- The results are supported by numerical simulations.

# Weaknesses

- The setting is limited and to some extent, appears artificial. In particular, the introduction of "sketching matrix" requires further motivation beyond it allowing the introduction of a "model size" parameter. Furthermore, for linear models, it is unclear if the sketching dimension "M" captures the model "size" since the capacity of the model does not grow once the sketching matrix becomes full-rank. The paper further assumes that $M << d$, which precludes the study of overparameterized regimes.
- The analysis is of limited novelty and largely builds upon existing works, in particular the work of Pillaud-Vivien et al. 2018.
- (minor) missing related work. Although for different reasons, another recent line of work also demonstrates improved sample complexity (and consequently, improved scaling laws by data reuse) in the feature learning regime for two-layer neural networks:

  Dandi, Y., Troiani, E., Arnaboldi, L., Pesce, L., Zdeborova, L., &; Krzakala, F. The Benefits of Reusing Batches for Gradient Descent in Two-Layer Networks: Breaking the Curse of Information and Leap Exponents. In Forty-first International Conference on Machine Learning.

  Lee, J. D., Oko, K., Suzuki, T., &amp; Wu, D. (2024). Neural network learns low-dimensional polynomials with sgd near the information-theoretic limit. Advances in Neural Information Processing Systems, 37, 58716-58756.

  Arnaboldi, L., Dandi, Y., Krzakala, F., Pesce, L., &amp; Stephan, L. (2024). Repetita iuvant: Data repetition allows sgd to learn high-dimensional multi-index functions. arXiv preprint arXiv:2405.15459.

---

> ### Author Rebuttal · Authors · 2025-07-31
>
> We are grateful to Reviewer 6mRL for their thorough examination of our work and for their helpful comments. We hope to address all questions below.
>
> ---
> > W1. The setting is limited and to some extent, appears artificial. In particular, the introduction of "sketching matrix" requires further motivation beyond it allowing the introduction of a "model size" parameter. Furthermore, for linear models, it is unclear if the sketching dimension "M" captures the model "size" since the capacity of the model does not grow once the sketching matrix becomes full-rank. The paper further assumes that $M\ll d$ , which precludes the study of overparameterized regimes.
>
> A1.
> The random sketching is introduced in our setting to control the dimension $M$ (or model size) of the linear models. We believe it is a basic and natural approach to introduce linear models with varying dimensions (model complexity) in regression problems. Similar random sketching approaches have been used to control model dimensions in many other works on the theoretical understanding of scaling laws [1,2,3].
>
> We respectfully disagree with the reviewer's point that “the model size does not grow once the sketching matrix becomes full-rank.” Note that we consider an infinite-dimensional linear regression setting where the dimension $d = \infty$ (see line 66 and e.g., Assumption 1C). Therefore, increasing the sketching dimension $M$ always increases the model capacity. Our setting does not preclude overparameterized regimes either, since our results hold for any finite sample size $N$ and model size $M$ (e.g., Corollary 3.2), covering both underparameterized regimes ($M \leq N$) and overparameterized regimes ($M > N$).
>
> ---
> > W2. The analysis is of limited novelty and largely builds upon existing works, in particular the work of Pillaud-Vivien et al. 2018.
>
> A2. We respectively disagree that our analysis has limited novelty.
> For multi-pass SGD, we generalize the results of Pillaud-Vivien et al. (2018) [4] in two key aspects.
> First, the upper bound on the risk of multi-pass SGD in Theorem 1 of [4] is restricted to the regime $\mu a \leq b \leq a$ for some $\mu \in (0,1]$. (It can be verified that our parameters $(a, b)$ equal  $(\alpha, 2\alpha r + 1)$ in  their work, and note that choosing $ \mu = 0 $ in their Assumption A3 leads to $\kappa = \infty $, making the bound vacuous.) In contrast, our upper bounds in Corollaries 3.3 and 3.4 apply to a strictly broader regime $ a> b-1 $.
> Second, [4] provides no lower bounds for the risk of multi-pass SGD. In contrast, we establish matching lower bounds in the regime $a>b-1 $, verifying the tightness of our upper bounds. We believe these generalizations are not possible without introducing technical innovations.  Note: the condition $b > a$ in lines 219–221 is a typo and should be revised to $b > a > b - 1$. Similarly, lines 129 and 169 should include the condition $a > b - 1$.
>
> More specifically, for the fluctuation error, we follow the standard practice as in [4] and the early work by Aguech et al., 2000 [6] to express the difference between the multi-pass SGD and GD trajectories, $\mathbf{v}_t -\boldsymbol{\theta}_t$ as a stochastic process (Eq. 17). We then bound the fluctuation error $\mathbb{E}[\\|\mathbf{\Sigma}^{1/2}(\mathbf{v}\_L - \boldsymbol{\theta}_L)\\|^2]$ through controlling the accumulated error of the stochastic process using Lemma D.2 and Lemma D.3, which involves a novel leave-one-out argument to control the model parameters.
>  Note that our Lemma D.2 also generalizes the corresponding results in [4], as the analysis in [4] does not apply when $a=b$ for infinite-dimensional problems. In addition, we derive a novel lower bound on the fluctuation error that matches the upper bound up to logarithmic factors in certain regimes (Lemma D.4 and D.5).
>
> ---
> > W3. (minor) missing related work. Although for different reasons, another recent line of work also demonstrates improved sample complexity (and consequently, improved scaling laws by data reuse) in the feature learning regime for two-layer neural networks…
>
> A3. We thank the reviewer for pointing out the related works on improved sample convexity in two-layer neural networks via data reuse. We agree that data reuse can indeed improve sample complexity in many other regimes. However,  a key distinction in our work is the use of random sketching, which introduces additional technical challenges. We will discuss these works and their connections to our results in the paper.
>
> ---
> > Q1. Could you provide brief motivations for the introduction of the sketching matrix and the assumption $M\ll d$?
>
> A4. As explained in A1, we believe random sketching is a basic and natural approach to allow us to analyze a sequence of linear models with varying dimensions (model complexity) in regression problems [1,2,3]. The assumption $d=\infty$ is made to mimic the real-world scenario where the true data distribution is highly complicated (e.g., in text data) and no model with a finite size $M$ can capture all information about the data.
>
> ---
>
> > Q2. Lines 151-153 appear unclear without looking into Koltchinskii and Lounici (2017). Could you provide a short, self-contained summary of the source of the condition on the initial step-size?
>
> A5. Since gradient descent is known to converge when the learning rate $\gamma \leq 1/L$ for convex objectives that are $L$-smooth, it suffices in our linear regression setting to choose $\gamma \leq 1/L = 1/\|\mathbf{S}\mathbf{X}^\top\mathbf{X}\mathbf{S}^\top/N\|_2 \asymp 1$. The final equivalence follows from concentration properties of Gaussian covariance matrices (see, e.g., Koltchinskii and Lounici, 2017).
>
> ---
> > Q3. The parameter $b$ is unspecified in the abstract and only defined later in the source condition.
>
> A6. The parameter $b$ is introduced in the form of $b-a$ in the abstract. We will clarify this in the revision.
>
> ---
>
> > Q4. What happens when $a<b$?
>
> A7. When $a<b<a+1$, as shown in Corollary 3.4 and the subsequent discussion, the optimal rate in $N$ is obtained when using the data once (i.e., $L_{\mathsf{eff}}=N$) and the learning rate $\gamma = L_{\mathsf{eff}}^{a/b-1}$. In this case, we are able to obtain the same rate $\widetilde{O}(N^{\frac{1-b}{b}})$ as in [3] for one-pass SGD. The case $b\geq a+1$ remains unclear to us and is left for future investigation.
>
> ---
> > Q5. What are the central challenges introduced by the introduction of the sketching matrix while bounding the fluctuation error (lines 274-276)?
>
> A8. Sketching introduces additional difficulties when translating the general bounds on the fluctuation error to the setting where the source condition holds (Lemma D.5), as it requires precise characterization of the spectrum of the sketched covariance $\mathbf{S}\mathbf{H}\mathbf{S}^\top$ to control the terms in the general bound (line 750). We will further clarify this in the revision.
>
>
>
> ---
> > Q6. What motivates the choice of the geometrically-decaying step-size schedule?
>
> A9. As discussed in our work (lines 82-85), our analysis applies to other stepsize schedules (such as polynomial decay) as well, but we focus on geometric decay since it is known to yield near minimax optimal excess test error for the last iterate of SGD in the finite-dimensional regime [5].
>
> ---
>
> [1]. Bordelon, Blake, Alexander Atanasov, and Cengiz Pehlevan. "A dynamical model of neural scaling laws." arXiv preprint arXiv:2402.01092 (2024)
>
> [2]. Paquette, Elliot, et al. "4+ 3 phases of compute-optimal neural scaling laws." Advances in Neural Information Processing Systems 37 (2024): 16459-16537.
>
> [3]. Lin, Licong, et al. "Scaling laws in linear regression: Compute, parameters, and data." Advances in Neural Information Processing Systems 37 (2024): 60556-60606.
>
> [4]. Pillaud-Vivien, Loucas, Alessandro Rudi, and Francis Bach. "Statistical optimality of stochastic gradient descent on hard learning problems through multiple passes." Advances in Neural Information Processing Systems 31 (2018).
>
> [5]. Ge, Rong, et al. "The step decay schedule: A near optimal, geometrically decaying learning rate procedure for least squares." Advances in neural information processing systems32 (2019).
>
> [6]. R. Aguech, E. Moulines, and P. Priouret. On a perturbation approach for the analysis of stochastic tracking algorithms. SIAM J. Control and Optimization, 39(3):872–899, 2000.

---

> > ### Comment · Reviewer_6mRL · 2025-08-09
> >
> > I thank the authors for the detailed clarifications, which would strengthen the paper if incorporated. I have increased my score.

---

### Decision · Program_Chairs · 2025-09-17

**Decision:**

Accept (poster)

**Comment:**

This paper advances the theory of scaling laws in simple settings, in this case sketched linear regression with power law source/capacity type conditions. The technical aspects of the paper appear sound, with new theoretical innovations and effective support from numerical simulations. The authors also provide lower bounds on errors, which give tight estimates.  The work makes an important contribution by extending the theory of scaling laws to multi-pass SGD, which requires new analysis beyond what was done in the one-pass case. It successfully shows the benefits of data reuse in improving test performance in one regime, which was enlarged in earlier work of Pillaud-Vivien et al. in kernel regression.

The paper is clearly written and easy to follow, with clear theorem statements. The comparison with related works was generally found to be complete.

The two closest existing works are those of Lin et al and Pillau-Vivien et al.  In comparison to the work of Pillaud-Vivien et al, which first established the benefits of data-reuse in a similar linear problem, this paper:
* covers a larger range of parameters, including the whole regime in which multipass-SGD is expected to improve over one-pass.
* incorporates a random sketch matrix, which allows the model to have parameterized capacity at the cost of additional technical difficulty.
* provides instance-dependent upper and lower bounds.

The paper also extends existing work Lin et al, which focused on single-pass SGD, by working with multipass-SGD, in a nearly identical setup.